

# Dinosaur track assemblages from mid-Cretaceous of Fujian Province, southeastern China: ichnotaxonomic review and faunal comparison

Lida Xing[1,2,*], Kecheng Niu[3,4,*], Qiyan Chen[2], Hendrik Klein[5], Anthony Romilio[6], Runsheng Chen[7], Min Lin[7], Ke Deng[7] and Jianrong Tang[7]

[1] Frontiers Science Center for Deep-time Digital Earth, China University of Geoscience (Beijing), Beijing, China
[2] School of the Earth Sciences and Resources, China University of Geoscience (Beijing), Beijing, China
[3] State Key Laboratory of Cellular Stress Biology, School of Life Sciences, Xiamen University, Xiamen, Fujian, China
[4] Yingliang Stone Natural History Museum, Nan'an, Fujian, China
[5] Saurierwelt Paläontologisches Museum, Neumarkt, Bavaria, Germany
[6] School of the Veterinary Science, University of Queensland, Gatton, Queensland, Australia
[7] Fujian Institute of Geological Survey, Fuzhou, Fujian, China
* These authors contributed equally to this work.

Corresponding authors
Lida Xing, xinglida@gmail.com
Qiyan Chen, qiyan_chen@163.com

## ABSTRACT

Among the the mid-Cretaceous strata in China, considerable dinosaur record are preserved in the southeastern mountainous and arc-related basins. The Shanghang Basin is one of the sporadic red-stratified basins distributed in western Fujian, SE China, and has previously been discovered as the home of an ornithopod-dominaited ichnofauna, which is also characterized by the large troodontid ichnogenera *Fujianipus*. Include the newly discovered fossils, further confirming that this tracksite is dominated by ornithopods, characterized by a significant proportion (>27%) of large ornithopods, with deinonychosaurians as the possible apex predators. As the only extensive mid-Cretaceous dinosaur tracksite in southeastern China, Longxing offers a temporal comparison with skeletal records from Zhejiang (SE China) and other fauna globally. The Longxiang herbivorous assemblage may suggesting limited faunal turnover by aligning more closely with pre-Cenomanian than the contemporaneous faunas in Zhejiang. Besides, unlike South American patterns associated with OAE2, the faunal shifts of SE China are relatively mild and appear more influenced by regional factors—topographic barriers and volcanic activities—rather than global climatic signal affected by marine conditions. Further research is needed to refine faunal chronology and assess the impact of regional environmental factors in shaping Cretaceous ecosystems of SE China.

## INTRODUCTION

Considerable dinosaur records, including fossil skeletons, eggs and tracks, are preserved in the Late Cretaceous reddish-dominated deposits in southeastern and southern China, including Zhejiang, Jiangxi and Guangdong, especially in the Jinqu Basin of Zhejiang (*Yu, 2013*; *Du et al., 2015*), the Ganzhou Basin of Jiangxi (*Lü et al., 2007*, *2016*; *Xing et al., 2020*) and the Nanxiong Basin of Guangdong (*Xing et al., 2017*, *2020*). Among these provices, Fujian has now become one of the emerging hotspots for Mesozoic dinosaur evolutionary studies in recent years, especially for the bird-line theropods, as the local geological surveys and the excavation of newly discovered fossil fauna continued (*Xu et al., 2023*; *Xing et al., 2024a*; *Chen et al., 2025*).

In November 2020, the Dinosaur Laboratory of China University of Geosciences, Beijing, in cooperation with the Yingliang Stone Natural History Museum, started to search for dinosaur fossils in the red beds of Fujian Province. The preliminary findings from the core area (upper and lower Longxiang site I, LXIU and LXID) and a few scattered specimens (Longxiang site II, LXII) of Kejiayuan Cultural Centre in Longxiang Village, Shanghang Basin is the first time that dinosaur tracks have been identified in Fujian Province (Fig. 1), and was described by *Niu & Xing (2023)*. The largest known specimen of troodontid tracks by far, subsequently named *Fujianipus*, was also found during this fieldwork (*Xing et al., 2024a*). In January 2021, the lead author of this article found a third footprint site adjacent to the core area (Longxiang site IN and III, LXIN and LXIII). In April of the same year, a detailed investigation of the site LXIN was carried out. Thereafter, in October 2022, an additional isolated slab was found in eastern Longxiang site I (LXIE) by Kecheng Niu.

This study focuses on describing the discovery of new specimens during fieldworks in 2021 and 2022. On this basis, this study will also synthesise the record of the entire Longxiang site and compare its ichnofauna with other skeletal faunas from East and Northeast Asia of the same period, focusing on similarities and differences between the faunas of the region, as well as the possible impact of environmental and preservation differences on the occurrence of distinctive members at the Longxiang site.

## GEOLOGICAL SETTING

### The tectonic setting of Fujian Province in early Cretaceous

Southeastern China is situated on the southeastern margin of the Eurasian Plate. During the Late Jurassic to Early Cretaceous, intense volcanism occurred, leading to the formation of a widely distributed, thick terrestrial volcanic-sedimentary rock system in the eastern part of the province (Fig. 2). In the latest Early Cretaceous to Late Cretaceous, the intensity of volcanism decreased, resulting in volcanic or nonvolcanic terrestrial red-bed arc-related basin deposits (Fig. 3; *Li, 1997*; *Xi et al., 2019*).

After the end of the subduction stage on the eastern side of the Cathaysia block in the Cretaceous, the tectonic stage in southeastern China shifted from a syn-orogenic shortening stage to a post-orogenic stage (*Li et al., 2014*). This stage was mainly represented by NW-SE extension in the late Valanginian to early Aptian. Subsequently, the

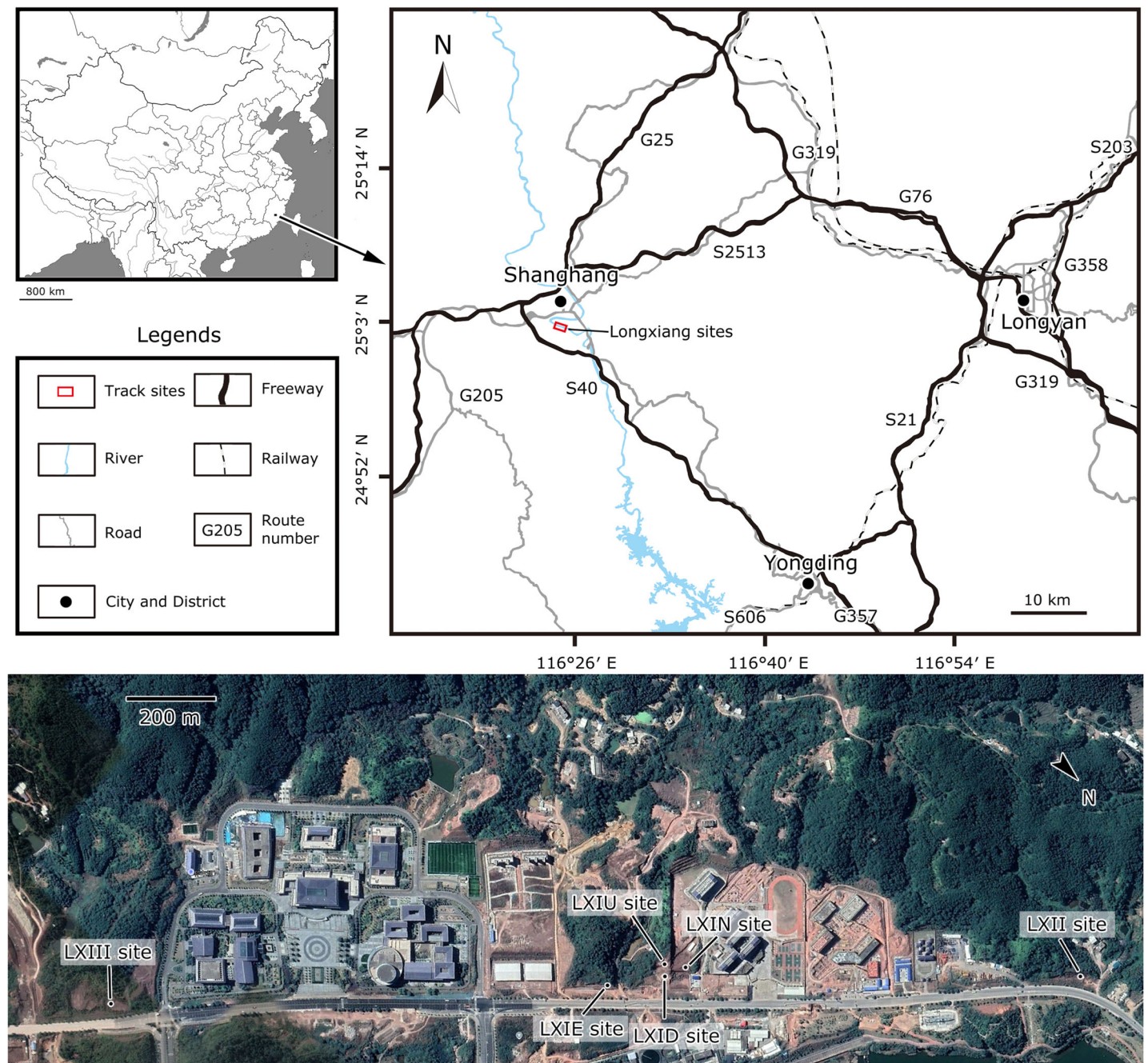

**Figure 1 Location of the Longxiang sites in Fujian Province, southeastern China (from *Niu & Xing, 2023*).**

collision of the South China continental crust and the West Philippine Plate transformed the region into an isotropic compressional event in the late Aptian to early Albian (*Charvet, Lapierre & Yu, 1994*; *Li et al., 2014*). During the compressional phase, the mountains included the area covered by Paleo-Yunmengze Lake and the associated inland river system, *i.e.*, most of Hubei, northern and northwestern Hunan, and part of Henan

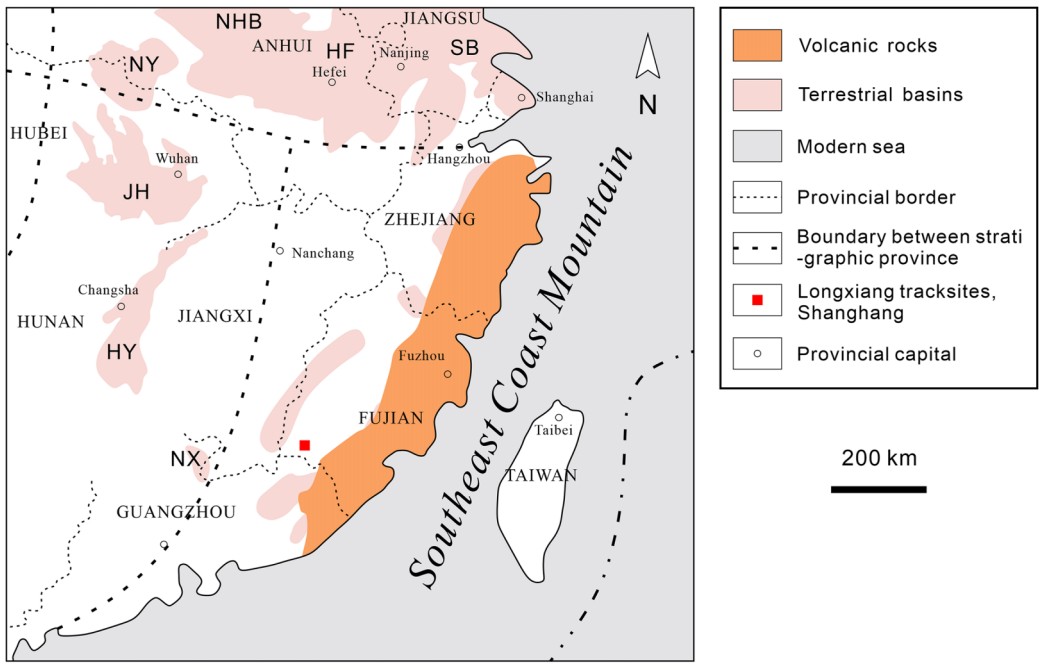

**Figure 2 Distribution of Cretaceous deposits in southeastern China.** HF, Hefei Basin; HY, Hengyang Basin; JH, Jianghan Basin; NHB, South China Basin; NX, Nanxiong Basin; NY, Nanyang Basin; SB, Subei Basin (modified after *Cao, 2018*; *Xi et al., 2019* and *Wang et al., 2022*).

and southwestern Jiangxi (*Chen, 1979*; *Compilation Committee of Geological Atlas of China, 2002*; *Xiaoqiao, Peiji & Mingjian, 2007*). The high and wide southeastern coast mountains made it difficult for the paleo-Pacific moisture flow to reach the above regions, causing these areas to develop into tropical-subtropical hot plains, hills and desert landscapes (*Chen, 1979*), where gypsum and halite were deposited (*Xiaoqiao, Peiji & Mingjian, 2007*). To the east and south of this region are dominated by plains and hills belonging to Zhejiang, Fujian, Jiangxi and Guangdong regions, where dinosaurs lived in these small and sporadic intermountain arc-related basins (*Chen, 2000*; *Li et al., 2013*).

Since the Aptian, SE China began to gradually move out of the dry climate phase, however, the mid- to Upper Cretaceous faulted, red basin composed of brick-red siltstone implies a long warm (or hot)-wet climate phase thereafter (*Wang et al., 2022*), and also containing gypsum layers, with conglomerates and sandy conglomerates at the base (*Zhou, 2007*).

## The geological background of Longxiang site

In the late Albian to Late Cretaceous, extensive layers of purplish-red beds were developed in west-central Fujian, divided into the lower Shaxian Formation and the upper Chong'an Formation (*Li, 1997*; *Xi et al., 2019*), and the latter is dominated by coarser-grained conglomerates (*Lü et al., 2019*).

In western Fujian, there are about 36 red bed-dominated Cretaceous basins spatially distributed in a northeast-trending belt in the west of the Zhenghe-Daipu Rift. The

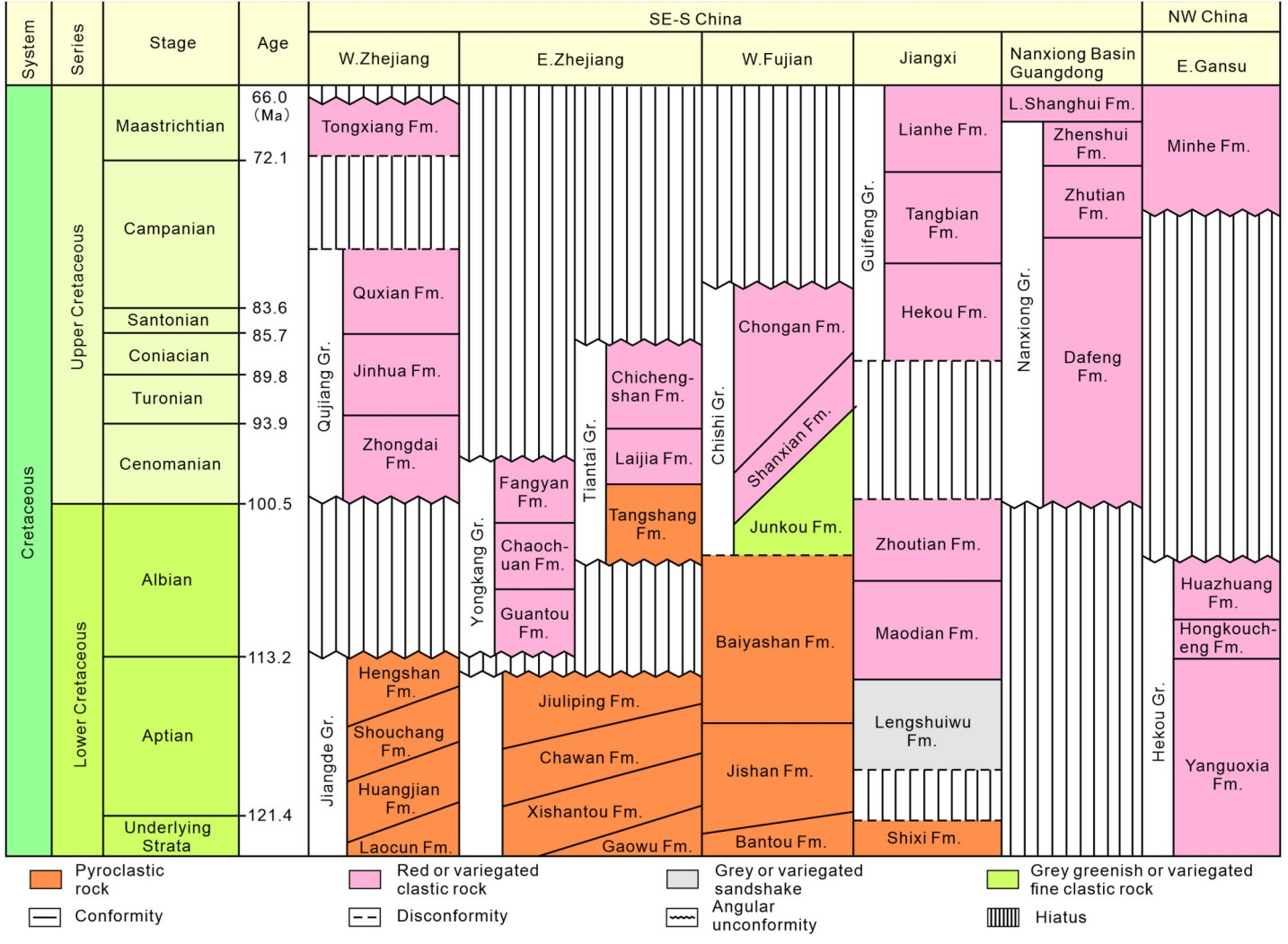

**Figure 3** Stratigraphic subdivision and correlation of Cretaceous strata in the typical stratigraphic provinces of southeastern, south-central and northwestern China (modified after *Xi et al., 2019*).

Shanghang Basin is one of the small basins in southwestern Fujian Province, where the Cretaceous red beds are a sequence of purplish-red coarse-fine clastic assemblages. The Shanghang Basin is dominated by the Chishi Group (Junkou, Shaxian and Chong'an Formations), of which the Shaxian Formation is the most widely distributed (*Fujian Institute of Geological Survey, 2016*).

The lithology of the Shaxian Formation consists mainly of purplish-red medium-thick bedded calcareous, fine sandstone and muddy siltstone, interspersed with purplish-red sandstone, and complex-composed conglomerate, while regionally interbedded with tuff, tuff lava, and marl, which is composed of terrestrial red clastic rocks. In Shanghang Basin, the maximum thickness of Shaxian Formation is about 1,913 m.

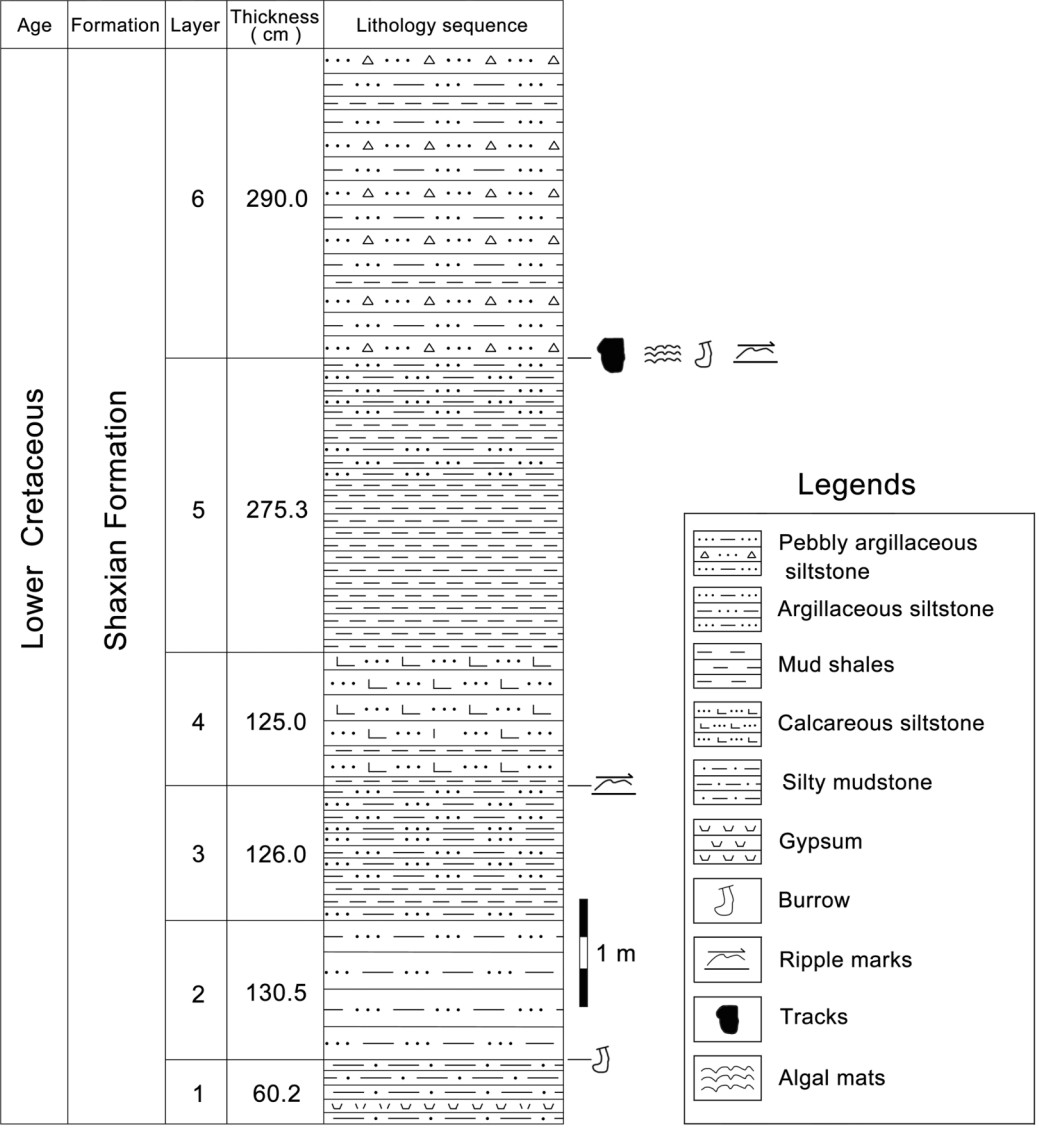

**Figure 4 The track-bearing layer of Longxiang tracksite in the Cretaceous Shaxian Formation from Shanghang Basin, Fujian, China.** The upper part of Shaxian Formation is not exposed at the Longxiang site.

According to the 1:200,000 regional geological survey report of Shanghang Area (G-50-27), the dinosaur tracks from Longxiang Village are preserved in the lower part of the Shaxian Formation (Figs. 3, 4).

At the Longxiang site, a continuous section is exposed which successively develops the alluvial fan fringe deposits (layers 1–12), the braided river channel and point bar deposits (layers 13–17), the shore lake and shallow lake deposits (layers 18–34). The lacustrine deposits in the Longxiang site are mainly dominated by graywacke, fine sandstones and siltstones, interspersed with thin layers of mudstones, with little gravel content. The grain size of the sediments is generally only about 2–10 mm, with a maximum of 2 cm. The

dinosaur tracks are found near the top of this section (layer 31), which is a medium-thin-bedded, light purple-red muddy siltstone. The upper beds are darker, thinly bedded silty claystone.

The sediments in the adjacent area also include thin gypsum layers and calcareous clasts, with local occurrences of copper (Cu)-bearing sandstones (*Chen, 2008*).

As with many Late Mesozoic terrestrial deposits in China, isotope chronology data for the Shaxian Formation in the Shanghang Basin are limited due to frequent volcanic and tectonic activity. *Hu (1990)* suggested that the depositional age of the Shaxian Formation in the red bed of Shanghang is between 80 and 105 Ma, based on magnetic stratigraphy. The $^{206}Pb/^{238}U$ dates of the Shaxian Formation in the Shanghang Basin vary from 87 to 101 Ma, with a weighted average of 96.0 ± 4.1 Ma (*Chen et al., 2020*). Therefore, the Shaxian Formation in the Shanghang Basin can be assigned to the latest Early Cretaceous to the early Late Cretaceous, which lies roughly within the Albian to Coniacian, with a weighted average in the Cenomanian.

Based on palynology (*Zheng & Li, 1986*; *Liang, Cao & Ma, 1992*) and paleosoil evidence (*Li et al., 2009*), the Shaxian Formation is a riverine and lacustrine detrital deposit that formed in an inland arc-related basin under a hot, dry and oxidizing environment (*Li, 1997*). Based on the evidence of stratigraphic magnetic characterization, *Lü et al. (2019)* suggested that hematite in the red strata of the Shaxian Formation and Chong'an Formation is indicative of a high-temperature climatic environment.

# MATERIALS AND METHODS

## Materials

At least three dinosaur tracksites were discovered in Longxiang Village, which are numbered as LXIs (GPS: 25°2′15.69″N, 116°23′58.29″E), LXII (GPS: 25°2′38.20″N, 116°23′35.37″E) and LXIII (GPS: 25°1′48.36″N, 116°24′32.25″E) (Fig. 1).

Site LXI is the main tracksite, divided into the upper (LXIU), lower (LXID), northern (LXIN) and eastern (LXIE) parts (Figs. 5–7), with a total area of approximately 1,600 m². These areas are originally from the same track surface. However, the connections between these regions were severed during the excavation of the tracksite, and the faults created by the excavation leave these regions with some differences in their current positions on the outcrop. There are currently more than 700 tracks exposed on the site, including 79 trackways and more than 100 isolated tracks (trackways see Table 1). Among the Longxiang tracks, 274 tracks in LXIU abd LXID has been already reported in *Niu & Xing (2023)*.

With the exception of the very large didactyl tracks, the vast majority of Longxiang footprints can be confirmed not to be undertracks. Some footprints, especially the more complete tridactyl tracks, show impressions of phalangeal pads or preserve traces left by sediment flow within the tracks. Furthermore, the deformation between these tracks is mostly continuous and can be correlated with one another. Some of these trackways even exhibit characteristics of penetrate tracks, with digit traces significantly narrowing, accompanied by sediment bulging inward from both sides of the footprint.

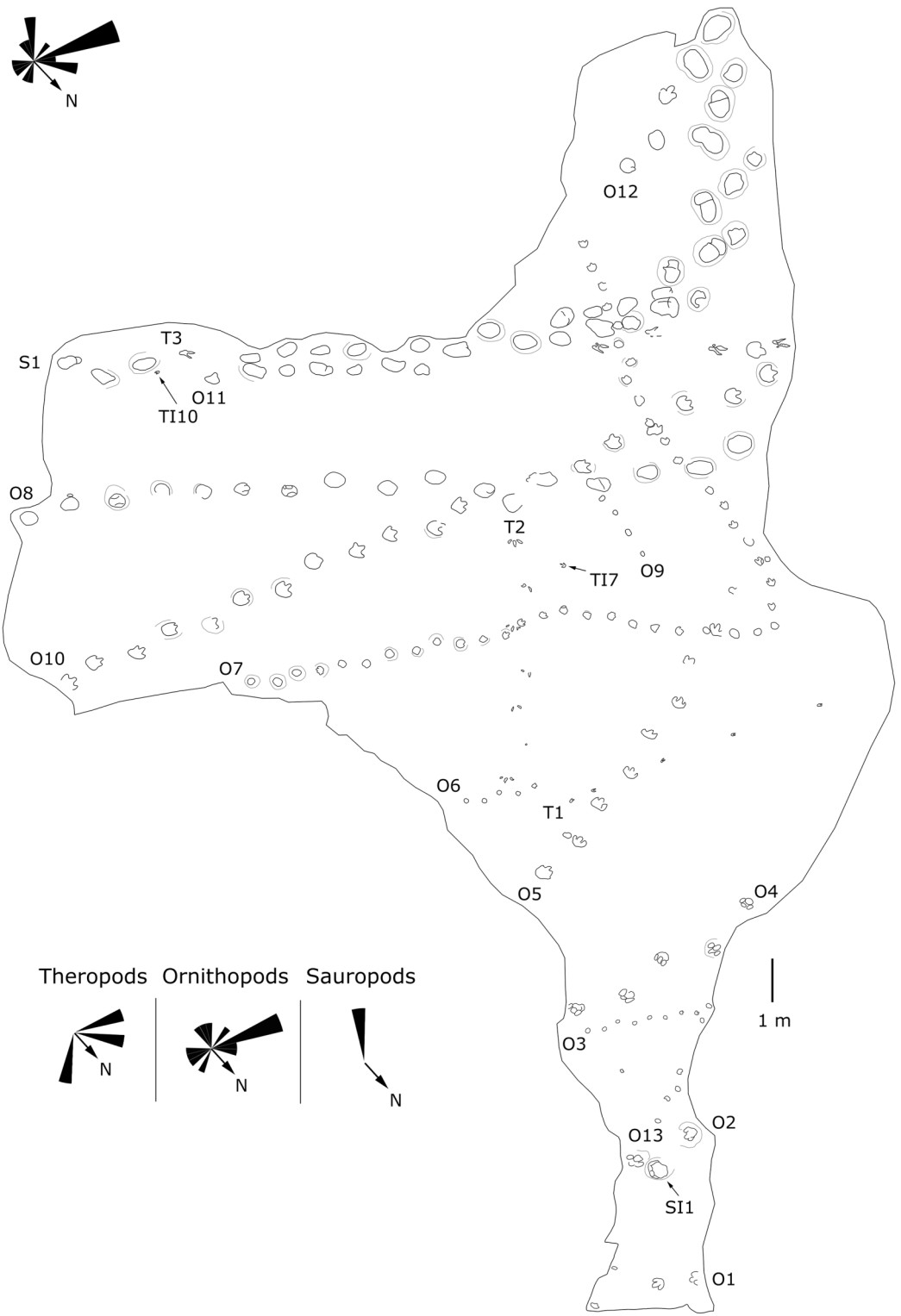

**Figure 5 Mapping of the Longxiang IU (LXIU) tracksite and the related trackway orientation rose diagrams.**

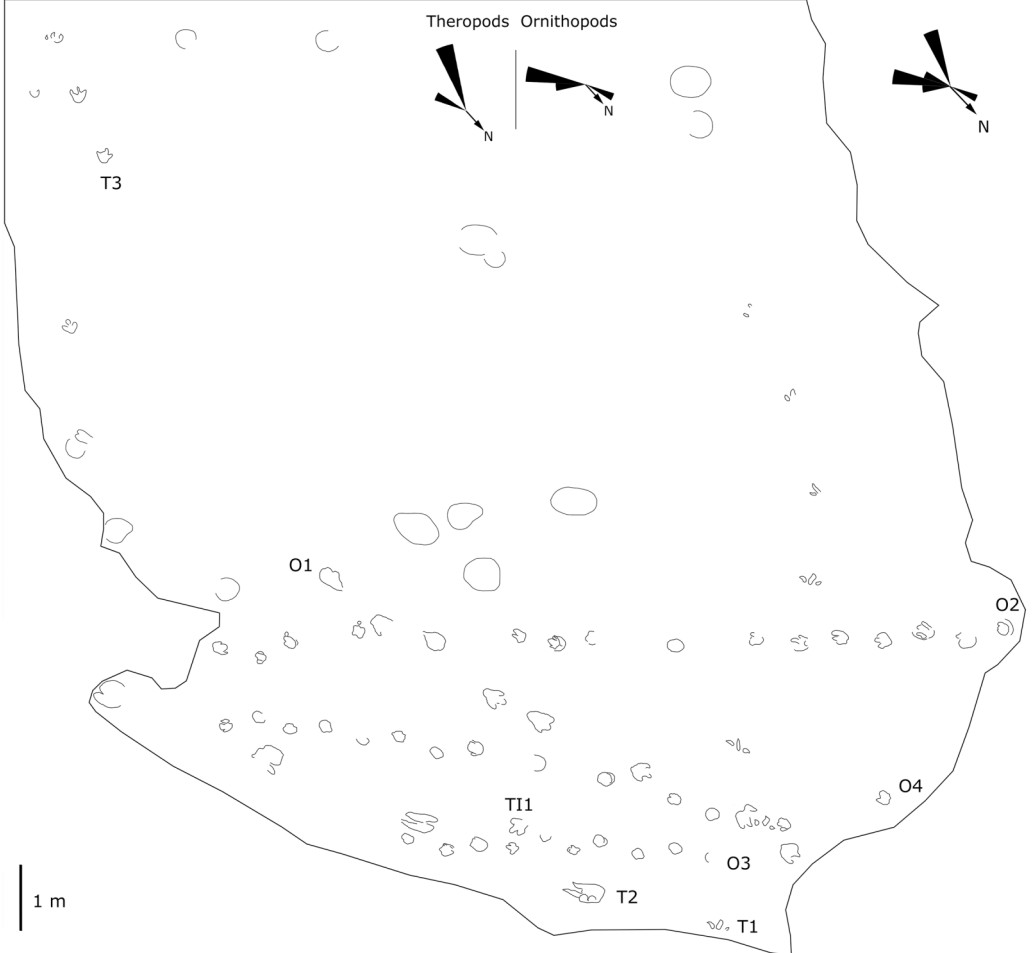

**Figure 6 Mapping of the Longxiang ID (LXID) tracksite and the related trackway orientation rose diagrams.**

The LXIU site covers an area of about 320 m², with 17 trackways and eight relatively clear isolated tracks, making a total of about 229 tracks (Fig. 5). Site LXID covers an area of about 170 m², with a large number of root trace, seven trackways and one relatively clear isolated track, totalling about 81 tracks (Fig. 6). The LXIN tracksite covered an area of about 290 m², with a large number of root trace, 54 trackways and 31 relatively clear isolated tracks, totaling about 398 tracks (Fig. 7). The LXIE site only including one slab (YLSNHM07318), with one trackway of two tracks. Site LXII is located approximately 950 m northwest of Site I and preserves only three isolated tracks (Fig. 8). Site LXIII is located approximately 1,250 m southeast of Site I and preserves only six isolated tracks (Fig. 8). The vast majority of the tracks at these sites are relatively well preserved, with only a small number (<10%) preserved in the form of scattered, structurally ambiguous shallow pits. The tracks in very poor preservation condition are not considered in this article.

YLSNHM07318 was found at the site of the landslide on the east side of the previously described Longxiang tracksite (25°2′13.76″N, 116°24′2.70″E), with light purple-red silty

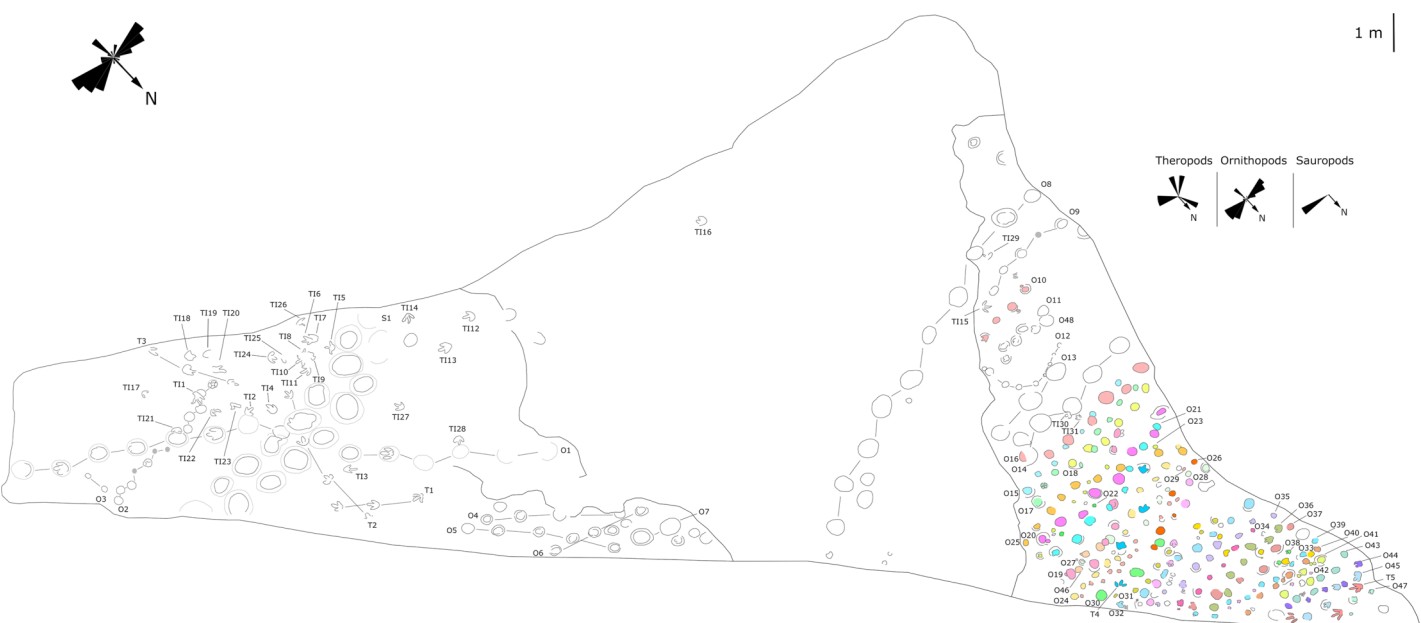

**Figure 7 Mapping of the Longxiang IN (LXIN) tracksite and the related trackway orientation rose diagrams.**

**Table 1 The ichnofanua composition of the LXI tracksites.**

|  | Frequency | Probability | Large/total, ornithopod |
|---|---|---|---|
| Total (LXIs included) |  |  |  |
| Total | 79 |  |  |
| Ornithopods | 66 | 0.84 | 0.27 |
| Sauropods | 2 | 0.03 |  |
| Theropods | 11 | 0.14 |  |
| LXIN |  |  |  |
| Total | 54 |  |  |
| Ornithopods | 48 | 0.89 | 0.19 |
| Sauropods | 1 | 0.02 |  |
| Theropods | 5 | 0.09 |  |
| LXID |  |  |  |
| Total | 7 |  |  |
| Ornithopods | 4 | 0.57 | 0.25 |
| Sauropods | 0 | 0 |  |
| Theropods | 3 | 0.43 |  |
| LXIU |  |  |  |
| Total | 17 |  |  |
| Ornithopods | 13 | 0.76 | 0.62 |
| Sauropods | 1 | 0.06 |  |
| Theropods | 3 | 0.18 |  |

|  | Frequency | Probability | Large/total, ornithopod |
|---|---|---|---|
| LXIE |  |  |  |
| Total | 1 |  |  |
| Ornithopods | 1 | 1 | 0 |
| Sauropods | 0 | 0 |  |
| Theropods | 0 | 0 |  |

Note:
The frequency of each trackmaker type is represented by the number of trackways, and it should be noted that the sole isolated track from the *Grallator* morphotype is the only trackway counted as such due to its uniqueness. The interval of large ornithopod includes all the trackways with a mean track length of ≥25 cm.

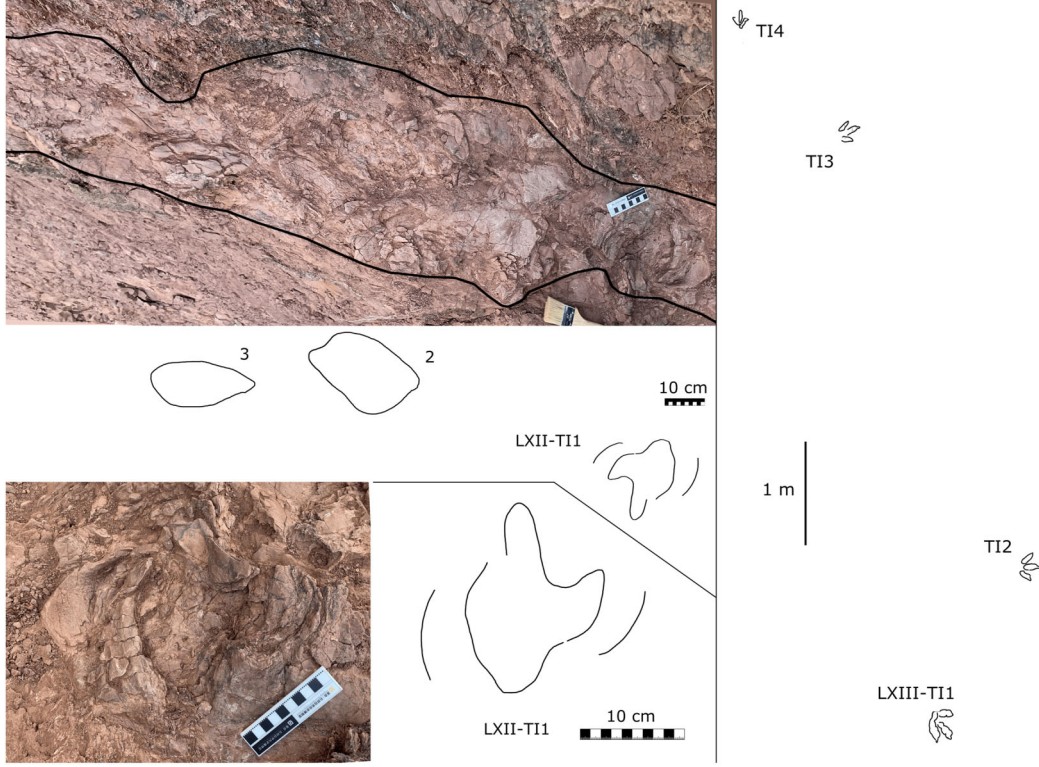

**Figure 8 Photographs and interpretive line drawings of the dinosaur tracks from Longxiang II (LXII) tracksite and mapping of Longxiang III (LXIII) tracksite.**

claystone. In the early rainy season, the mountain slid down the slope and formed some fragmented rock specimens with preserved dinosaur tracks. This slab is now preserved in Yingliang Stone Natural History Museum, Nan'an, China.

## METHODS

All the exposed footprints were photographed, outlined with chalk, and traced on large sheets of transparent plastics. In addition, a representative area of well-preserved tracks was mapped manually using a simple chalk grid. Latex molds of representative tracks were

made. Detailed tracings of selected tracks were made on transparent acetate film. Latex molds, plaster replicas, and most tracings were reposited at China University of Geosciences, Beijing.

The entire exposed surface was photographically recorded using a remote controlled four axis quadcopter (DJI Inspire 1: weight: 3,400 g; max service ceiling above sea level: 4,500 m; max flight time: 15 min; max wind speed resistance: 10 m/s and with DJI GO App, iOS 8.0 or later) with a 12 mega–pixel camera (model X5, with a 15 mm lens). After taking off from the ground, the DJI Inspire 1 was controlled by remote and it provide real–time HD video through a mobile APP (DJI GO version 3.1.23).

Digital 3D models were created of the *in situ* track-bearing surface following photogrammetry methods outlined by *Romilio (2020)*. Digital photographs were taken from multiple viewpoints of the *in situ* tracks with an Apple iPhone XS Max (focal length 4.25 mm). Virtual 3D models were created following the step-by-step process outlined by *Romilio (2020)*, which included adding photographs to Agisoft Metashape Professional (v.1.6.3), repositioning and centering models using Meshlab (*Cignoni et al., 2008*), and visualising the surface topography using Paraview (v. 2020.06; *Ahrens, Geveci & Law, 2005*) and CloudCompare (v. 2.10.2; http://www.cloudcompare.org/) filters.

Maximum length, maximum width, pace length, stride length, pace angulation and rotation of tracks were measured according to the standard procedures of *Leonardi (1987)* and *Lockley & Hunt (1995)*. For the trackways of quadrupeds, gauge (trackway width) was quantified for pes and manus tracks using the ratio WAP/P'ML (*Marty et al., 2010*). The distance between the pes and manus imprints was measured from the proximal margin of the manus to the distal margin of the pes following the method of *Xing et al. (2014a)*. Hip heights and speed estimations of the theropod, sauropod and ornithopods trackmakers were derived from the trackways following the methods of *Alexander (1976)*, *Thulborn (1990)*, and *González Riga (2011)* respectively (see below).

## RESULTS

### Sauropod tracks

#### Description

The sauropods trackways at the Longxiang tracksite are mainly distributed in the site LXIU and LXIN (Figs. 5, 7, 9, 10; Table 2).

The site LXIU contains medium-large sized sauropod trackway, with a specimen number of LXIU-S1, containing 48 tracks in 24 pairs of manus-pes sets (Figs. 5, 9). LXIU-S1 is medium-large in size, with a preservation of 0–1 from *Belvedere & Farlow (2016)*, with distinguishable manus and pes tracks retaining only general outlines. Due to the tight spacing between the manus and pes of LXIU-S1, it is more difficult to accurately recognise the boundaries between the two, so only the general morphology of the two was taken into account when making regular measurements. Their total length was 55.4 cm and the width was 33.2 cm for an elliptical impression, with an L/W ratio of 1.7. Judging from some of the tracks with a relatively clear separation between the manus and pes tracks, the pes tracks averaged 40.8 cm in length, ~74% of the total length, and 33.2 cm in width, with an L/W

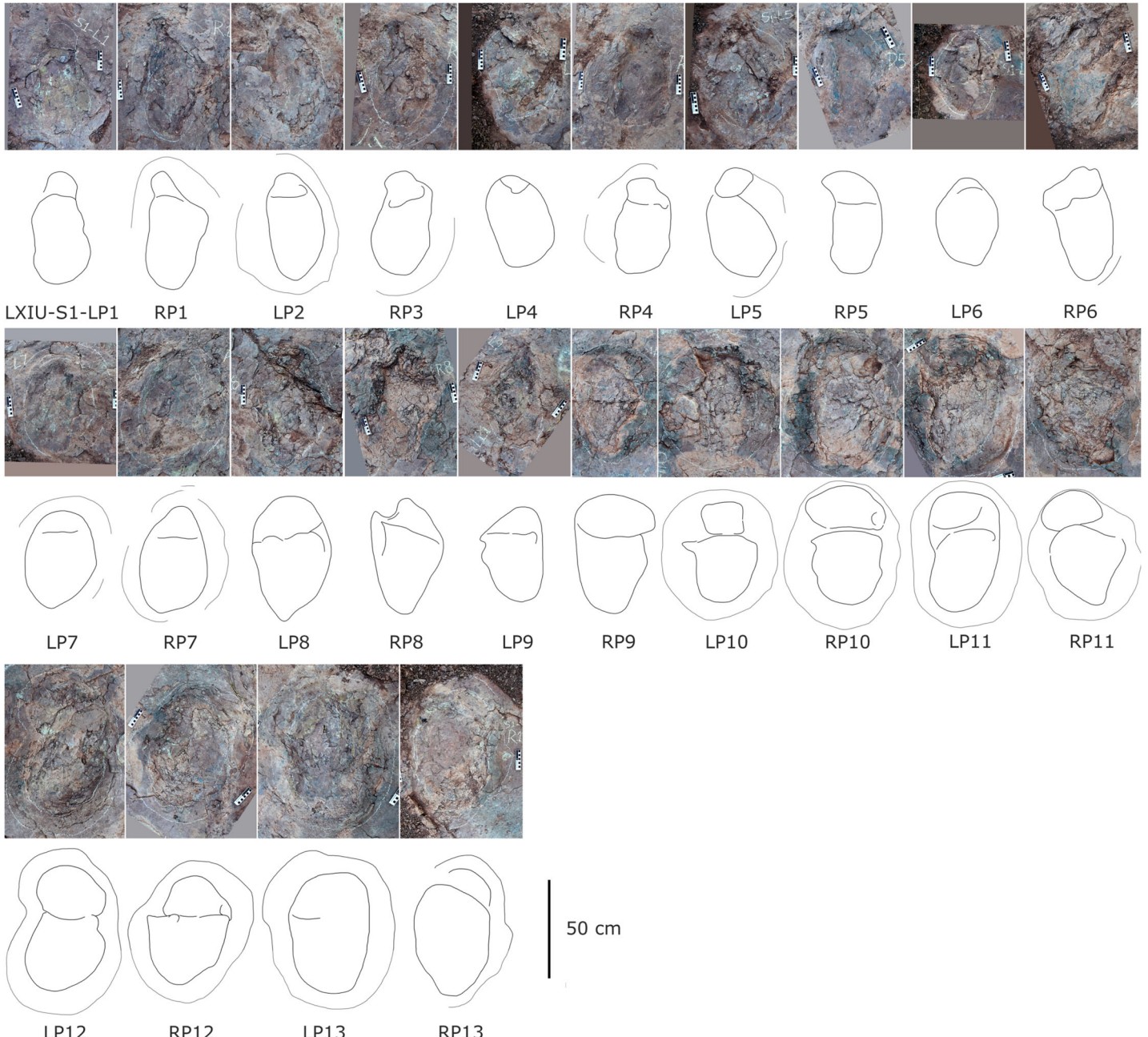

**Figure 9 Close-up photographs and interpretive line drawings of tracks from *cf. Brontopodus* trackway LXIU-S1.**

ratio of 1.2; the manus tracks averaged 16.6 cm in length and 25.9 cm in width, with an L/W ratio of 0.7.

Once the hindlimb of the trackmaker has left the substrate, the previously deformed sediment undergoes reflux to varying degrees, contingent on the dissimilarities in the nature of the local substrate and the forces exerted on the ground during travel. The relatively elongated pes track (*e.g.*, S1-LP2, RP13) may have originated from the closer

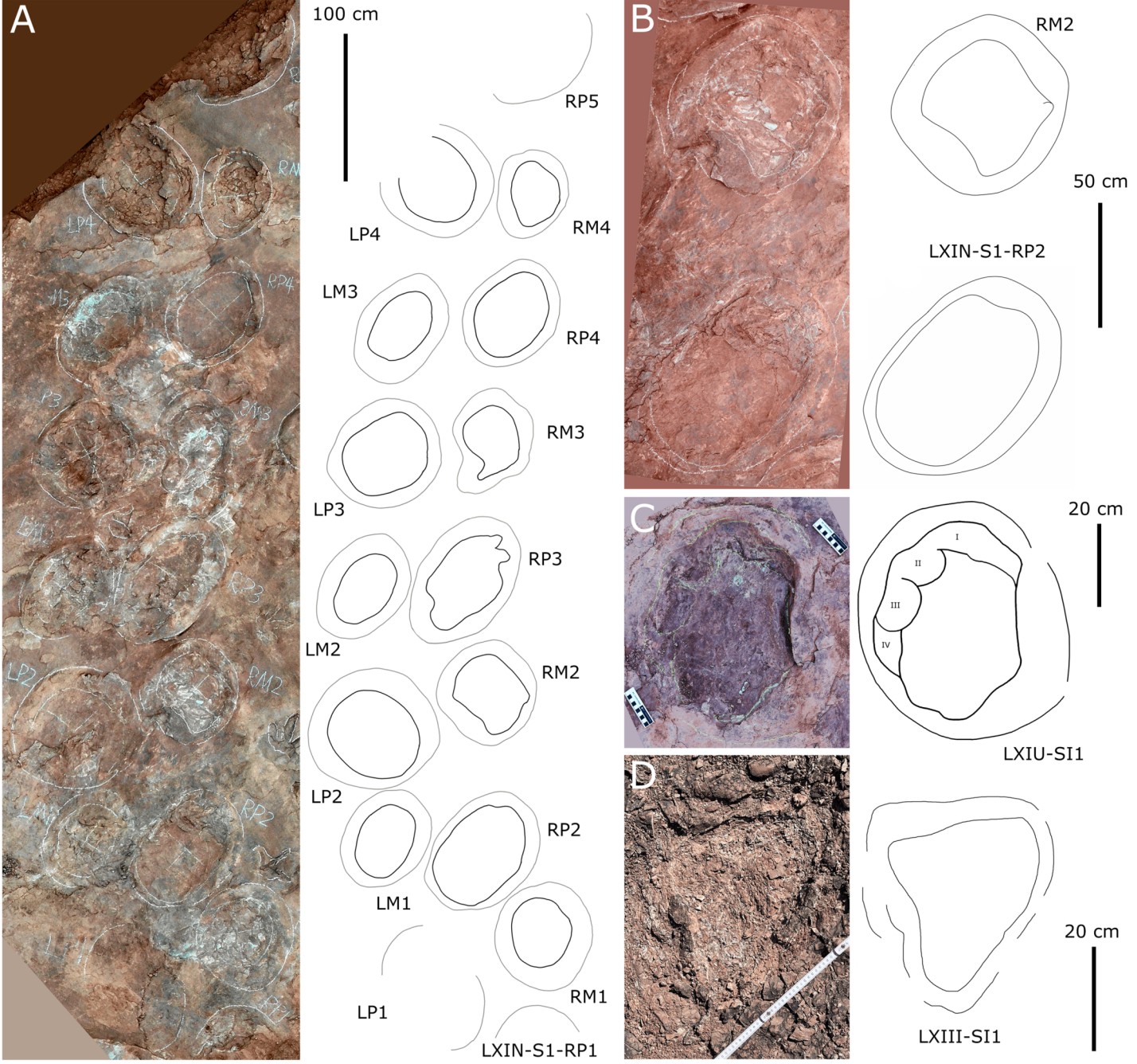

**Figure 10 Other photographs and interpretive line drawings of *cf. Brontopodus* from LXI tracksites.** (A) Trackway LXIN-S1; (B) Relatively well-preserved tracks in LXIN-S1; (C and D) Isolated sauropod tracks from LXIU and LXIII.

proximity of its centroid to that of the relevant manus, or from the deeper depth at which the trackmaker stepped into the sediment. In the adjacent deformations, the deformed sediment undergoes gradual recovery at varying velocities in response to both flow and pressure gradients. In instances where the ridge of sediment separating the two is not sufficiently stable, the lesser deformation will tend to be filled in first in order to reduce the

**Table 2 Measurements (in cm and degrees) comparison of *Brontopodus* morphotype trackways from Cretaceous Longxiang site I, Fujian, southeastern China.**

|  | L | W | R | L/W | PL | SL | PA | A | H | WAP | WAP/P'ML | TL* |
|---|---|---|---|---|---|---|---|---|---|---|---|---|
| LXIU-S1p | 40.78 | 33.22 | \|26\| | 1.25 | 91.24 | 161.99 | 127 | 1079.72 | 4.40 | 39.88 | 1.04 | 55.37 |
| LXIU-S1m | 16.62 | 25.91 | \|27\| | 0.70 | 99.81 | 162.58 | 111 | 358.88 | — | 56.59 |  |  |
| LXIN-S1p | 65.30 | 51.22 | 43 | 1.29 | 114.31 | 187.58 | 108 | 2578.26 | 1.82 | 67.58 | 0.97 | 81.20 |
| LXIN-S1m | 36.63 | 48.82 | 61 | 0.75 | 120.51 | 172.63 | 92 | 1452.94 | — | 81.56 | — | 59.74 |

Notes:
L, Track length; W, track width; R, rotation angle; L/W, length/width ratio; PL, pace length; SL, stride length; PA, pace angulation; A, area of the track; H, heteropody; WAP, width of angulation pattern (=Gauge width); WAP/P'ML, the ratio of WAP and P'ML; TL*, total length of pes and manus; —, no measurement available.
* All data are averaged within precise trackway, and Sxp and Sxm are the mean values of pes and manus respectively within the same trackway.

surface energy of the entire depression (*Cross & Hohenberg, 1993*; *Israelachvili, 2011*). In other words, the partially capped manus track undergoes further shrinkage and shallowing during the restoration of deformation. The degree of shrinkage may be related to the level of overlapping of the manus-pes tracks and the initial depth of the pes.

Almost all of the manus tracks of LXIU-S1 lie anterior to the pes, partially overlapped by the pes track but not very likely affected by the deformation rim. Taking the best-preserved examples, S1-LP8/RM8 and S1-RP10/RM10 (Fig. 10C), the manus prints are oval-shaped and the marks of the digit II–IV and metacarpophalangeal regions are indistinct. Digit I and V are observable, but without detailed morphology. The pes tracks are oval-shaped, and the digits I–IV are indistinct, with the smoothly curved metatarsophalangeal region. The rounded to rectangular-shaped heel pads are not well-preserved but rather distinctive in most pes tracks, especially in RP5, RP8, LP9 and RP13 (Fig. 9), and some of which are proximally elongated (*e.g.*, RP8) and related to form a V-shaped posterior rim. The manus and pes impressions are rotated approximately 27° and 26° outwards from the trackway axis. The mean pace angulation of the pes is 127°.

Similar to the variation in the footprint outline, poor preservation also results in a large variation in the size of the manus and pes tracks in trackway S1. The heteropody (ratio of manus to pes size) ranges from 1.6 to 13.3, with a mean of 4.4, a median of 3.0, and only 21% reaches 5. The range of width of S1 pes tracks reaches 62.2% of the width of the smallest (S1-RP5).

Additionally, an isolated left pes track, designated LXIU-SI1, is present on the northernmost side of the LXIU site (Fig. 10C). This sauropod track is located on the periphery of the LXIU site and exhibits superior preservation in comparison to trackway S1 at the same site (level 2 in *Belvedere & Farlow, 2016*). The LXIU-SI1 is slightly larger than the mean value of the pes track in LXIU-S1, measuring 50.1 cm in length and 38.0 cm in width. In comparison to the tracks in LXIU-S1, LXIU-SI1 did not retain the associated manus track. However, it did retain four more discernible digit traces (I–IV) and a possible rounded, faint digit V trace adjacent to the digit IV posteriorly. The widths of the digit II to digit IV are comparable, but the anteroposterior width of the digit I is comparable to that of the remaining three visible digits, that may link to a highly twisted position of digit I (including the pes ungual I) during travelling.

Site LXIN also contains a medium to large-sized sauropod trackway, LXIN-S1 (Fig. 10A). It contains 16 tracks with seven manus-pes sets and two separate pes tracks. The preservation status of the tracks is level 1 on the *Belvedere & Farlow (2016)* scale. The manus and pes tracks of LXIN-S1 were very loosely combined, with the pes tracks averaging 65.3 cm in length and 51.2 cm in width, and an L/W ratio of 1.3; the manus tracks averaging 36.6 cm in length and 48.8 cm in width, and an L/W ratio of 0.8. Almost all the manus tracks in site LXIN are subrounded or horseshoe-shaped, and anteromedial to the pes along the trackway orientation. In the best preserved examples S1-RP2 and RM2, the manus impressions are oval and the knots made by the digits I, II, IV and V are somewhat visible but not clearly defined (Fig. 10B). The metacarpophalangeal regions are distinct. The pes prints are oval and the digits I–IV are indistinct. Both the anterior rim of metatarsophalangeal and heel region is smoothly curved, in which the latter may suggest a kidney-shaped pad. RM3 has a well-developed digit V. RP3, corresponding to RM3, has distinct digits I and II. These features are not seen in the other manus impressions, probably because the substrate in this region are wetter and softer, leaving the tracks with more detail. The manus and pes impressions are rotated approximately 61° and 43° outwardly. The pace angulation of the pes is 108°.

Almost all LXIN-S1 tracks had distinct outer sediment rims. Including these rims, the length of the pes track is approximately 80.1 cm with an L/W ratio of 1.2, and the length of the manus track is 59.7 cm in length with an L/W ratio of 0.9. The area of the track, including the rims, can be up to 1.5 to 2 times the original area for pes and 2 to 2.5 times the original area for manus. The heteropody is 1.8, reduced to 1.5 if the rims are included.

Site LXIII preserves an isolated sauropod pes track LXIII-SI1 (Fig. 10D). SI1 is surrounded by very distinct sediment displacement rims. The pes prints are oval, with a length of 34.6 cm and an L/W ratio of 1.1. The marks of digits I–IV are indistinct and the metatarsophalangeal region is smoothly curved. Unlike other sauropod pes tracks in the Longxiang area, the heel of SI1 is quite narrow, with a width of about 1/2 of the digit region. In the absence of other related tracks, it is not possible to determine whether this character is a stable morphological feature or an ectomorphic variation.

## Comparison

Considering both footprint morphology and trackway configurations, both quadruped trackways LXIU-S1 and LXIN-S1are typical of sauropods affinity (*Lockley & Hunt, 1995*).

The diagnostic features of quadruped ichnotaxa usually include the footprint morphology (the number, size, shape and distribution of claw, digit and metatarophalangeal/heel pad), manus-pes set (the distance, relative size and position of manus and pes print), and trackway parameters (stride, footprint rotation and gauge). Track size (length and width) would be taken into account. However, given that sauropod track morphology is distinct from theropod and ornithopod tracks and does not show other variations that correlate with their body size, this parameter, while included in the diagnosis, is not necessarily considered when categorizing by morphology. For sauropod tracks are often only preserved as pits that lack sufficient footprint morphology, the heteropody (size discrepancy between manus and pes print/heteropody) and gauge are the

most commonly used as distinguishing features the diagnostic features of sauropod tracks, and the claw trace (number, orientation) are often taken into consider when preserved.

The preservation of both LXIU-S1 and LXIN-S1 are somewhat limited in the footprint morphology (Figs. 9, 10), especially the former with overlapped manus and pes that is challenging to obtain a independent heteropody. However, subject to the above conditions, the heteropody is relatively stable for both trackways: which of LXIU-S1 falling between 1:2 and 1:3. This is slightly lower than the 1:1.8 observed in LXIN-S1, and the overlap may causes higher ratio for the former. Both LXIU-S1 and LXIN-S1 tracks are characterised by distinct outward rotation.

The heteropody of both LXIU-S1and LXIN-S1 are lower than the Cretaceous *Brontopodus*, *B. birdi* (>1:3; *Farlow, 1989*) and *B. changlingensis*(~1:2.5; *Lockley et al., 2002*), and more similar to Jurassic *B. plagnensis* (1:1.86; *Mazin, Hantzpergue & Olivier, 2017*) with well-separated manus-pes sets. For other *Brontopodus*-type ichnotaxa with wide gauge and similar outline, heteropody is generally higher than 2, that holotype of *Gyeongsangsauropus* ("Brontopodus" *pentadactylus*) and *Parvieouspodus* ("Brontopodus" isp.) from Cretaceous East Asia reach ~1:2 and ~1:3.5 respetively (*Xing et al., 2024b*). The heteropody of previously named *Brontopodus*-type regional ichnotaxa, mid-Cretaceous "Sauropodichnus" from South America (*Calvo & Mazzetta, 2004*) and "Rotundichnus" from lower Cretaceous of Germany (*Hendricks, 1981*) are also estimated to be higher than 1:2.

The Jurassic *Parabrontopodus* defined in Laurasia (*P. mcintochi* and *P.* ("Elephantopoides") *barkhausensis*), with which *Brontopodus* is often compared in gauge width, has a generally smaller manus (>1:5 and 1:3.2; *Lockley, Farlow & Meyer, 1994*; *Meyer et al., 2021a*). It is clear from the 3D depth maps of *P. barkhausensis* (*Meyer et al., 2021a*) that at least some of the axial flattening of the manus that occurs in its type series is related to pes-induced substrate deformation.

For the most refered series of other highly regional sauropod ichnotaxa that vary in gauge, including Jurassic *Breviparopus* from Morroco (1:2.4; *Marty et al., 2010*), *Galinhapodus* ("Polyonyx") from Protugal (1:2; *Santos, Moratalla & Royo-Torres, 2009*; *Santos et al., 2024*); Jurassic-Cretaceous *Iniestapodus* from Spain ("Polyonyx" assosiated; ~1:2.5; *Torcida Fernández-Baldor et al., 2021*); Cretaceous *Oobardjidama* from Australia (1:2.9; *Salisbury et al., 2016*); *Teratopodus* and *Titanopodus* from different basins of Argentina (1:2.3 and 1:3.1; *Gonzalez Riga & Calvo, 2009*; *Tomaselli, David & González Riga, 2021*), their manus-pes sets are all loosely compacted and have a statistically stable heteropody between 2 and ~3. Of these, *Titanopodus* has a relatively high heteropody (*Gonzalez Riga & Calvo, 2009*), but this is not the only feature that distinguishes it from most sauropod ichnotaxa. As can be seen from the faint digit trace inside its manus print, its manus track and pes are very far apart, but still subject to compression from the posteromedial direction, resulting in a difference in the degree of deformation in the mediolateral direction in addition to the axial deformation.

Only Jurassic *Occitanopodus* from France has *P. mcintochi*-like higher heteropody that can be linked to the distinctly packed manus-pes set (1:4 to 1:6; *Moreau et al., 2019*), and Cretaceous *Calorckosauripus* from Bolivia that can be compared to *B. plagnensis* in slightly

lower heteropody and *B. changlingensis* in axially shortened pes track (1:1.85?; *Meyer, Marty & Belvedere, 2018*; *Meyer et al., 2021a*). *Titanosaurimanus* from Cretaceous of Croatia (*Dalla Vecchia, 2000*) and *P. frenki* defined in Chile (*Moreno & Benton, 2005*), on the other hand, lack sufficient records of manus-pes set.

Besides, for the ratio of the WAP/length of the pes (P'ML), a value of 1.0 separates narrow-gauge from medium-gauge trackways, whereas the value 1.2 is arbitrarily fixed to distinguish between medium-gauge and wide gauge trackways (*Marty, 2008*). Therefore, LXIU-S1 and LXIN-S1 can be catagorized as medium-gauge trackways for their WAP/P'ML are both ~1.0 (Table 2).

The outline of both LXIN-S1 fits the *Brontopodus*-type tracks, especially LXIN-S1 is similar to *Gyeongsangsauropus pentadactylus* occur in Cretaceous of East Asia (*Kim & Lockley, 2012*). And the loosely compacted LXIN-S1 manus-pes set can be compared to *B. plagnensis* and *Calorckosauripus* in heteropody and the absence of any claw trace, while the size, recognisable axial elongation (L/W > 1), and the degree of outward rotation of pes track is more comparable to morphotype E described in *Meyer et al. (2021b)*.

The LXIU-S1 with highly compacted manus-pes set and the more distinctive outward rotation may be more comparable to *Parabrontopodus*-type (*Lockley, Farlow & Meyer, 1994*; *Moreno & Benton, 2005*), while the relative width of manus (*i.e.*, potential size of manus) may wider than the former. However, as the compactness of manus-pes set is somewhat associated with the relative speed of trackmaker (*Lallensack & Falkingham, 2022*), we suggest to assign LXIN-S1 with more valid features to the *Brontopodus* -type trackway (or *cf. Gyeongsangsauropus* considering the track size and the similar Albian-Cenomanian age), and LXIU-S1 to an unclassified sauropod track or *cf. Brontopodus* comparing to the regional ichnofauna.

### The implication of sauropod ichnotaxonomy

It seems clear from the currently named (or formerly named), well-preserved sauropod ichnotaxa that, given the relative consistency of manus/pes track depth (Table 3), the heteropody in these records more readily reflects the original manus/pes size relationships. As a result, the valid intervals for this character should be confined to a very limited range. Given the extent to which heteropody can reflect differences in support function between manus and pes (*Strickson, 2020*), this phenomenon may indicate that in the later stages of sauropod divergence, in contrast to the differences brought about by gait (or speed) (*Lallensack et al., 2018*), differences in the role played by manus and pes in supporting body weight are likely to be relatively small in terms of sauropod morphology (including phylogeny, individual differences and ontogeny). This hypothesis needs to be further explored in a comprehensive comparative study of well-preserved material with low heteropody trackways found to date.

Also, the question of whether large angles or significant rotation of manus prints is a valid diagnostic feature needs to be treated with caution, as there are quite a few factors that influence this feature. *Lallensack et al. (2018)* analyzed sauropod trackways from the global record and found that strong lateral or postero-lateral rotation (supination) of the manus is restricted to trackways of small- and medium-sized individuals (pedal impression

**Table 3  The comparison of the track depth of the worldwide sauropod ichnogenera.**

| Region | Age | | Ichnotaxa | PL (cm) | PW (cm) | ML (cm) | MW (cm) | PD (cm) | MD (cm) | H | Manus/ Pes set | Ref |
|---|---|---|---|---|---|---|---|---|---|---|---|---|
| North America Texas, USA | K1 | Albian | *Brontopodus birdi* | 86.50 | ? | ? | 43.70 | ? | ? | >3 | Loose | *Farlow (1989)* |
| Colorado, USA | J3 | Kimmeridgian | *Parabrontopodus mcintochi* | 78.00 | 56.00 | 38.00 | 24.00 | ? | ? | (pronounced) | Packed | *Lockley, Farlow & Meyer (1994)* |
| East Asia South Korea | K1—2 | Albian—Cenomanian | *Gyeongsangsauropus* ("Brontopodus") | 50.39 | 44.33 | 32.46 | 41.36 | (Particularly shallow) | ? | ~2 | Very loose | *Kim & Lockley (2012)* |
| South Korea | | | *Brontopodus* (*cf. Gyeongsangsauropus*) | 39.54 | 36.43 | 25.18 | ~34.79 | (Particularly shallow) | ? | ~2 | Very loose | *Kim & Lockley (2012)* |
| Yunnan, China | K2 | Turonian—Coniacian | *Brontopodus changlingensis* | 38.30 | 32.00 | 14.67 | 26.67 | ? | ? | 2.5 | packed | *Lockley et al. (2002)* |
| Shandong, China | K1—2 | Albian—Cenomanian | *Parvicouspodus* holotype | 27.10 | 25.20 | 13.70 | 19.10 | 8~10# | 8~10# | ~3.5 | very loose | *Xing et al (2013)* |
| | | | *Parvicouspodus* paratype | 27.00 | 20.70 | 8.50 | 13.90 | ~4 | ~4 | ~3.5 | Very loose | *Xing et al (2013)* |
| Europe Switzerland | J3 | Kimmeridgian | *Brontopodus* | 30.00 | 23.90 | 9.60 | 19.10 | 4.3 | 3.3 | 1.8 | Packed | *Marty et al. (2010)* |
| | | | *Parabrontopodus* | 46.90 | 41.00 | 12.30 | 23.30 | 3.3 | 2.6 | 2.4 | Packed | *Marty et al. (2010)* |
| Germany | J3 | Kimmeridgian | *Parabrontopodus* ("Elephantopoides") | 35.5 | 25.00 | 12.00 | 24.00 | ~8?# | ~8?# | 3.2 | Packed | *Meyer et al. (2021a)* |
| | K1 | Berriasian | "Rotundichnus" | ? | ~60 | ? | ~40 | ? | ? | ? | Packed | *Hendricks (1981), Lockley, Wright & Thies (2004)* |
| France | J2 | Bathonian | *Occitanopodus* | 85.00 | 102.00 | 34.00 | 62.00 | ? | ? | 4~6 | Packed | *Moreau et al. (2019)* |
| | | | Sauropoda indet. | 77.00 | 103.00 | 27.00 | 65.00 | ? | ? | 3~4 | Packed | *Moreau et al. (2019)* |
| | | | Sauropoda indet. | 122.00 | 100.00 | — | — | ? | ? | — | — | *Moreau et al. (2019)* |
| | J3 | Tithonian | *Brontopodus plagnensis* | 97.80 | 74.90 | 48.40 | 71.00 | ~10 | ~10 | 1.9 | Loose | *Mazin, Hantzpergue & Olivier (2017)* |
| Portugal | J2 | Bajocian—Bathonian | *Galinhapodus* ("Polyonyx") | 90.00 | 60.00 | 38.00 | 58.00 | ~6? | ~6? | 2.0 | Loose | *Santos, Moratalla & Royo-Torres (2009), Santos et al. (2024)* |
| Spain | J3—K1 | Tithonian—Berriasian | *Iniestapodus* ("Polyonyx") assosiated | 63.00 | 43.00 | 28.00 | 31.00 | >11? | ~5? | 2.5 | Loose | *Torcida Fernández-Baldor et al. (2021)* |
| | | | *Iniestapodus* ("Polyonyx") assosiated | 60.00 | 43.00 | 29.00 | 40.00 | ~8? | >7? | 2.5 | Loose | *Torcida Fernández-Baldor et al. (2021)* |

(Continued)

# Table 3 (continued)

| Region | Age | Ichnotaxa | PL (cm) | PW (cm) | ML (cm) | MW (cm) | PD (cm) | MD (cm) | H | Manus/Pes set | Ref |
|---|---|---|---|---|---|---|---|---|---|---|---|
| | | Iniestapodus ("Polyonyx" assosiated) | 55.00 | 41.00 | 23.00 | 39.00 | ~10# | ~10# | 2.7 | Loose | Torcida Fernández-Baldor et al. (2021) |
| Africa | Morroco | J3 Oxfordian — Kimmeridgian | Breviparopus | 110.70 | 99.00 | 29.70 | 55.40 | 13.0 | 4.9 | 2.4 | Packed | Marty et al. (2010) |
| | | J3 Tithonian | Parabrontopodus frenki | 62.00 | 35.00 | — | — | ? | ? | — | — | Moreno & Benton (2005) |
| South America | Neuquen, Argentina | K2 Cenomanian | Brontopodus ("Sauropodichnus") | 70.00 | 60.00 | 25.00 | 40.00 | ? | ? | ? | Very loose | Calvo & Mazzetta (2004) |
| | Mendoza, Argentina | K2 Campanian | Teratopodus | 41.00 | 37.00 | 20.00 | 26.10 | ~5? | ~5? | 2.3 | Loose | Tomaselli, David & González Riga (2021) |
| | Mendoza, Argentina | K2 Campanian | Titanopodus | 46.00 | 42.40 | 19.60 | 32.20 | ? | ? | 3.1 | Very loose | Gonzalez Riga & Calvo (2009) |
| | Croatia | K1 Albian | Titanosaurimanus holotype | ? | ? | 20/24.5? | 19/21? | ? | ? | — | — | Dalla Vecchia (2000) |
| | | | Titanosaurimanus | 21.5/27? | 21.0/27? | ? | ? | ? | ? | ? | Loose? | Dalla Vecchia (2000) |
| | | | Titanosaurimanus | 29/30.0 | 26.5/27.5 | ? | ? | ? | ? | — | — | Dalla Vecchia (2000) |
| | Bolivia | K2 Maastrichtian | Calorckosauripus | [49] | [42] | [34] | [29] | 5.7#? | 8.8#?? | [1.85] | Loose | Meyer, Marty & Belvedere (2018), Meyer et al. (2021b) |
| Oceania | Western Australia | K1 Valanginian—Barremian | Oobardjidama | 75.30 | 61.00 | 41.00 | 48.00 | ~10# | ~10# | 2.9 | Very loose | Salisbury et al. (2016) |

**Notes:**
PL, pes length; PW, pes width; PD, pes depth; ML, manus length ;MW, manus width; MD, manus depth; H, Heteropody (=the ratio of the manus/pes track area);?, no recorded data (measurable); —, no measurement available.
#Maximum depth.
?Estimated from partial specimens of the trackway.
[]The average data of the morphotype from the presice site.

length <60 cm), this feature is also correlated with low speed and narrow gauge. *Lallensack et al. (2018)* also conclude that pronation occurs when the forelimb is actively contributing to the progression, at higher speed or when performing a wider gauge with the center of mass (COM) shifted anteriorly. There may also be relationships between the rotation of the pes and manus in trackways (*Xing et al., 2021a*).

Moreover, most sauropod trackways in China are wide- (or medium-) gauge and have been therefore, referred to the ichnogenus *Brontopodus* form lower Cretaceous of Colorado, USA (*Farlow, 1989*; *Lockley et al., 2002*), as the palaeofauna and ichnofauna are both considered to be comparable between East Asia and North America during Cretaceous (*Matsukawa, Lockley & Jianjun, 2006*; *Lockley et al., 2014a*), rather than a strict correspondence on dianogistic features from different perspectives.

Not only in Asia and North America, however, but also in other regions, a similar ichnotaxonomy focusing on the similarity of members of regional faunas (especially skeletal faunas) is used, rather than one dominated by track/trackway morphology. As in South America, one of the centres of sauropod evolution in the Cretaceous, the relative number of ichnotaxa there currently corresponds to the diversity of the sauropod skeletal record (*González Riga et al., 2015*; *Tomaselli, David & González Riga, 2021*; *Tomaselli et al., 2022*; *Pol et al., 2022*; *Carvalho & Leonardi, 2024*). It should be noted, however, that not only are there few sauropod records in the region that can be taxonomically classified (*Tomaselli, David & González Riga, 2021*), but also those specimens that are theoretically classifiable and were considered to have a distinguishable feature at the time of their discovery, including *Calorckosauripus*, *Titanosaurimanus*, *Titanopodus*, *Teratopodus* and 'Sauropodichnus', all of which are or were once considered valid holotypes of ichnogenera, with the range of comparisons essentially limited to specific basins based on potential maker differences or occurring only in specific tracksites.

This is not only due to contrast difficulties associated with differences in preservation (*Marchetti et al., 2019*), but is also related to the functional limitation of sauropod autopods to weight bearing (*Hutchinson, 2021*). In contrast, the current diagnostic features used in sauropod tracks are more similar to those of theropods and ornithischians, with an additional emphasis on gauge only in the trackway parameters (see the discussion above). The differences in foot function between the latter two and sauropods are more pronounced. Therefore, it seems appropriate for future studies to incorporate more 3D data and to include additional qualitative characteristics for plantar details other than digits that are statistically different between regions or between epochs within the same region. For example, heel morphology, which was mentioned in *Farlow (1989)* and has been the focus of some of the subsequent studies (*e.g.*, sauropod morphotypes in *Salisbury et al. (2016)*), or other features that may be related to weight bearing modality/COM.

### Speed estimation

For sauropods, *Alexander (1976)* first suggested that hip height be estimated as h = 4 × foot length, whereas, later, *Thulborn (1990)* estimated hip height as h = 5.9 × foot length. *González Riga (2011)* estimated hip height as h = 4.586 × foot length. Relative stride length (SL/h) may be used to determine whether an animal was walking (SL/h ≤ 2.0), trotting

(2 < SL/h < 2.9), or running (SL/h ≥ 2.9) (*Alexander, 1976*; *Thulborn, 1990*). Based on the formula of Thulborn and González Riga, the SL/h ratios of the LXIU-S1 and LXIN-S1 sauropod trackway are between 0.67–0.87, 0.49–0.63, and accordingly suggest walking. Using the equation to estimate speed from trackways (*Alexander, 1976*), the locomotion speed of the trackmaker of LXIU-S1 is between 2.27–3.02 km/h, LXIN-S1 is estimated as 1.66–2.23 km/h, and are consistent with most Chinese *Brontopodus*-type trackways (*Xing et al., 2016*; *Xing & Lockley, 2016*), for which speed estimates are always low.

It is notable that the relative stride length values of LXIU-S1 are highly consistent with those of BTY-S1 from the Tuchengzi Formation (across the J-K boundary) in western Liaoning (*Xing et al., 2021b*). In other words, the LXIU-S1 and LXIN-S1 specimens demonstrate that the outward rotation of the manus and pes tracks, as well as the degree of MPL, can exhibit notable differences in the gait during low-velocity movements.

### Trackmaker and the analogy of sauropod quadruped gaits

The gleno-acetabular distance (GAD) has been proposed as an independent trackway parameter for estimating the body size of the sauropod trackmaker (trunk length) by *Lallensack et al. (2018)*. This method is adapted for estimating trunk length when the trackmaker is in a stable gait (limb phase) (*Lallensack & Falkingham, 2022*). Ideally, GAD may not be influenced by the limb lengths in the direction of trackway orientation when the gait is known. Furthermore, the actual gait of sauropods has been re-evaluated in order to determine which pair of manus and pes tracks should be selected for GAD measurement (*Lallensack & Falkingham, 2022*; *Stevens, Ernst & Marty, 2022*).

Based on the above study, we selected only the limb phase of 25% and 50% as two plausible gait end members to measure the GAD of the sauropod trackway at sites LXIU and LXIN. Among them, LXIU-S1 has adjacent manus and pes tracks, with strong reduction of the manus track by deformation. Therefore, for LXIU-S1, we only use the more clearly-delimited segment from LXIU-S1-RP9/RM9 to LP13/LM13 in the southwest for estimation. Since the pes tracks of LXIU-S1 cover the correlative manus track, the measured value of this trackway is theoretically smaller than the actual value. It should also be noted that in LXIU-S1 there is a marked change in orientation between RP11 and LP11, so in this case we used two measurement methods for the 25% limb phase of LXIU-S1: (1) using RP11 as the boundary, dividing LXIU-S1 into two segments and considering the orientation in each part of the trackway as constant; (2) using the four adjacent manus-pes sets as a cluster to define the orientation of the posterior stride.

For LXIU-S1, the interval of its GAD was 1–1.63 m. When estimating the midline of the track using the different methods, the mean of the maximum estimates were all about 1.53 m (limb phase = 25%), with a variance of about 0.0080, and the mean of the minimum estimates were all about 1.09 m (limb phase = 50%), with a variance of about 0.0035. For the LXIN-S1 trackmaker, the GAD interval was 1.75–3.19 m. The mean of the maximum value was 3.14 m with a variance of about 0.0078–0.0080, and the mean of the minimum value was 1.83 m with a variance of about 0.0060. The greater variance in the LXIN-S1 data is due to: (1) apparent turning point; (2) increased stride length (RP12/LP13).

The wide/medium gauge of the *Brontopodus*/*Gyeongsangsauropus*-type trackways, especially the wide gauged trackways, are commonly attributed to titanosaurian sauropods by their compulsive abducent femur posture (*Wilson & Carrano, 1999*; *Lockley et al., 2002*; *Henderson, 2006*; *Mannion & Upchurch, 2010*). This posture is co-occurred with the ante-displacement with the COM (*Henderson, 2006*), and is hypothesised to be the consequence of adaptation to gigantism, as opposed to being merely phylogenetic-related in Macronaria (*Blazquez et al., 2024* (preprint)). Furthermore, the discovery of early medium-sized wide-gauge trackways in the Lower-Middle Jurassic can be attributed to the basal Eusauropoda (*Xing & Lockley, 2016*). However, it is also probable that these trackways could be the consequence of intense orientation-turning in their path (medium-gauge instead).

Both *Lallensack & Falkingham (2022)* and *Stevens, Ernst & Marty (2022)* demonstrate a tendency for sauropods to utilise a symmetrical, diagonal-supported walking gait. The diagonal-supporting gait is similar to the wide-gauge, classically hypothesised to be correlated with demand for stability maintaining of gigantic trackmakers (*Henderson, 2006*).

The hippopotamus is the most typical of the extant mammals that can use a torting gait (*Hildebrand, 1989*). They differ from other extant large-sized quadrupeds, and is the only known compulsory diagonal-supporting quadruped (*e.g.*, elephants and giraffes; see Fig. 1 in *Lallensack & Falkingham, 2022*). With the exception of hippopotamus, rather than graviportal elephants, which are more commonly analogised with sauropods, rhinoceros are also capable of utilising this gait at low-speed running (*Henderson, 2006*; *Hutchinson, 2021*; *Lallensack & Falkingham, 2022*). The body size, body mass and total track area of rhinoceroses are comparable to that of hippopotamuses (the track size estimation is from *Van den Heever et al., 2024*), yet the overall gait performance differs markedly (*Hutchinson, 2021*).

We hypothesise that this extreme obligate gait in hippopotamus terrestrial locomotion is likely to be related to its significantly shorter limbs relative to body mass (or possibly trunk length), in addition to the gravitational constraints it faces in common with other giant mammals (*Hutchinson, 2021*). First, the sub-ellipsoidal, elongated trunk is associated with the presence of its amphibius-adapted habit, with a specialisation logic approximating Sauropterygia (*Endo et al., 2019*; *Gutarra et al., 2023*). For hippopotamus and rhinoceros, which have similar other body parameters, the proportion of trunk length relative to the legs clearly distinguishes the two, and this particular proportion for hippopotamus is also rare in the full range of extant large-sized terrestrial quadrupeds (*Christiansen, 1999*). Second, given that modern giant mammals all use a narrow-gauge, the simultaneous usage of a diagonal-supporting gait during walking may limit the location of the pes drop point in the presence of a large limb length. To illustrate, narrow-gauged trackways for extant mammal trackmakers have been observed to avoid collisions between the front and hind limbs during walking. This has led to the hypothesis that terrestrial mammal trackmakers with longer hindlimbs are more likely to choose a lateral-supporting gait to ensure walking efficiency. However, the need to maintain trunk stability is not as pronounced in extant mammalian clades (*Vermeij, 2016*), as it is not as extreme as in the case of titanosaurian
sauropods, which have undergone a extreme process of gigantism (the phylogenetic relationship of paraceratheriids to extant rhinoceros see *Bai et al., 2020*).

It can be reasonably deduced that the underlying causes of the disparate trotting gaits observed in sauropods and certain other gigantic mammals are likely to be inconsistent. Sauropods exhibit a distinctive array of intra-clade body plan variations with regard to their mechanical locomotion, when compared to the extant gigantic mammals. For example, large mammals exhibit considerable variation in their relative trunk length (GAD-associated) in comparison to limb length (*Hutchinson, 2021*), as previously discussed. Conversely, the typical sauropod trackmakers display a notable discrepancy in the relative length of their forelimb and hindlimb along with the antedisplacement of COM (*e.g.*, see the reconstructions of macronarian brachiosaurus and diplodocoids Diplodocus by *Henderson, 2006*). It is similarly conceivable that these factors facilitated the advent of the hypothetical specialised gaits utilised by the makers of wide-gauged trackways *via* the disparate COM and hindlimb postural adaptations. However, this hypothesis requires further investigation in the future. However, since the estimation of GAD trunk size is independent of the functional reasons for the generation of torting gait, it does not affect the possibility of using the aforementioned approach to constrain the trackmaker's trunk size.

Besides, handful of questionable trackways made by large Paleocene-Eocene mammals shows similar wide-gauge as sauropods' (*e.g.*, *Henderson, 2015*; *Wroblewski & Gulas-Wroblewski, 2021*). *Wroblewski & Gulas-Wroblewski (2021)* argue that some wide-gauge mammal trackways are possible parallel pairs of narrow-gauge ones. However, the relative size of their tracks suggests that the body width of their makers was probably wider than the width of the inner part of their trackways, and also that if they are treated as parallel trackways, their gait angulation does not seem to match the non-cursorial morphology of their trackmaker's autopods. Therefore, we suggest that the wide-gauged trackways of early mammals should be further considered together with sauropods.

## Tridactyl tracks

The tridactyl tracks at Longxiang sites are distributed in the three main layers of LXI and LXIII (Figs. 5–9), and can be divided into three different forms. Given that the diagnostic feature of many isolated tracks are not sufficient for classification (*e.g.*, LXIN-TI17 to TI23), only the best preserved tracks were selected for discussion, and the undiscribed tracks can be found in the distribution map.

### *Grallatorid morphotype*

*Grallator* was originally defined in the Lower Jurassic, and used in a narrow sense for describing a type of small, slender tridactyl biped with possible early theropod affinities (*Olsen, Smith & McDonald, 1998*). It is also characterised by the markedly anterior extension of digit III, tightly packed digit II–IV, and the particularly low divarication angle of digit II–IV, which make it comparable to *Atreipus* in Triassic, which may be related to non-theropod dinosauromorphs (*Haubold & Klein, 2000*).

Due to the very well-preserved state of the *Grallator* holotype and the detailed comparisons and collations it received early on (*Olsen, Smith & McDonald, 1998*), it has been subsequently frequently cited by researchers from different regions of the world (*e.g.*, *Lockley et al., 2011*; *Xing et al., 2014b*, *2024b*; *Castanera, Piñuela & García-Ramos, 2016*; *Niedźwiedzki & Pieńkowski, 2016*; *Melchor et al., 2019*; *Klein et al., 2023*) and has defined a group of ichnotaxa produced by small theropods by comparison with *Grallator*. These include some of the regional ichnospecies commonly used under *Grallator*, as well as some of the grallatorid ichnogenera. For example, the former include *G. ssatoi* from the Cretaceous of East Asia (*Xing et al., 2024c*), while the latter include *Kalohipus* from the Upper Jurassic of Europe (*Castanera, Piñuela & García-Ramos, 2016*), *Picunichnus* and *Deferrariischnium* from the Lower Cretaceous of South America (*Calvo & Rivera, 2018*; *Melchor et al., 2019*), and *Jialingpus* from the Jurassic-Cretaceous of East Asia (*Xing et al., 2014b*). As a consequence, grallatorid is currently used in a broad sense to describe a class of small theropod track morphotypes with a rather large morphospace.

Relative to the originally defined *Grallator parallelus*, other grallatorids generally possess a wider size range (up to ~25 cm in *G. ssatoi* and *Picunichnus*), slightly higher divarcation of digit II–IV (up to ~60° when measuring the angle between the midline of digit), stronger digits or more developed digit pads, and more tightly compacted or fused heel region.

However, there are some examples of lower divarcation angles, such as the well-preserved referred specimens of *Picunichnus* with a typical maximum of <45° (*Melchor et al., 2019*), very close to the definition of <40° in *G. parallelus*. The distinguishable feature of most typical grallatorid in Jurassic-Cretacaous of East Asia (*Jialingpus* and *G. ssatoi*) from other grallatorid likely for statistically lower mesxomy and the developed, rounded metatarsophalangeal pad IV in the well-preserved *Jialingpus* specimen.

Within regions, the use of grallatorid ichnotaxa is often influenced by the morphological continuity of tracks within the same region, as well as the taxonomic perspective of historical studies. As a result, the choice of whether to establish new ichnogenera or new ichnospecies varies considerably. For example, the morphospace of *G. ssatoi*, which is widely distributed in the Lower Cretaceous of China, may be sufficient to establish an ichnogenera by simultaneous comparison with *Jialingpus* and *G. parallelus* under criteria from other regions (*Xing et al., 2024c*). However, preliminary size statistics (*Lockley & Xing, 2021*), together with morphological comparisons, indicate that it is difficult to find a stable morphological boundary between the tracks attributed to this morphotype. On this basis, a significant number of tracks are limited by preservation and lack track support to discuss their mode of deformation, hence the aforementioned peculiarities of the grallatorid ichnotaxa in this region. Adjustment for interregional differences in these ichnotaxa will require careful integration of detailed differences in representative tracks and statistical differences in interregional tracks over the same period. As this is not the focus of this article, we suggest further discussion in conjunction with data from large tracksites where the data are statistically significant.

In the Longxiang tracksite, none of the trackways resemble the grallatorid morphotype, although a few isolated tracks and tracks within certain trackways show similarities to it. Among the isolated tracks, only LXIU-TI7, LXIN-TI3 and TI4 are similar to grallatorid morphotype in the extended digit III and the associated L/W ratio and mesxomy (Fig. 11; measurements see the following tables). However, the spindle-shaped digit III and the linear digit II and IV indicate that this footprint has undergone significant distortion, making it uncertain whether its original characteristics truly resembled the grallatorid morphotype. For LXIN-TI3 and TI4, although their preservation is slightly better than that of LXIU-TI7, the lack of well-defined digit traces and the presence of sediment deformation structures indicate that neither represents its original morphology. Additionally, their size (~35 cm in length) is significantly larger than the general range of grallatorids.

Furthermore, the LXIN site also contains LXIN-T5-R1, which is part of a robust theropod trackway. This footprint resembles grallatorids in its high length-to-width (L/W) ratio (>1.5) and high mesaxony (~0.66), but its digits II and IV are distinctly medially directed. This suggests that at the Longxiang site, footprints resembling the grallatorid morphotype may have resulted from a robust trackmaker adopting a specific behavior of drawing digits II and IV inward while moving on a soft substrate, combined with substrate deformation, rather than representing a distinct trackmaker.

### The comments on the ichnotaxonomy of large theropod track

The large-sized tridactyl tracks (>30 or >25 cm in length) are widely distributed across the Jurassic and Cretaceous strata (Foster et al., 2024; Xing et al., 2024b). Excluding the distinguishable *Caririchnium* and *Hadrosauropodus* exhibiting derived ankylopollexian affinities, which characterized by large, rounded metatarsophalangeal pad and short, snout-like digits (Diaz-Martinez et al., 2015), and the typical theropod tracks with slenderer, asymmetrically developed digits and sharper claws, a considerable number of intermediate ornithopod and theropod ichnotaxa with highly comparable morphology are also found (Lallensack, van Heteren & Wings, 2016; Lallensack, Engler & Barthel, 2020), for example, *Iguanodontipus* with smaller, less separated heel pad than *Caririchnium*.

Considering the morphology of known theropod ichnotaxa, generally three roughly defined morphotypes could be used to describe the large tracks with possible theropod affinity. These morphotypes mainly differ in the shape and relative position of the trace of digits II–IV and the metatarsophalangeal pad IV, including:

① The elongated *Eubrontes* morphotype is characterized by a higher length/width ratio, generally evenly developed digit II and IV, typically exhibit a lower divarication angle (<40°). The definition here differs from that of eubrontid, which is used to describe an overly large morphospace of Eubrontidae, which simultaneously includes the widely used *Kayentapus*, *Megalosauripus* and *Eubrontes*. The most typical member of this morphotype is the *Eubrontes* specimens that most closely match the currently recognized diagnostic features (Olsen, Smith & McDonald, 1998), such as type ichnospecies *E. giganteus* and the exceptionally well-preserved *E. nobitai* (Fig. 6 in Xing et al., 2021c).

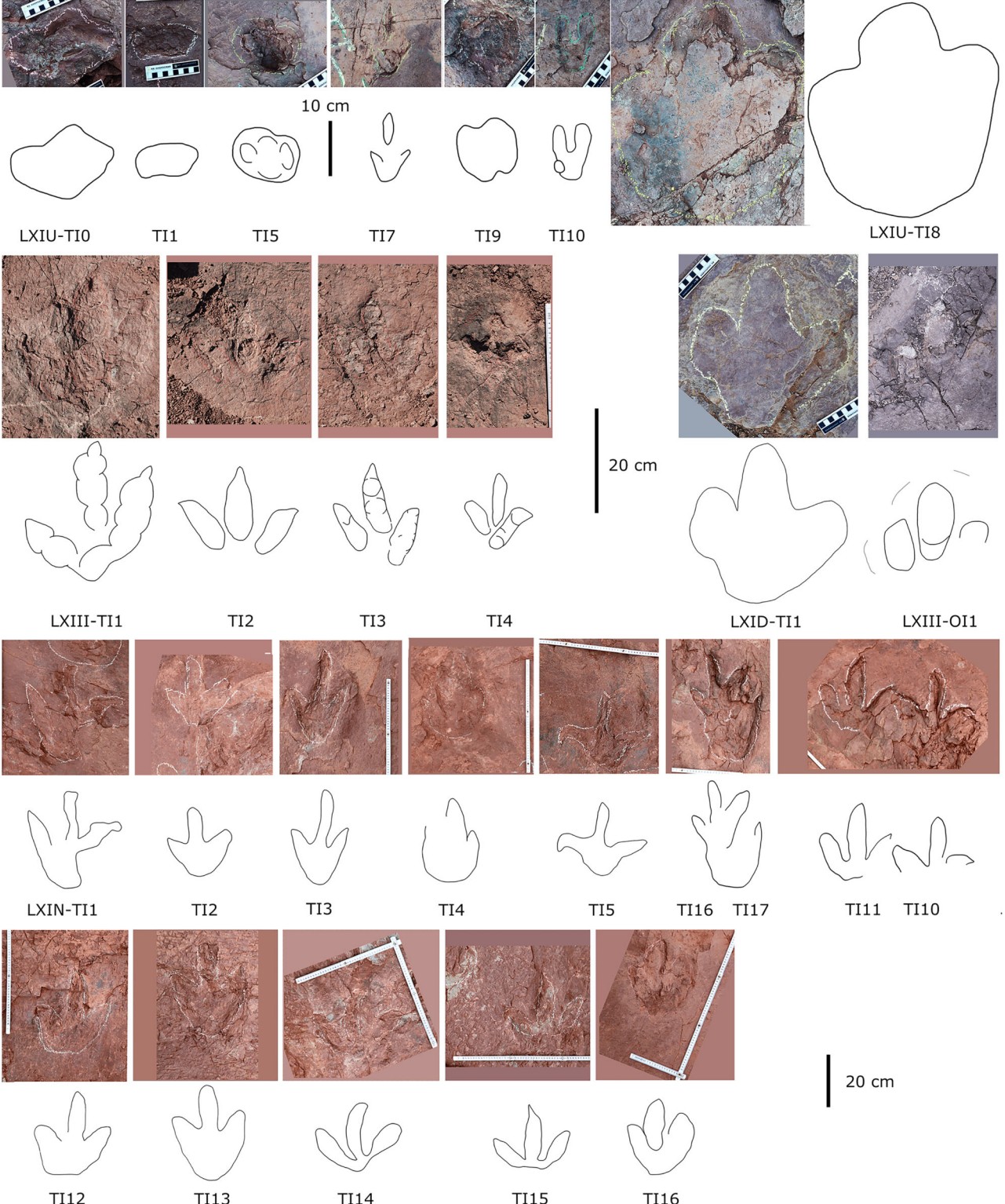

**Figure 11 Photographs and interpretive line drawings of isolated theropod tracks from LXIs and LXIII.** The tracks posited in the same row or from the same tracksite share the similar scale bar.

Other specimens can be broadly assigned to this morphotype based on the similar development of digits II and IV (*i.e.*, exhibiting a more comparable range of motion) and a more pronounced anterior symmetry, while possess a higher divarication angle. These members with mainly include *Asianopodus*-like tracks (*Matsukawa, Lockley & Jianjun, 2006*; *Xing et al., 2021d*; or *Jialingpus*-like), which is characterized by the presence of a significantly developed bulbous metatarsophalangeal IV that is essentially positioned along the central axis of the footprint, and they are statistically more symmetric than the following mentioned *Megalosauripus* morphotype. These tracks are most typical and well-preserved in the Cretaceous of East Asia and include the representative *Asianopodus* and the comparable *Chapus* (*Lockley et al., 2018*; *Xing et al., 2021d*), as well as *Changpeipus* from the Jurassic (*Xing et al., 2014c*; *Klein et al., 2023*) and the larger *Jialingpus* (*Xing et al., 2014b*), which lacks a definitive diagnosis based on size.

Some Cretaceous specimens previously classified as *Irenesauripus* from North America (*Lockley et al., 2014b*) also loosely resemble *Eubrontes* morphotype. Gondwana morphotypes appear to be relatively lacking in this category (*Salisbury et al., 2016*; *Melchor et al., 2019*; *Leonardi & Carvalho, 2021*). A previously identified *Irenesauripus* from Argentina is closer to small tracks resembling *T. pandemicus* (Fig. 1 in *Melchor et al., 2019*).

Among the *Asianopodus*-like tracks, most non-Asian tracks, in addition to Asian *Changpeipus*, show a slightly lower mesaxony compared to *Asianopodus* and *Jialingpus*, with digit III and metatarsophalangeal IV being more tightly connected. Sometimes, digits II or IV exhibit a larger range of motion (RoM). For example, in Moroccan *Changpeipus* (*Klein et al., 2023*), the notable asymmetry between digits II and IV, is more similar to *Megalosauripus*. Furthermore, except the indistictive metatarsophalangeal region, non-*Changpeipus* morphotypes (*e.g.*, the aforementioned *Irenesauripus*) often display less posteriorly compacted digits, with digits II and IV being relatively parallel. These tracks are more likely to be associated with ornithischian *Dinehichnus* of similar size (*e.g.*, *Castanera et al., 2020*), with some specimens possibly belonging to *Iguanodontipus*? ("*Therangospodus*") *oncalensis* (Fig. 6 and 11 in *Castanera et al., 2013*). However, ornithischian ichnogenera tend to have more robust digits II–IV with indistinct phalangeal pads, and possibly co-occur with manus tracks (*Lallensack, Engler & Barthel, 2020*). Similar digit III connections have been found in smaller emu and possible ornithomimid ichnogenera (*Ornithomimipus* and *Irenichnites*), as well as large bird-like tracks (*Magnoavipes* and *Saurexallopus*). Comparative studies of the Cretaceous theropod fauna in Laurasia (*e.g.*, *Novas et al., 2013*) reveal that the East Asian Cretaceous paleoenvironment is largely restricted to terrestrial deposits (*e.g.*, *Zhou et al., 2018*), potentially suggesting that habitat, rather than phylogeny, strongly influences the foot morphology of possible theropod trackmakers in this morphotype.

The scarcity of *Eubrontes* morphotype, especially *Asianopodus*-like tracks in Gondwana may be linked to the relatively low presence of derived coelurosaurs in that region. The unique arctometatarsal structure in coelurosaurs (*Holtz, 1995*; *Tanaka et al., 2021*) and other body plans may have contributed to their bird-like compacted heel (*Xing et al., 2014b*), and backward-shifted centers of mass (*e.g.*, *Farlow et al., 2000*), potentially limiting

the distribution of *Asianopodus* (or *Jialingpus*)-like morphotypes. Validation of this hypothesis remains beyond the scope of this study and will require further investigation, including more specimens and biomechanical reconstructions in future research.

② The elongated *Megalosauripus* morphotype with moderate length/width ratio, typically have more flexible and well-developed digit II that results in a separated digit trace II with a more sharply defined outline, an asymmetrical divarication angle between II–III and III–IV, and a tendency to exhibit an unstable anterior angle within the trackway.

This morphotype is represented by the worldwide-distributed *Therangospodus* (*T. pandemicus*)—*Megalosauripus* (*M. transjuranicus* and *M. uzbekistanicus*) (*Gierlinski, Niedzwiedzki & Pienkowski, 2001*; *Barco, Canudo & Ruiz-Omenaca, 2006*; theropod morphotype C in *Salisbury et al., 2016*; *Razzolini et al., 2017*; *Belvedere et al., 2019*; *Xing et al., 2024b*). Some other moderate-preserved ichnogenera, especially many of which defined in Europe, are similar but usually regionally compared, include *Boutakioutichnium* (*Nouri, Díaz-Martínez & Pérez-Lorente, 2011*), *Bueckerburgichnus* (*Cobos et al., 2014*), "*Hispanosauropus*" (*Lockley et al., 2007*), *Eutynichnium* (*Belvedere et al., 2019*) from upper Jurassic and lower Cretaceous of Europe. The Asian ichnogenus *Gigandipus*, on the other hand, is a relatively small (~30 cm in length) and slender representative among the possible members of this group (*Xing et al., 2018a*).

In contrast to other large ichnogenera, the similar but narrower tracks of this type are more commonly found in small theropod tracks, although in the former, the metatarsophalangeal pad is notably more developed with the increasing body size.

It is important to note that some other gigantic theropod ichnogenera also exhibit similar asymmetry, such as *Tyrannosauripus* and *Bellatoripes*, which are suggested to have a tyrannosaurid affinity (Fig. 7.31 in *Xing et al., 2024b*). However, these tracks not only have more developed heel regions, but also display shorter digits and a slightly more parallel alignment compared to typical *Megalosauripus*. Large theropod tracks (~30 cm in width) with similar features are also found in the Jurassic of Europe, including *Hispanosauropus* and *Eutynichnium* (*Foster et al., 2024*), with *Eutynichnium* being tetradactyl. The differing development and orientation of digit I suggest significant variation in trackmaker.

According to the aforementioned classification, some morphotypes exhibit intermediate characteristics between the *Eubrontes* and *Megalosauripus* types. These are primarily relatively symmetrical forms with a higher divarication angle and a relatively isolated small heel pad, though the connection between the latter and the digits is not particularly pronounced. The posterior margin of such morphotypes commonly exhibits a V-shape, and they appear to be generally more slender, tending to occur within a smaller size range (30–40 cm in length).

One example is *Kayentapus*, which has been widely used but is primarily defined in the Jurassic and is characterized by a less distinct heel region (*Lockley, Gierlinski & Lucas, 2011*). Among the more slender forms, some specimens, such as those illustrated by *Xing et al. (2021e)*, show a relatively symmetrical morphology and a larger divarication angle, closer to that of bird tracks. In contrast, more robust forms, such as the type specimen of *K. hopii* (*Lockley, Gierlinski & Lucas, 2011*) and the Asturias specimen (*Avanzini, Piñuela*

*& García-Ramos, 2012*), tend to be more asymmetrical, with the latter exhibiting a more separated digit II.

The previously mentioned *Irenesauripus* generally exhibits thinner digits II and IV, which are relatively consistent in length and divarication angle compared to other large theropod ichnogenera from the same region. In contrast to the *Megalosauripus* type—where digit IV is more frequently connected to the heel—digit II in *Irenesauripus* is more isolated (*McCrea, 2000*; *Lockley et al., 2014b*). By comparison, *Megalosauropus*, which has been redefined exclusively from Australia, more closely resembles the aforementioned *Kayentapus* in both morphology and its relatively smaller size range (*Salisbury et al., 2016*).

Furthermore, it should be noted that the above-mentioned *Eubrontes* and *Megalosauripus* are not well-separated morphotypes by definition. In fact, the distinguishing features in their diagnoses are quite different. The primary difference between them lies in the *Grallator* (*-Anchisauripus*)*-Eubrontes* plexus and the *Megalosauripus-Therangospodus* plexus, and the differences between the two are not purely morphological, but also involve issues such as temporal limits (*Lockley, Meyer & dos Santos, 1998*; *Lockley, Meyer & Moratalla, 1998*). Therefore, it is important to emphasize again that the two morphotypes discussed above are based on distinctive commonalities observed through the re-examination of type specimens and other well-preserved specimens, rather than simply a morphological comparison at the ichnotaxon level.

③ The stout *Jurabrontes* morphotype with ornithopod track-like (*Dinehichnus* or *Iguanodontipus*) outline that possess short digits, generally low length/width ratio (~1) and mesaxony (~0.5), but with distinctive phalangeal pads, fainter, smaller but often separated metatarsophalangeal pad IV, as well as sharp claw traces. The majority of giant morphotypes (generally >50 cm in length), although differing in the position of digits II and IV and the occurrence of digit I, can largely be classified within this morphotype based on the evenly developed, robust digits II and IV, as well as the higher heel pad length/footprint length ratio (indicating a shorter digit III).

This morphotype is well-represented by specimens such as *Jurabrontes* from the Upper Jurassic of Europe (*Marty et al., 2018*; *Belvedere et al., 2019*; *Antonelli et al., 2023*). These tracks show some asymmetry, but to a lesser extent than *M. transjuranicus* (*Belvedere et al., 2019*).

Giant ichnotaxa with similar weight-bearing adaptations include the *Tyrannosauripus* and *Bellatoripes* from the Cretaceous of North America and East Asia, and *Bueckeburgichnus* and "Hispanosauropus" from lower Cretaceous and upper Jurassic of Europe respectively, while all of these morphotypes possess a digit I trace with different length, and develop the somewhat develop *Megalosauripus*-type lateral digits. Some referred specimens of *Iberosauripus* from Upper Jurassic to Cretaceous of Europe represents rather more *Eubrontes* type-like, with unfused heel region, relatively elongated and paralleled digits (*Castanera et al., 2015*), while the part of the type series represent indistinct heel region that somewhat similar to *Jurabrontes* (*Belvedere et al., 2019*). However, the latter still differ with the moderate-preserved *J. teutonicus* also without the occurrence of heel pad in the compact arrangement of digits, and there is no tendency for

fusion between the metatarsophalangeal pads or a significant enlargement of one of them (Fig. 5 in *Belvedere et al., 2019*).

*Moraesichnium* and *Salfitichnus* from the Cretaceous of South America, comparable to *Jurabrontes* morphotype in track size, have been subject to criticism regarding their original outline interpretations (*Masrour et al., 2017*; *Xing et al., 2024a*), making morphological comparisons with morphotype 3 and the aforementioned two types challenging. Notably, the interpretation of *Moraesichnium* seems to differ significantly from the photograph (Fig. 4.71 in *Leonardi & Carvalho, 2021*). In particular, digit IV in SOPP 2/5 appears to have sunk deeply into the sediment, exhibiting a trace of sediment collision (*e.g.*, Fig. 3 in *Campos-Soto et al., 2025*). This deformation resembles that seen in Track 1321 from the Piau-Caiçara tracksite (Fig. 4.38 in *Leonardi & Carvalho, 2021*). We recommend further imaging or 3D depth mapping of the specimens associated with *Moraesichnium* before revisiting its morphological interpretation. Compared to these problematic ichnotaxa mentioned by *Melchor et al. (2019)* and *Leonardi & Carvalho (2021)*, *Jurabrontes*-like tracks are fit more to describe the South American indeterminate bipedal dinosaur footprint fall within this size range.

The other large theropod ichnogenera defined in South America, the mid-Cretaceous *Abelichnus*, is slightly smaller in size (40–50 cm in length, possible present smaller in size due to the moderate preservation), but could be compared with the small *Jurabrontes*, *J. melphicticus* in the general outline than the giant *J. curtedulensis* considering the less robust, V-shaped heel. Other large ichnogenera mainly occupy similar size ranges as *Abelichnus* usually have slenderer, longer digit III proportionally than the former, including the European *Boutakioutichnium* and the Asian *Chapus* and *Ordexallopus*. However, compared to the Asian morphotypes, *Boutakioutichnium*, despite preserving the trace of digit I, more closely resembles *Abelichnus*.

In Australia, large theropod tracks classified as the Broome Theropod Morphotype B resemble *Jurabrontes*-like tracks (*Salisbury et al., 2016*). However, due to average preservation, it is unclear whether they are more similar to Broome Theropod Morphotype C, which exhibits a greater degree of asymmetry that more closely resembling the European *Megalosauripus* type.

Generally, large theropod tracks that resemble *Asianopodus*/large *Jialingpus*-like morphotypes may be more directly associated with specific theropod clades among the theropod morphotypes. This hypothesis requires further validation through future studies, including posture reconstructions of coelurosaurian and non-coelurosaurian trackmakers. Additionally, the distinct ceratosaurian and megaraptoran-dominated theropod fauna of Cretaceous South America (*Novas et al., 2013*; *Lamanna et al., 2020*) urgently requires comparative analysis with the Jurassic records of Laurasia, considering possible deformations and subsequent ichnotaxon revisions.

### Theropod morphotype A

**Description.** The majority of theropod tracks and trackways in Longxinag site belongs to Theropod morphotype A, including five trackways (LXIN-T1–T5; Figs. 12, 13) and at least

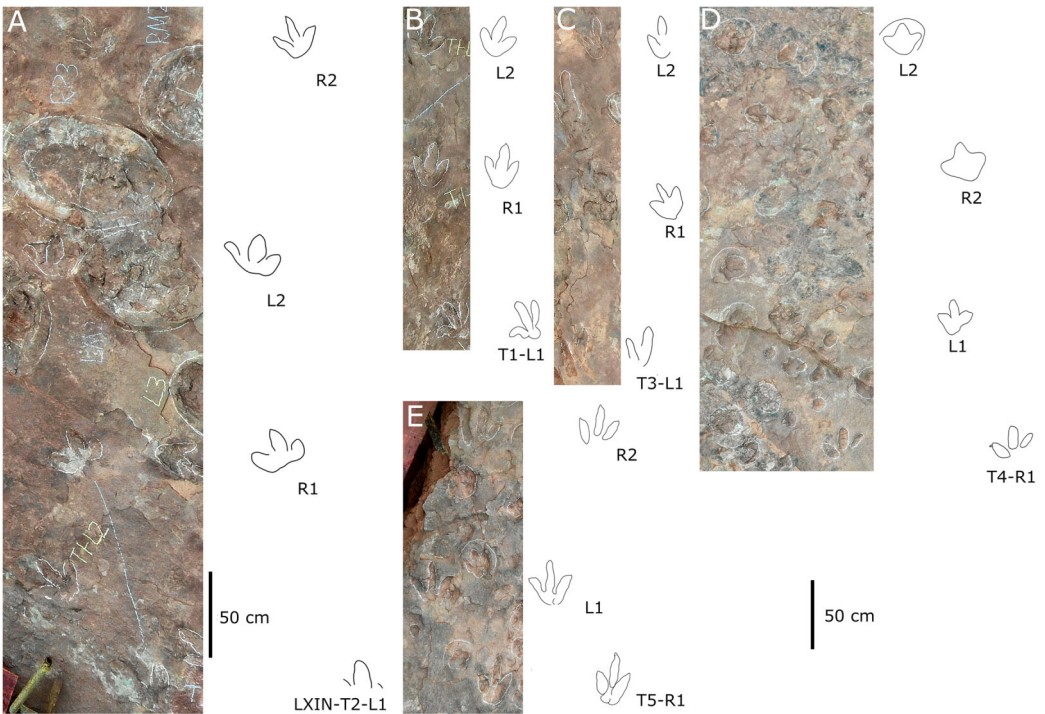

**Figure 12 Photographs and interpretive line drawings of theropod morphotype A trackways from LXIN.** LXIN-T2 (A, left) and the other trackways (B–D, right) share the similar scale bar separately.                                                                                     

12 isolated footprints (LXIN-TI1–6, TI10–16) are from the LXIN site (Fig. 11), one trackway (LXID-T2; Fig. 13) and one isolated track (LXID-TI1) from LXID site (Fig. 11), one isolated track from LXIU site (LXIU-TI7) and four isolated tracks in LXIII (LXIII-TI1–4; Fig. 11) (Tables 4, 5).

Five trackways from the LXIN site including LXIN-T1-T5 (Fig. 12E; Table 4). These tracks vary in length due to the erratic occurrence of heel region, however, they exhibit a relatively stable width, concentrated within the range of 25–35 cm. Only some of the isolated small tracks (<~20 cm in length) with very similar outlines to the large track from the same site, including LXIN-TI6, LXIU-TI7, LXIII-TI2 to TI4, are also classified in this morphotype,that may represent some small individuals. All the trackways are short but distinguishable, and contains three to eight continuous footprints. In most of the trackways, the footprints are slightly inward-rotated overall.

Among these trackways, the three well-preserved tracks in LXIN-T5 (R1, L1, R2) essentially represent the two main deformation patterns within this morphotype (Fig. 13). This further supports that these morphologically diverse yet size-consistent footprints should be classified under the same morphotype. In the Theropod morphotype A, the majority of trackways (LXIN-T1, T2, T4, T5) and a considerable number of isolated tracks (LXIN-TI5?, TI10–11, TI12?, TI14–15; LXIII-TI2–3) exhibit a digitigrade form similar to T5-L1 or R2, preserving only a partial metatarsophalangeal pad.

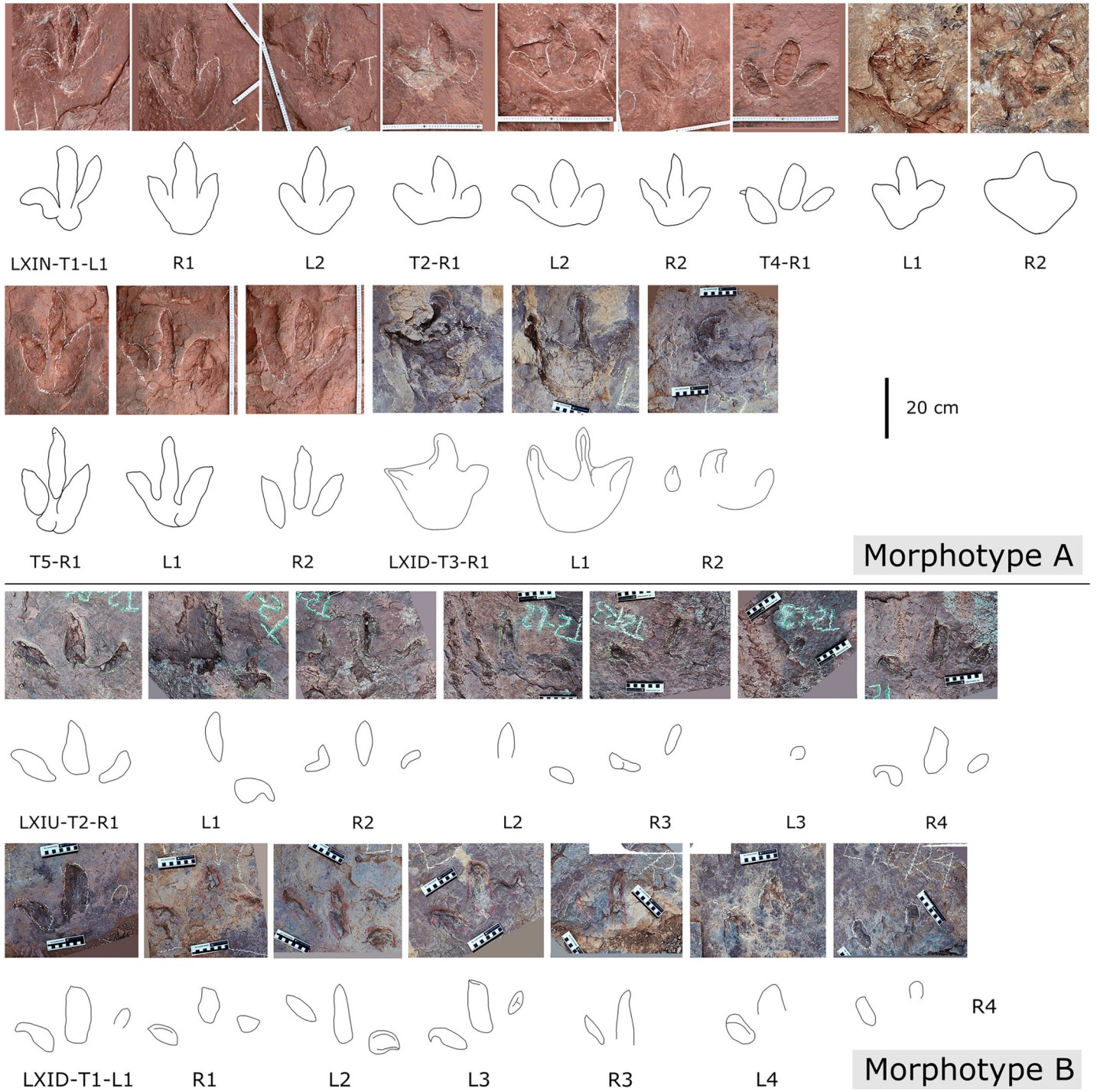

**Figure 13 Close-up photographs and interpretive line drawings of theropod morphotype A & B tracks from trackways in LXI sites.** All the tracks share the similar scale bar.

The preservation state of LXIN-T5 corresponds to level 1–2 on the scale of *Belvedere & Farlow (2016)* (Fig. 12E). The digit imprints are stable in width, possess quite clear and sharp outline, and claw marks and some phalangeal pads can be faintly identified.

Table 4 Measurements (in cm and degrees) comparison of theropod morphotype A and B trackways from Longxiang sites. All data are averaged within precise trackway.

| | L | W | L/W | II–III | III–IV | II–IV | PL | SL | PA | R | WAP |
|---|---|---|---|---|---|---|---|---|---|---|---|
| **Theropod morphotype A** | | | | | | | | | | | |
| LXIN-T1 | 31.48 | 27.03 | 1.19 | 40 | 31 | 70 | 103.43 | 206.70 | 176 | 27 | 0.46 |
| LXIN-T5 | 30.71 | 24.82 | 1.23 | 37 | 29 | 66 | 108.37 | 216.70 | 177 | 27 | 0.44 |
| LXIN-T3 | 29.95 | 22.60 | 1.27 | 34 | 28 | 62 | 113.30 | 226.70 | 177 | \|6\| | 0.41 |
| LXIN-T2 | 23.67 | 28.27 | 0.85 | 42 | 49 | 91 | 128.53 | 247.85 | 166 | \|15\| | 0.35 |
| LXIN-T4 | 24.30 | 28.66 | 0.85 | 43 | 42 | 85 | 104.08 | 207.39 | 152 | \|11\| | 0.37 |
| LXID-T3 | 23.55 | 27.20 | 0.86 | 37 | 44 | 80 | 96.49 | 192.43 | 171 | 36 | 0.26 |
| LXIU-T2 | 14.73 | 29.35 | 0.50 | 62 | 45 | 107 | 99.07 | 179.79 | 133 | \|9\| | 0.30 |
| LXID-T1 | 14.01 | 30.68 | 0.46 | 63 | 46 | 109 | 144.69 | 279.63 | 156 | \|28\| | 0.29 |
| **Theropod morphotype B** | | | | | | | | | | | |
| LXIU-T2 | 14.73 | 29.35 | 0.50 | 62 | 45 | 107 | 99.07 | 179.79 | 133 | \|9\| | 0.30 |
| LXID-T1 | 14.01 | 30.68 | 0.46 | 63 | 46 | 109 | 144.69 | 279.63 | 156 | \|28\| | 0.29 |

Note:
L, Track length; W, Track width; L/W, Length/width ratio; II–III, III–IV and II–IV, the divarication angle between digit II–III, III–IV and II–IV; PL, pace length; SL, stride length; PA, pace angulation; R, rotation angle; M, mesaxony (=AT, anterior triangle length/width ratio); —, no measurement available.

Table 5 Measurements (in cm and degrees) comparison of the isolated theropod tracks from Longxiang sites.

| | L | W | L/W | II–III | III–IV | II–IV | M |
|---|---|---|---|---|---|---|---|
| **Tridactyl** | | | | | | | |
| LXIU-TI7 | 12.9 | 7.2 | 1.8 | 31 | 35 | 66 | 0.99 |
| LXIN-TI1 | 37.9 | 36.9 | 1.0 | 29 | 37 | 66 | 0.27 |
| LXIN-TI2 | 26.4 | 21.7 | 1.2 | 34 | 33 | 67 | 0.45 |
| LXIN-TI3 | 36.8 | 23.0 | 1.6 | 31 | 25 | 56 | 0.67 |
| LXIN-TI4 | 33.4 | 18.1 | 1.8 | — | 21 | — | 0.68 |
| LXIN-TI5 | 30.5 | 36.6 | 0.8 | 61 | 53 | 114 | 0.50 |
| LXIN-TI6 | 19.7 | 19.9 | 1.0 | 32 | 49 | 80 | 0.36 |
| LXIN-TI11 | 27.3 | 25.3 | 1.1 | 43 | 34 | 77 | 0.45 |
| LXIN-TI12 | 31.4 | 28.3 | 1.1 | 41 | 33 | 75 | 0.49 |
| LXIN-TI13 | 36.6 | 36.7 | 1.0 | 36 | 37 | 73 | 0.32 |
| LXIN-TI14 | 25.2 | 31.4 | 0.8 | 47 | 39 | 86 | 0.28 |
| LXIN-TI15 | 24.1 | 28.4 | 0.8 | 48 | 27 | 74 | 0.40 |
| LXIN-TI16 | 26.1 | 23.5 | 1.1 | 34 | 35 | 69 | 0.40 |
| LXIII-TI1 | 28.9 | 28.3 | 1.0 | 50 | 24 | 74 | 0.31 |
| LXIII-TI2 | >17.5 | 25.1 | — | 33 | 46 | 79 | 0.32 |
| LXIII-TI3 | 21.1 | 17.6 | 1.2 | 32 | 31 | 62 | 0.50 |
| LXIII-TI4 | 15.8 | 14.0 | 1.1 | 37 | 39 | 76 | 0.53 |
| **Didactyl** | | | | | | | |
| LXIN-TI10 | — | 30.0 | — | 47 | 88 | 135 | 0.48 |

Note:
L, Track length; W, track width; L/W, length/width ratio; II–III, III–IV and II–IV, the divarication angle between digit II–III, III–IV and II–IV; M, mesaxony (=AT, anterior triangle length/width ratio); —, no measurement available.

Trackway LXIN-T5 shows functionally tridactyl, digitigrade-like and mesaxonic pes imprints that have an average length of 32.4 cm and length/width ratio of 1.2. Digit II is the shortest and digit III and digit IV are similar lengths. There is a generally wide divarication angle (59°) between digit II and digit IV and the mean mesaxony of LXIN-T5 is 0.49. The tracks are rotated slightly outwards towards the axis of the trackway. The average pace angulation is 127°. The T5-R1 is the most complete footprint amontg this trackway (Fig. 13), and as previously mentioned in the section "grallatorid morphotype", it has been radially compacted due to the behavior of trackmaker. The metatarsophalangeal pads of digit II are fairly well-developed in T5-R1, with a size almost as large as the phalangeal pad of digit IV. However, this feature is not preserved in the other two footprints in the trackway. T5-L1 and R2 exhibit a more digitigrade form with radially spread digits II and IV. L1 lacks a prominent metatarsophalangeal pad II, while R2 preserves only distinct digit traces II–IV.

The footprints in LXIN-T1 represents the more standard theropod-type semi-digitigrade pattern within this morphotype (Fig. 12B; Table 4). It contains three consecutive tracks with an average length of 31.5 cm and a length/width ratio of 1.2. Distinct external morphological changes can be found in trackway T1-L1, with the curved impression of number IV, possibly signalling its entry and exit. Digits II to IV are sandwiched between the semicircular heel track and have an overall shuttlecock-shape, with a possible large, rounded trace of digit I proximally. The morphologies of T1-R1 and L2 are comparatively conventional (Fig. 13), with a mean mesaxony of 0.54, which is overall quite close to the LXIN-T5 traces. Among other tracks and trackways, excluding size considerations, LXIN-T4, LXID-TI1 and LXIII-TI1 exhibits the greatest morphological difference compared to LXIN-T1.

The LXIN-T4 comprises four consecutive tracks (Fig. 12D), with a mean length of 24.3 cm and a mean width of 28.7 cm and mean L/W ratio of 0.9. The LXIN-T4-R1, shows a digitigrade track somewhat similar to LXIN-T5-R2 with the digit III occupies 80% of the length of the track, while its L/W ratio (0.6) is significantly lower relative to the latter (1.0). Meanwhile, in the relatively well-preserved specimens LXIN-T4-L1, R2 (Fig. 13), and LXID-TI1 (Fig. 11), the robust and proportionally shorter digits exhibit a greater resemblance to the more stoutly built *Jurabrontes* or the ornithopod ichnotaxon *Iguanodontipus* than to other theropod tracks from the Longxiang site. However, considering that the proximal portions of digits II and IV in these footprints still display marked asymmetry, and that ornithopod tracks from the Longxiang site are characterized by a distinctly elongated metatarsophalangeal region, an ornithopod affinity is deemed unlikely. Furthermore, given the considerable morphological variation within this morphotype attributable to substrate deformation, and the fact that LXIN-T4-R2 can be morphologically linked to the digitigrade forms represented by LXIN-T2-L2 and LXIN-T5-L1, these footprints are best interpreted as belonging to the same morphotype.

The LXID-T3 closely related to the outline represented by LXIN-T4, which consists of three consecutive but highly distorted tridactyl tracks (Fig. 13; Table 4). In this trackway, the digit traces appear disproportionately short and slender due to sediment backflow. This effect also results in a notably enlarged heel region in the deeper tracks, R1 and L1, whereas

in the shallower R2, only the anterior outline of the digit traces is preserved. Additionally, T3-R2 exhibits a distinct sedimentary ridge between digits III and IV. In LXID-T3-R1 and L1, the large metatarsophalangeal region is well-preserved, displaying a notable degree of development and bearing a resemblance to the tracks of large ornithopods rather than those of theropods. The sedimentary deformation in the interdigit region of L1 suggests a highly fluid substrate that differ from which of LXIN when the track formed. As the sediment becomes more hydrated, the likelihood of the occurrence of the metatarsophalangeal region increases, particularly in birds or bird-footed dinosaurs, such as the extant helmeted guineafowl (*Numida meleagris*; *Gatesy et al., 1999*).

Anomalous and significant deformation is also observed in LXIN-TI5 and TI10 (Fig. 11; Table 5); however, for these isolated track, the deformation occurs based on a more digitigrade form. This results in a larger divarication angle with excessively slender digits, rather than shorter digits combined with a more prominent heel region.

As for the remaining trackways and isolated tracks, their morphology generally falls between the aforementioned members:

In the LXIN site, the degree of deformation of the footprints is generally lighter compared to the LXIU site, and it preserves more complete trackways relative to the LXIII site.

The LXIN-T2 contains four consecutive tracks, of which R1 and L2 are in better preservation (Fig. 12A; Table 4). The average length of T2 tracks is 23.7 cm, with an average L/W ratios of 0.8. Three of these tracks have been preserved as tridactyl, and the II–III angles of all three tracks are smaller than those of III–IV. The tracks exhibit a rotation of approximately −15° with respect to the trackway axis. T2-R1 is the most typical among the LXIN-T2 tracks (Fig. 13), yet no phalangeal pads could be identified. All the digits of R1 are connected posteriorly, with deep claw marks, and occupy 62–74% of the length of the track ($N = 3$). The digit II and III traces of R1 are notably wider than those of digit IV, which are narrow and connected to the posterior margin at a small heel.

LXIN-T3-L1 and L2 only preserve digit III and IV, while the former is poorly preserved in R1. The overall L/W ratio of T3 is 1.3, similar to that of L1 and T5 (Fig. 12C).

As previously mentioned, LXIN-TI3 and 4, as well as LXIU-TI7 from LXIU site could be compared to LXIN-T5-R1, while LXIU-TI7 is in the size range of small theropod track. And of the other nine isolated tracks from LXIN (excluding TI3–TI5, TI10), are all comparable to the well-preserved LXIN-T5-L1 or T1-R1 (Fig. 13). These isolated tracks are from 26.4 to 36.8 cm in length, with mesaxony greater than 0.45 and L/W ratios between 1.1 and 1.6. Distinct external morphological changes can be found in isolated tracks TI1 and TI5, reflected in a pronounced curvature of the digit or a deformity of the heel. There was an overlap between TI6 and TI7, with the former destroying the track of the latter.

The LXIII site preserves four tridactyl theropod tracks (Fig. 11), of which LXIII-TI1 shows functionally tridactyl, digitigrade and mesaxonic pes imprint that have an average length of 28.9 cm and length/width ratio of 1.0. Digit II is the shortest, and digit III is shorter than digit IV. Digit III accounts for 64% of the total length of the footprint. There is a wide divarication angle (76°) between digit II and digit IV. The divarication angle

between digits II and III (37°) is almost equal to the one between digits III and IV (39°). Each digit impression ends in a sharp claw mark. The mean mesaxony of LXIII-TI1 is 0.51.

The other three tridactyl theropod footprints from site LXIII are all less than 25 cm in length. TI2 lacks the heel but comparable to TI1 in width, TI4 is 15.8 cm in length, has a length/width ratio of 1.1, and a mesaxony of 0.53, and the overall morphology is similar to the the Theropod morphotype A.

**Comparison and discussion.** Among the best preserved trackways, LXIN-T1 and T5, the average mesaxony of is about 0.54 and 0.49 (Fig. 12, Table 4), which is typical for the morphofamily Eubrontidae (*Lull, 1953*). LXIN-T5-R1 track shows convergent traits with non-*Asianopodus* *Eubrontes*-type tracks (see the defination in previous section), especially the overall evenly developed digit II and IV, and the separated metatarsophalangeal pad II and IV. However, the capability of reaching higher divarication angle in the footprints shows higher digitigrade level is also differ from the most typical *Eubrontes*, such as the holotype AC 151 (*Olsen, Smith & McDonald, 1998*) and the Cretaceous *E. nobitai* from East Asia (*Xing et al., 2021c*).

Conversely, their loosely arranged digits, combined with a size range positioned at the lower boundary of large theropods (25–30 cm), suggest an intermediate morphotype between *Eubrontes* and *Megalosauripus*, more closely resembling the *Eubrontes*-type. Examples include *Kayentapus*, particularly the type specimen of *K. hopii* (*Lockley, Gierlinski & Lucas, 2011*), as well as the typical Cretaceous North American *Eubrontes* from the Mail Station Dinosaur Tracksite (*Lockley et al., 2021*), which exhibits a larger divarication angle compared to its Jurassic counterparts. The more robust footprints, such as LXIN-T4, LXID-TI1, and LXIII-TI1, resemble the *Kayentapus* Asturias specimen (*Avanzini, Piñuela & García-Ramos, 2012*). However, compared to *Eubrontes* and *Kayentapus*, the heel region of the Longxiang Theropod morphotype A is less distinctly preserved, evoking similarities with *Carmelopodus* (*Lockley et al., 2014b*), which is primarily defined within the small-to-medium size range. Nonetheless, despite some similarities in exhibiting a relatively strict digitigrade posture, the large theropods from the Longxiang site display relatively low mesaxony, distinguishing them from large grallatorid tracks.

Besides, the fleshy digits and the large, rounded, posteriorly slightly protruded heel pads in some of the morphotype A tracks are somewhat reminiscent of large neornithischian tracks with fleshly pads for weight-bearing, including basal ankylopollexian mainly from lower Cretaceous of Europe (*Iguanodontipus*), and axially compacted dryosaurid ornithopods or relatively large basal neornithischian tracks mainly record in upper Jurassic of Europe, North America and lower Cretaceous of Europe (*Dinehichnus*) (*Diaz-Martinez et al., 2015*; *Castanera et al., 2020*; *Foster et al., 2024*; *Xing et al., 2024b*). They are also similar in structure to moa footprints from Newzeland (especially the ornithopod-like *Moapus* and *Dinornipus*; *Hunt & Lucas, 2024*). However, digit II of T4-R1 preserves slender, sharp claw trace that more comparable to theropod footprints, and they are remarkablely distinguished from the aforementioned ornithopod and moa ichnospecies by the frequent absence of metatarsophalangal pad within a particular trackway.

In summary, among the currently established ichnotaxa, *Kayentapus* or *Eubrontes* may be the most appropriate classification for this morphotype. However, given that *Kayentapus* remains primarily defined within the Jurassic, we propose classifying the large-sized and morphologically distinct members of Theropod morphotype A as *Eubrontes isp.*, while the significantly smaller tracks should be assigned to *cf. Eubrontes*, or left unclassified due to the absence of referable trackways.

*Xing et al. (2021c)* reviewed most of the records of *Eubrontes* in China. The Lower Cretaceous of China has yielded a considerable record of theropod tracks of the generalized *Grallator- Eubrontes* plexus type, which are typical in Jurassic formations of North America and China (*Xing et al., 2021c*). This has extended the stratigraphic and palaeobiogeographic range of these generalized theropod trackmakers into Cretaceous East Asia (*Xing et al., 2018b*). The records of this morphotype extend this assembly to the early Late Cretaceous.

**The primary comments on the validity of *Tridentigerpes*.** It is worth noting that in trackways LXIN-T2, T4, and T5, which comprise a mixture of both semi-digitigrade and digitigrade footprints, the semi-digitigrade forms—characterized by a combination of lower mesaxony, a higher divarication angle, and a reduced proportion of the metatarsophalangeal region (*e.g.*, T2-L2 and T4-R2; Fig. 13)—resemble *Tridentigerpes huashibanleei* from its type series, particularly HSB-T2-R1. Additionally, LXIN-T2-R1 and L2 exhibit morphological consistency with HSB-T2-L1 and R2 (Fig. 3 in *Xing, Lockley & Romilio, 2021*).

Within Theropod morphotype A, footprints such as LXIN-TI5 and TI10, which have undergone significant distortion, also show similarities to *T. huashibanleei*. These footprints are characterized by elongated digit traces and an overall robust appearance, with divarication angles ranging from 88° to 135°, which closely aligns with the type series of *T. huashibanleei* (93°–130°). However, they exhibit slightly lower values compared to the large-sized referred specimens of *Tridentigerpes* from the Morrison Formation of Colorado, which reach a maximum of approximately 150° (Fig. 6 and Table 1 in *Xing, Lockley & Romilio, 2021*). Regarding the relatively low mesaxony observed in large *Tridentigerpes* specimens (ranging from 0.26 to 0.34 in the Huashiban specimens and 0.38 to 0.51 in the Colorado specimens), the footprints assigned to Theropod morphotype A at the Longxiang site exhibit a comparable range (0.29–0.5) despite their relatively elongated digit lengths. The co-occurrence of *Tridentigerpes-like* and *Eubrontes-type* non-avian dinosaur tracks within the same area, and even within the same trackway at the Longxiang site, preliminarily suggests that these two morphotypes may be linked through transient variations in digitigrade levels during locomotion. Furthermore, all referred specimens of large *Tridentigerpes* exhibit significant distortion, with no known examples displaying the less-deformed morphology observed in the type series.

Although Theropod morphotype A does not entirely match the type series of *Tridentigerpes*, the findings suggest that, rather than resembling *Carmelopodus*, the distinctive "digitigrade" form of large *Tridentigerpes*, characterized by a high divarication angle, is unlikely to result from the trackmaker's intrinsic foot morphology, stable locomotor behavior, or an avian-like trackmaker. Instead, it is more plausibly interpreted

as a temporary locomotor adaptation in response to a slippery substrate. This interpretation raises doubts about the ichnotaxonomic validity of *Tridentigerpes*, particularly regarding the relationship between *T. pinuelai*, a smaller morphotype that more closely resembles typical bird ichnotaxa, and *T. huashibanleei*, the type ichnospecies. Consequently, we propose the removal of *T. pinuelai* from *Tridentigerpes*. However, due to the weathering of the *in situ* specimens of *T. pinuelai*, and the lack of clear reference material beyond the low-quality images provided in *Xing, Lockley & Romilio (2021)*, we refrain from formally reclassifying it in this study. The issue of *Tridentigerpes*' validity will be further examined in the following sections, incorporating additional specimens for a more comprehensive evaluation.

**Speed estimates.** *Thulborn (1990)* put forth the proposition that hip height (h) = 4.5 × track length (Fl) for small theropods (track length < 25 cm), and h = 4.9 × FL for large theropods. After this, the stride length/hip height ratios of the LXIN-T1, T2, T4 and T5 trackway are 1.34, 2.32, 1.9 and 1.14 respectively, mostly indicating a trotting gait, except T4 shows fast walking gait; the largely distorted LXID-T3 rather show a trotting gait (1.82). The speed of LXIN-T1, T2, T4 and T5 trackmakers are 5.69, 11.92, 8.58, and 4.39 km/h respectively, and which of LXID-T3 is 7.86 km/h.

Given the considerable density of tracks at the LXIN site, it seems reasonable to suggest that subsequent trackmakers slowed down when confronted with the rugged substrate in the bustling area. It is important to note, however, the co-existence of the absence of the heel traces, low divarication angle and the low L/W ratio is correlated with an incomplete estimation of foot length in LXIN-T2 and T4, by which the estimated velocity would be relatively large when using the aforementioned methods.

### Theropod morphotype B

**Description.** The Theropod morphotype B trackways includs LXIU-T2 and LXID-T1 (Table 4), which are characterized by their large divarication angles of ~100°–110° and the total absence of distinguishable metatarsophalangeal region. These resulting in exceptionally low L/W ratio of ~0.5 to 0.6, and the concomitant weak mean mesaxony of <0.3 (Fig. 13).

The LXIU-T2 trackway is representative of the type B from the Longxiang site, comprising seven consecutive tridactyl tracks (Fig. 13). The preservation condition of the LXIU-T2 is classified as level 2 on the scale proposed by *Belvedere & Farlow (2016)*. The mean length of the T2 tracks is 14.7 cm, with a mean width of 29.3 cm. The mean L/W ratio is 0.5, and mesaxony is 0.29. The digit traces are clearly discernible and distinct, and the claw marks are readily apparent. The length of digit III occupies between 87% and 92% of the length of the track, as observed in three specimens. All the tracks exhibit markedly distinct divarication angles between digits II and IV, with a mean value of 107° and a range of 100° to 114°. The trackway is notably broad, with a pace angulation of 133° and a stride length of approximately 6.7 times the track length. The tracks exhibit a rotation of approximately 9° (between −9° and 8°) from the trackway axis.

LXIU-T2-R1 is the most well-preserved of the LXIU-T2 tracks (Fig. 13). However, no phalangeal pads can be identified. The trace of digit III is notably broader in comparison to

the traces of digits II and IV, and the posterior margin of the footprint is not distinctly delineated. The digit III of R2 is distinctly conical in shape, exhibiting the sharpest claw marks; digit II is subfusiform with the deepest claw marks; digit IV is the smallest and oval in shape, with claw marks only visible at the anterior margin. Comparatively, L1, L2, and R3 in T2 only preserved digit II and III, and digit II of them show claw marks tangent to the median axis of the digit, as an ectomorphological feature as the lifting of digit.

Morphologically, almost all the features and dimensions of track LXID-T1 match those of LXIU-T2 (Fig. 13). LXID-T1 consists of seven tridactyl footprints with an average length of 14 cm and an average width of 30.7 cm. This gives an average L/W ratio of 0.5. LXID-T1 appears to have a longer stride length, which is approximately 10.3 times the length of the track. The tracks have an outward rotation of approximately 28° (with a range of −45° to 24°) from the track axis.

**Comparison and discussion with _Tridentigerpes_.** _Xing, Lockley & Romilio (2021)_ provided diagnostic features of _Tridentigerpes_ including: weakly mesaxonic tridactyl tracks with unusually wide digit divarication, averaging from ~107°~135° in three samples, and short digit III giving pes length width ratios of ~0.5–0.8. Trackway narrow with step typically 4 time track length. Compared to the aforementioned Theropod morphotype A, the Theropod morphotype B in Longxiang sites generally more consist with the diagnostic features of _Tridentigerpes_, especially in the exceptionally low length/width ratio of ~0.5 that separates _Tridentigerpes_ from the other ichnogenera. Specifically, the other detailed characteristics of morphotype B are almost completely consistent with _T. huashibanleei_, including wide digit divarication angles (both mean 107°), the mesaxony (both are 0.29), and the tendency of the inward rotation of pes (Table 4). Minor difference occurrence between overall feature of morphotype B tracks and _T. huashibanleei_: the former is higher in the relative speed with an average step length for about 6.7 times footprint length, different from 4 times of _T. huashibanleei_. In addition, _T. huashibanleei_ tracks are rotated approximately 25° inward from the trackway axis, type B tracks rotated inward and outward with small angles. Among the Longxiang tracks, LXIU-T2-R1 show very similar outline to HSB-T2-L1, including the similar anterior extension of digit II. However, in LXIU-T2 and LXID-T1, the relative positions of the ungual traces of digits II and IV are inconsistent. For instance, in LXID-T1-L1 and L2, the degree of distal extension of digit II varies, while in LXIU-T2, there are significant differences in the abduction of the lateral digits between R2 and L2. This indicates that for large _Tridentigerpes_, the only remaining diagnostic features are the L/W ratio and divarication angle, which are more influenced by environmental factors rather than the trackmaker itself.

In conclusion, this morphotype does not exhibit a stable and distinctive pedal morphology, consistent kinematic characteristics (_e.g._, high-digitigrade posture), or significant behavioral indicators (_e.g._, swim tracks) that would justify its classification as an independent ichnogenus. Therefore, we consider _Tridentigerpes_ to be a _nomen dubium_. Considering that Theropod morphotype A and morphotype B coexist at both the LXIN and LXIU sites and share a similar size range, it is more reasonable to classify morphotype B as _cf. Eubrontes_. However, it is important to note that, unlike morphotype A, which
preserves more complete footprints, morphotype B lacks sufficient diagnostic features and may correspond to more than one ichnogenus.

Furthermore, if the claw traces are not sharply defined, the possibility of an ornithischian affinity should also be considered. This suggests that directly assigning *Tridentigerpes* from the Huashiban site—which resembles Theropod morphotype B—to *cf. Eubrontes* would be inappropriate. At the Huashiban site, the tridactyl ichnogenus that coexists with *Tridentigerpes* within the same tracksite and falls within a similar footprint width range (25–35 cm) is *Dinehichnus*. The primary differences between *Dinehichnus* and *Eubrontes*-like tracks with a higher divarication angle (*e.g.*, *Kayentapus*) lie in the connection between the metatarsophalangeal region and the digits, the clarity of phalangeal pads, and statistically distinct symmetry patterns. However, these distinguishing features cannot be observed in the Huashiban *Tridentigerpes*, as only the anterior portions of the digits are preserved. Therefore, following the classification approach for Longxiang Theropod morphotype B, we propose that the Huashiban *Tridentigerpes* should be regarded as *cf. Dinehichnus*.

**Speed estimates.** Similar to which applied to morphotype A, hip height (h) = 4.9 × FL for large theropods from *Thulborn (1990)* are used to estimate the speed of trackmaker. After this, it is inferred that the relative stride length (SL/h) of the LXIU-T2 and LXID-T1 are 2.72 and 4.44, which indicate a trotting (slow running) and scarce running gait, and have estimated speeds of 12.14 and 26.93 km/h, respectively. Also similar to the LXIN-T2 and T4, estimated velocity of morphotype B would be relatively higher when considering the footprint length of LXIU-T2 and LXID-T1as the actual length of the pes. Additionally, compared to morphotype A trackways, in which the metatarsophalangeal region is preserved to varying degrees and track length is relatively complete, the estimation of speed for morphotype B trackways—where this region is entirely absent—is more significantly affected by the aforementioned factors. Consequently, the current speed estimates are likely to be considerably higher than the trackmaker's actual speed and should be interpreted as representing only the upper limit.

**The formation of morphotype B.** The short, pointed tracks with extremely low mesaxony, can also be seen in buoyancy-related punting tracks (*Romilio, Tucker & Salisbury, 2013*; *Navarro-Lorbés et al., 2023*). It may be hypothesized that the morphology resembling morphotype B and the "T. huashibanleei" may be formed in a swimming-like state with the pes in a distinctly oblique position to the ground. However. the specific genesis of this state is not unique and may be behavioural (*McAllister & Kirby, 1998*) or by substrate condition (*Milàn, 2006*). Both of these can be manifested as the distal end of digit III moving relatively proximally in the plane. With regard to the large divarication angle, it may be a consequence of the oblique intersection with the ground during footfall (*Milàn, 2006*). However, in purely substrate-related speculation, the large divarication angle is only observed on the upper surface of muddy substrate that is sufficiently wet and soft to retain plastic deformation at the track (*Milàn, 2006*). The presence or absence of heel traces on the upper surface of the substrate is indicative of the degree of weathering. In lower surfaces where heel traces are absent, the divarication angle is not significantly larger than

**Table 6 Measurements (in cm and degrees) comparison of *Caririchnium* trackways from multiple area of Longxiang site I.** All data are averaged within precise trackway, and Oxp and Oxm are the mean values of pes and manus respectively within the same trackway. In trackway LXIU-O2, 4 and 8, only one manus track can be identified in the trackway.

| | L | W | W′ | L/W | L/W′ | II–IV | II–III | III–IV | PL | SL | PA | R | M |
|---|---|---|---|---|---|---|---|---|---|---|---|---|---|
| **LXID** | | | | | | | | | | | | | |
| LXID-O1 | 33.11 | 24.18 | 28.35 | 1.33 | 1.20 | 49 | 24 | 25 | 91.47 | 186.08 | 161 | \|7\| | 0.32 |
| LXID-O2p | 22.80 | 17.19 | 20.27 | 1.40 | 1.10 | 49 | 28 | 21 | 60.37 | 120.41 | 154 | \|10\| | 0.27 |
| LXID-O2m | 6.78 | 4.97 | — | 1.37 | — | — | — | — | — | — | — | — | — |
| LXID-O3 | 20.77 | 16.62 | 16.17 | 1.29 | 1.26 | 61 | 38 | 24 | 54.11 | 101.40 | 144 | \|11\| | 0.49 |
| LXID-O4 | 21.86 | 17.17 | 19.82 | 1.34 | 1.09 | 46 | 22 | 21 | 64.79 | 122.65 | 146 | \|12\| | 0.28 |
| **LXIU** | | | | | | | | | | | | | |
| LXIU-O1 | 27.50 | 20.65 | 31.81 | 1.24 | 0.90 | 59 | 30 | 29 | 87.00 | — | — | — | 0.33 |
| LXIU-O2p | 31.10 | 26.05 | 34.89 | 1.20 | 0.89 | 59 | 31 | 28 | 141.00 | — | — | — | 0.24 |
| LXIU-O2-L1m | 14.04 | — | 5.96 | — | 2.36 | — | — | — | — | — | — | — | — |
| LXIU-O3 | 9.98 | 7.60 | 9.15 | 1.24 | 1.10 | 55 | 23 | 32 | 36.39 | 71.16 | 153 | 6 | 0.25 |
| LXIU-O4p | 27.21 | 21.11 | 30.99 | 1.29 | 0.89 | 51 | 26 | 25 | 121.83 | 227.33 | 141 | \|8\| | 0.14 |
| LXIU-O4-R3m | 12.69 | — | 8.66 | — | 1.47 | — | — | — | — | — | — | — | — |
| LXIU-O5 | 26.99 | 25.33 | 33.73 | 1.07 | 0.80 | 63 | 34 | 28 | 97.89 | 192.47 | 160 | \|5\| | 0.21 |
| LXIU-O6 | 11.10 | 8.86 | 10.71 | 1.39 | 1.04 | 52 | 28 | 24 | 40.98 | 79.77 | 154 | — | 0.34 |
| LXIU-O7 | 18.27 | 15.12 | 17.57 | 1.21 | 1.05 | 55 | 24 | 32 | 52.26 | 102.33 | 161 | \|12\| | 0.24 |
| LXIU-O8p | 44.54 | — | 33.30 | — | 1.38 | — | — | — | 112.01 | 221.23 | 160 | \|3\| | — |
| LXIU-O8-L1m | 13.00 | — | 7.88 | — | 1.65 | — | — | — | — | — | — | — | — |
| LXIU-O9 | 9.99 | 9.01 | 10.15 | 0.91 | 1.01 | — | — | — | 52.57 | 104.15 | 162 | — | — |
| LXIU-O10 | 35.86 | 25.71 | 32.09 | 1.48 | 1.13 | 46 | 23 | 23 | 93.02 | 184.88 | 155 | \|8\| | 0.28 |
| LXIU-O11 | 36.90 | 21.70 | 25.96 | 1.34 | 1.43 | 60 | 35 | 25 | 92.70 | 168.75 | 124 | 12 | 0.42 |
| LXIU-O12 | 39.50 | 27.08 | 32.80 | 1.50 | 1.21 | 45 | 25 | 20 | 100.25 | 194.60 | 152 | 29 | 0.37 |
| LXIU-O13 | 12.23 | 14.03 | 14.39 | 0.87 | 0.82 | 97 | 60 | 37 | 36.46 | 60.79 | 129 | \|14\| | 0.18 |
| **LXIE** | | | | | | | | | | | | | |
| LXIE-O1 | 21.46 | 13.45 | 17.01 | 1.46 | 1.26 | 20 | 21 | 46 | 50.68 | — | PA | — | 0.26 |

Note:

L, Track length; W, track width; W′, track width with the inclusion of deformation rim; L/W and L/W′, length/width ratio with different width measurements; II–III, III–IV and II–IV, the divarication angle between digit II–III, III–IV and II–IV; PL, pace length; SL, stride length; PA, pace angulation; R, rotation angle; M, mesaxony (=AT, anterior triangle length/width ratio); —, no measurement available.

that of conventional tracks (*Falkingham & Gatesy, 2014*; *Turner, Falkingham & Gatesy, 2020*).

In conclusion, it suggests that the digit-only morphology of Theropod morphotype B may have a behavioural genesis that approximates swimming. Further speculation is contingent upon further observation and comparison of both referred trackways of morphotype B and HSB-T2 from Hushiban site.

### *Caririchnium* morphotype

#### Description

The ornithopods left the most abundant footprints in the Longxiang area, which spreading through all the LXI, LXIII sites (Figs. 5–7; Table 6), and the size of the tracks exhibited considerable variation. The size variation in LXI sites can be divided into five intervals: less

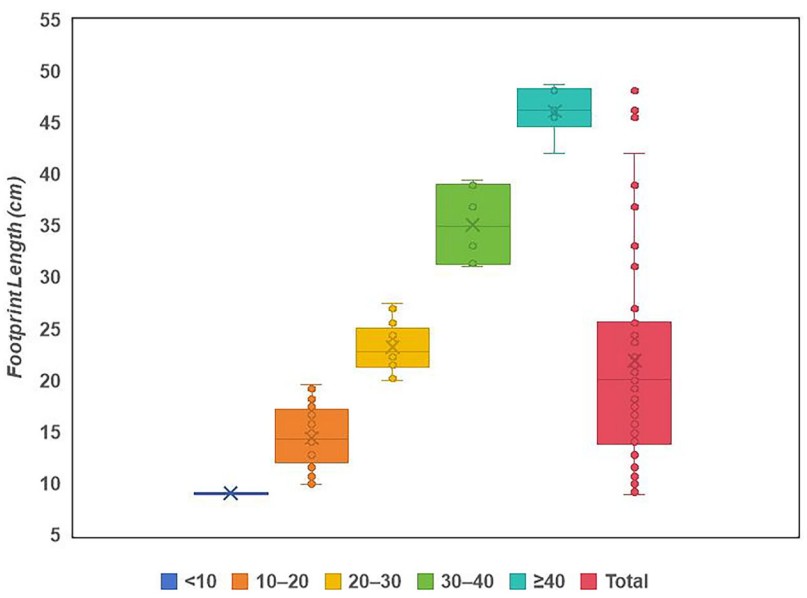

**Figure 14 Box plot of ornithopod tracks size distribution from all the LXI tracksites.** The ornithopod track size is represented by the track length (in cm), for no anomalous length/width ratios are observed.

than 10, 10–20 (20 cm excluded), 20–30 (30 cm excluded), 30–40 (30 cm excluded), and ≥40 cm. The number of preserved tracks within each interval is as follows: 2, 31, 21, 6, and 6, respectively. Although the length appears to be continuous, a relatively clustered area of track lengths between 12 and 26 cm can be observed in the box plots (Fig. 14). However, the preservation condition of ornithopod tracks in this area is quite limited, with the majority being at level 1. These tracks are shallow, blurred, or deformed, yet still recognisable as digit traces. Some claw traces were also present; the manus and pes tracks are distinguishable, with the former retaining only a rough outline. Of the 66 ornithopod trackways, 14 (21%) preserved manus tracks.

Four ornithopod trackways are preserved in LXID site, numbered LXID-O1–O4, with track lengths ranging from 20.8 to 33.1 cm (Figs. 15, 16).

Eight pes tracks are preserved in LXID-O1. The pes tracks of LXID-O1 are mesaxonic and functionally tridactyl, with a mean length of 33.1 cm and an average length/width ratio of 1.2 (Fig. 15; Supplemental Information 2; Table 6). The mean mesaxony is 0.32. In the better-preserved O1-R3 and L5, digit III is observed to project the farthest anteriorly, followed by digits IV and II. Each digit trace exhibits a pronounced ungual or claw mark. A distinct border is absent between the heel and the three digits. The O1-R2 and L3 specimens exhibit an extended heel, which may be part of the metatarsal impression. This is potentially caused by the deeper deposition of the metatarsal. The mean interdigital divarication II–IV is 49°. The average pace angulation of LXID-O1 trackway is 161°. The pes traces show outward and inward rotation mean 7° from the trackway axis.

The LXID-O2, 3 and 4 trackways preserved approximately 15, 10 and 14 tracks respectively. All tracks were very similar in length, ranging from 20.8 to 22.8 cm (Fig. 16;

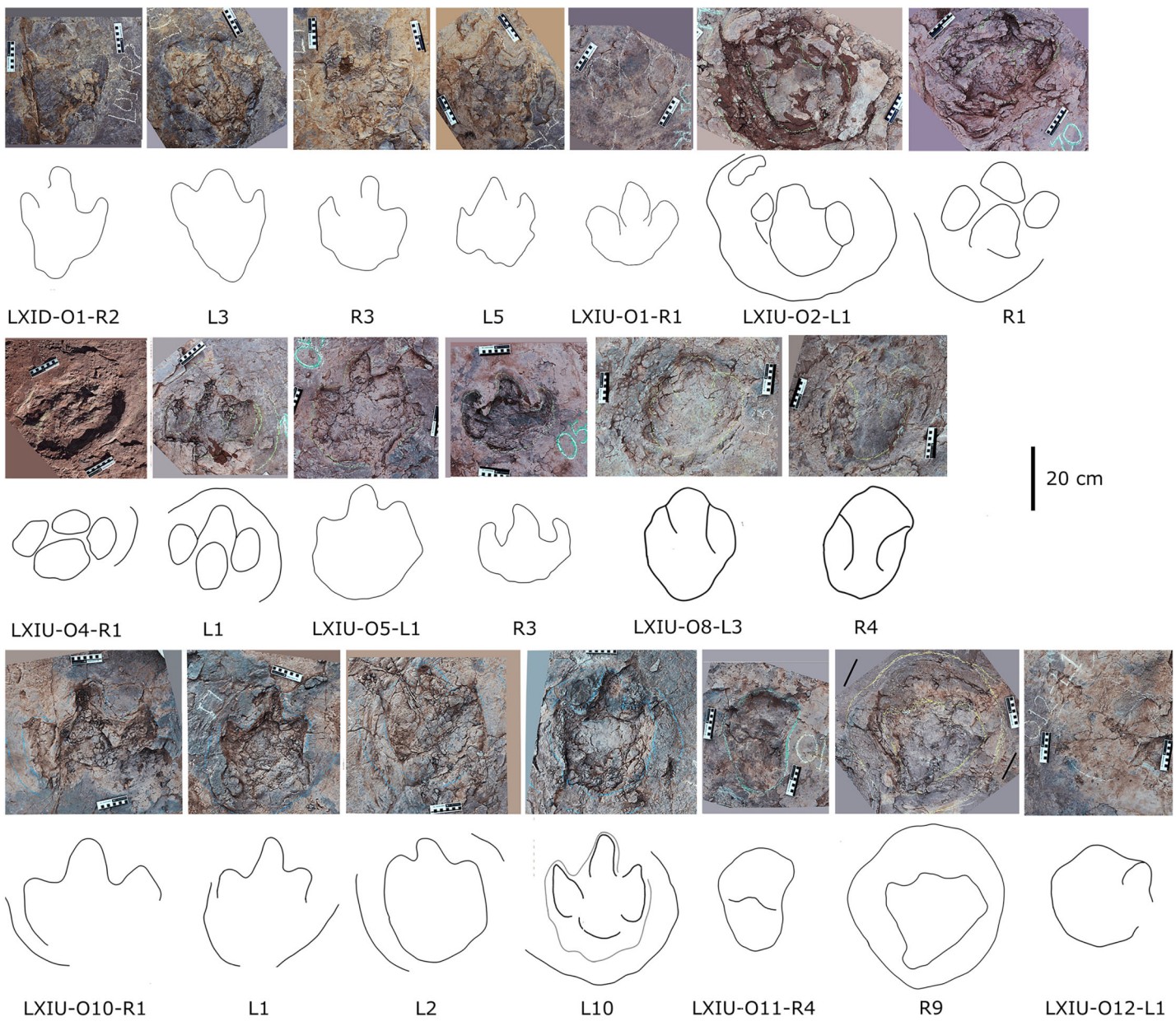

**Figure 15 Close-up photographs and interpretive line drawings of relative larger *Caririchnium* morphotype tracks from trackways in LXID and LXIU.** All the tracks share the similar scale bar.

Supplemental Information 2; Table 6). Out of a total of 39 tracks, only about four had better preserved morphological characteristics. For example, of the 10 tracks in O3, only L4 shows a distinct tridactyl pattern, the length is 19.4 cm, the length/width ratio is 1.2 and the mesaxony is 0.44. The other tracks in O3 are irregularly elliptical, but the L/W ratios are essentially in the range of 1.1 to 1.4. O2 preserves two possible manus tracks: R9m and R10m are located in the outer part of digit IV of the pes, with an average length of 5.2 cm and an L/W ratio of 0.7. Due to the poor preservation of the individual

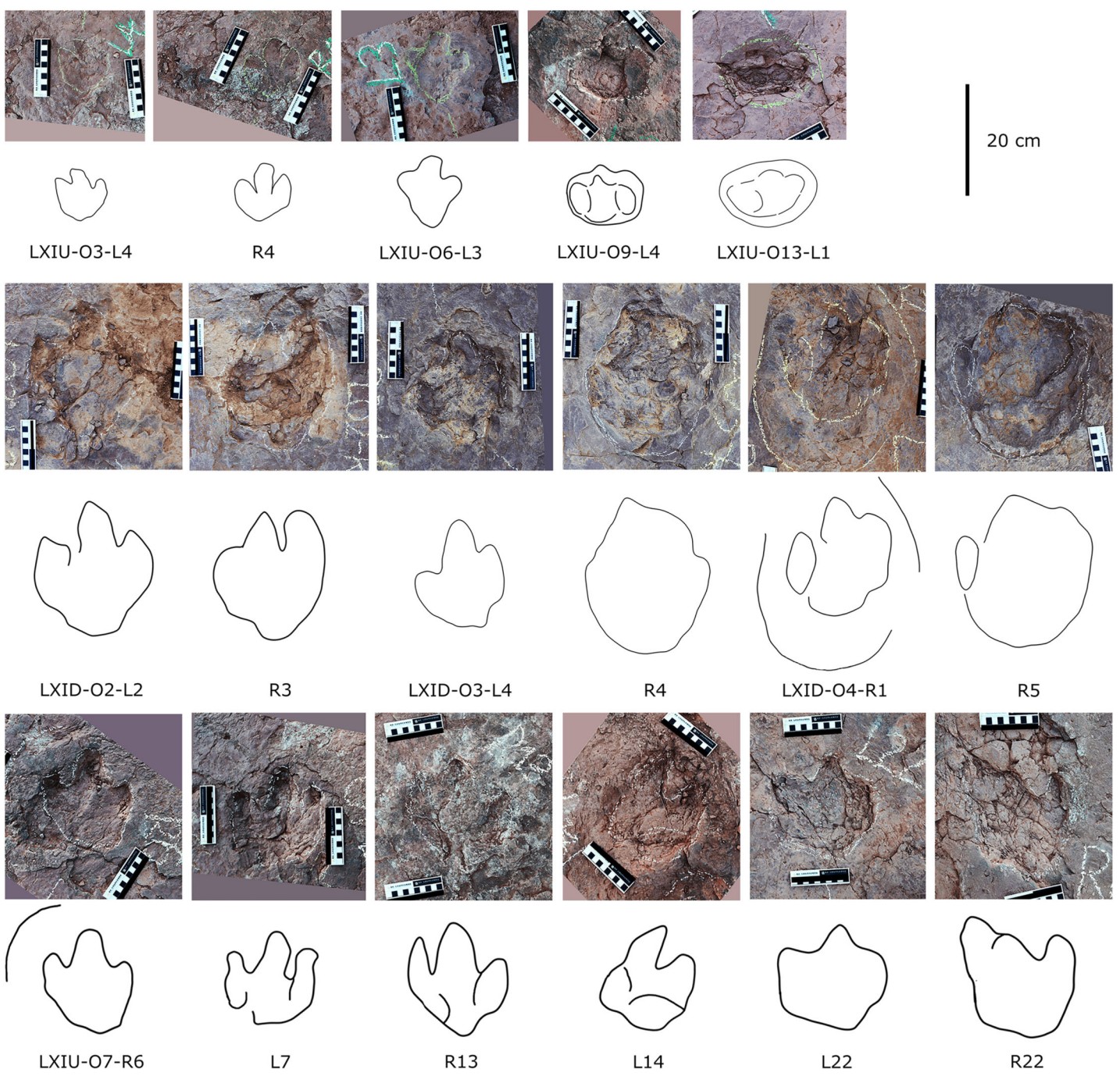

**Figure 16 Close-up photographs and interpretive line drawings of relative smaller *Caririchnium* morphotype tracks from trackways in LXID and LXIU.** All the tracks share the similar scale bar.

tracks, the information on the trackways is remarkable. The pace angulation of LXID-O2, 3 and 4 are 154°, 144° and 146° in average, respectively. The tracks are predominantly outwardly rotated, being 10°, 11° and 12° from the track axis in the three trackways respectively.

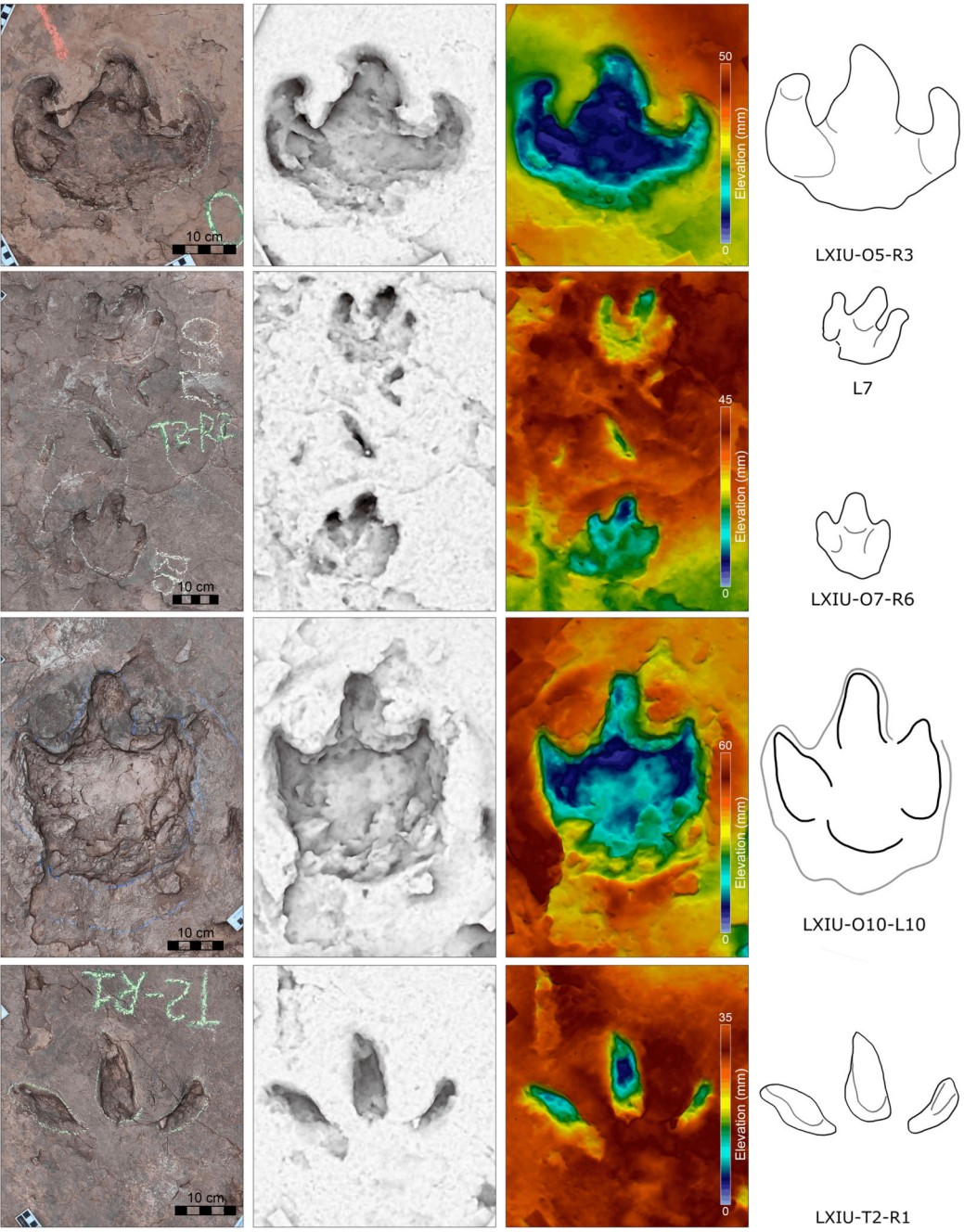

**Figure 17 Close-up photographs, 3D models & depth maps and interpretive line drawings of representative ornithopod tracks in trackways of different sizes from LXIU site.** With the comparison with theropod morphotype B (LXIU-T2-R1).

LXIU site preserves 13 ornithopod trackways and a number of isolated tracks, the former numbered LXIU-O1-O13, with tracks ranging from 10 to 44.5 cm in length (Figs. 13, 14, 15–17).

LXIU-O3, O6, O9 and O13 represent the smallest ornithopod trackways (10 to 12.2 cm in length) at the LXIU site (Fig. 16; Supplemental Information 2; Table 6).

LXIU-O3 preserved nine pes tracks that are mesaxonic, functionally tridactyl with an average length of 10 cm, an average length/width ratio of 1.2 and an average mesaxony of 0.25 (Fig. 16). In the better preserved O3-R4, digit III projects most anteriorly, followed by digits IV and II. The digits II and IV are almost equal in length. Each digit trace has a strong and blunt claw or ungual mark. There is no distinct border between the heel and the three digits. The average interdigital divarication II–IV is 55°. The average toe angulation of the LXIU-O3 trace is 153°. The pes traces show an average outward rotation of 6° from the trackway axis.

LXIU-O6, O9 preserved five and four pes traces, respectively (Fig. 16). Both trackways had limited preservation, with tridactyl morphology discernible in O6-L3, and numbers II and IV discernible in O9, the other tracks representing oval pits.

LXIU-O13 has preserved five pes tracks (Fig. 16). All except O13-L1 have very pronounced ectomorphological distortions, most likely due to highly fluid sediments. O13-L1 is mesaxonic, functionally tridactyl with a length of 8.2 cm, and the mean length/width ratio is 0.6, the mean mesaxony is 0.18, representing very broad contours and extremely low mesaxony.

LXIU-O1, O4, O5 and O7 represent medium sized ornithopod tracks ranging from 18 to 27.5 cm from the LXIU site (Figs. 15–17; Supplemental Information 2; Table 6).

LXIU-O1 has only a single step, R1 is well preserved, functionally tridactyl with a mean length of 31.6 cm, with a mean length/width ratio of 0.9 and a mean mesaxony of 0.31. The interdigital divarication of digits II-IV is 69°. The three digits are almost equal in width and length from the hypex, while the claws of digit III are more acute than those of digits II and IV.

LXIU-O4 is probably the best preserved ornithopod trackway on the LXIU site (Fig. 15). It contains five tracks. All tracks are functionally tridactyl with an average length of 26 cm, the average length/width ratio is 0.8, the average mesaxony is 0.14, the interdigital divarication of digit II–IV is 51°. There is a distinct border between the heel and the three digits. The digits II and IV are almost the same length. Each digit has a strong and blunt claw or ungual mark. The average pace angulation of LXIU-O4 trackway is 141°. The pes traces show inward rotation, on average 12° from the trackway axis. LXIU-O4-R1 has a slightly smaller digit III than II and IV, and its heel is quite large, almost twice the size of the individual digits. This feature is not present on the other tracks. The mesaxony of O4-L1 is significantly above average at 0.35. The heel of R3 has a drag mark and preserves a large manus track located between digits III and IV, the track is wider than long, measuring 8.6 cm in length, with a L/W ratio of 0.7, and the track is rotated outwardly by 43°, relative to the axis of the pes. The ratio between the area of the manus track and the pes track is 0.22. Overall, the average length of the right track is 24.1 cm, which is less than that of the left track at 28.7 cm, and may be the result of a large difference between the left and right foot sizes of this trackmaker. Even when well preserved, where each track is clearly quartered, the features of LXIU-O4 are unstable, which is an ectomorphological distortion caused by slippery substrates.

LXIU-O5 preserved nine tracks of which three were incomplete (Fig. 15). The mean length of the tridactyl tracks is 27 cm, with an average length/width ratio of 0.8 and a mean

mesaxony of 0.21. The interdigital divarication of digit II–IV is 51°. A clear delineation between the heel and the three digits is absent. The length of digits II and IV is approximately equal. Each digit trace has a strong and blunt claw or ungual mark. The mean pace angulation of the LXIU-O5 trackway is 160°. The pes traces exhibit outward and inward rotation, with a mean deviation of 5° from the trackway axis. O5-L2, R2, L3 and R3 are particularly well-preserved, with clear claw marks visible on digit IV of L2 and on digits II–IV of R3 (Fig. 17). The claw marks on R3 are the most well-preserved, with a clearly defined boundary. However, other areas of the internal deposition have not been completely removed, which impedes further observation. O5-L1 has a large heel that is twice the length of the other tracks in the same trackway, which may be the result of metatarsals being involved in the formation of the impression, or it may have been formed by the fragmentation of the track layer.

LXIU-O7 is the longest ornithopod trackway at the Longxiang site, with a length of 23 m and preserving about 43 tracks, which is probably the longest ornithopod trackway in China (Fig. 16). It commences with an orientation to the northwest and then turns 90° to the southwest after ~13 m, subsequently extending progressively in a southward direction. The pes tracks of LXIU-O7 are mesaxonic and functionally tridactyl, with a mean length of 18.3 cm (Fig. 17). The average length/width ratio is 1.1, and the mean mesaxony is 0.24. In the well-preserved R6 and R13, it can be clearly observed that digit III projects the farthest anteriorly, followed by digits II and IV. Each digit trace has a strong and blunt claw or ungual mark. A clear border between the heel and the three digits is absent; however, at times, the boundaries between digits or between digits and the heel area can be discerned. The mean interdigital divarication II–IV is 55°. The mean pace angulation of the LXIU-O7 trackway is 161°. The pes tracks predominantly exhibit outward rotation, with an average deviation of 23° from the trackway axis. Within the trackway, O7-R11, L12, R12, and L13 represent the location of the change of direction, and the outward rotation angles of them were 17°, −58°, 25°, and −8° separately. From R11, there is a notable decrease in stride length from 114.8 to 86.8 cm, followed by a further decrease to 66.8 cm, after which it gradually recovers. From L12, pace angulation decreases from 154° to 102°, before recovering to 166°. In the aforementioned process, the trends of the track rotation angle and pace angulation were found to be largely consistent. These observations indicate a reduction in the speed of the trackmaker during changes in direction.

The trackways LXIU-O2, O8, O10, O11, and O12 comprise the largest tracks observed on the same track layer (Figs. 15, 17; Supplemental Information 2; Table 6). Of these, O8 is the largest, with an average track length of 44.5 cm, while the other trackways range from 31.1 to 39.5 cm in length.

The LXIU-O2 trackway preserves only two tracks, forming a single step (Fig. 15). The mean length of these tracks is 31.1 cm, with an average length/width ratio of 0.9 and a mean mesaxony of 0.24. The interdigital divarication of digit II–IV is 59°. Among them, L1 contains a manus track on the lateral side of digits III and IV, which represents about 6 cm in length, with a L/W ratio of 0.4. However, as the sediment filled L1, forming a unweathered natural cast, the true morphology of the specimen remains uncertain.

The preservation of LXIU-O8 is rather limited (Fig. 15). Although the average length is known to be 44.5 cm, a considerable number of ectomorphic distortions affect this measurement. The average pace angulation of the LXIU-O8 trackway is 160°. The best preserved track in O8 is O8-L3, with a length of 36.9 cm, which can be assumed to represent the true size of LXIU-O8. The length/width ratio of L3 is 1.2, and the mean mesaxony is ~0.3. O8-L1 preserves a manus track on the lateral side of digits III and IV, which is ~7.6 cm in length, with the L/W ratio of 0.6. Given the similarity of the pace angulation to that observed in other ornithopod trackways and the presence of a manus track, it can be posited that this rather limited preserved trackway still has strong ornithopod affinities. The L/W ratio of R7 and R8 reaches 1.8–1.9, which is a relatively uncommon elongation-related distortion observed in ornithopod tracks. This may have resulted from either sliding due to slippery sediments or as partial metatarsal impressions.

The large heel traces are also evident in LXIU-O11 and can be classified into two categories: wide and narrow (Fig. 15). The type with wider heels is exemplified by O11-L1 and L7, both of which exhibit a L/W ratio of 2.1. The elongated heel of L7 is approximately three times as long as the anterior part of the track, and the proximal section is similar in width to the distal end. The distal end preserves recognisable tridactyls. The presence of triple elongated digit traces with identifiable claw traces anteriorly is indicative of a substrate that is slippery. This can be observed in the case of the theropod track from the Middle Jurassic of the Turpan Basin, Xinjiang (*Xing et al., 2014c*). Nevertheless, the digits of LXIU-O11-L7 have not been separated as in theropod tracks. The type with narrower heels includes O11-R1, L2 and R6. These tracks exhibit a markedly constricted heel, with the proximal end measuring between one-half to one-third the width of the distal end. These morphologies are comparable to those observed in metatarsal impressions commonly associated with tridactyl theropod tracks or basal ornithischian tracks.

LXIU-O10 trackway preserved 20 footprints, and tridactyl morphology was observed in 90% of the footprints (Figs. 15, 17). These tracks are functionally tridactyl, with a mean length of 35.9 cm and an average length/width ratio of 1.1. The mean mesaxony is 0.28, and interdigital divarication II–IV is 53°. There is no clear boundary between the heel and the three digits. The digits II and IV are almost equal in length. Each digit trace features a prominent and blunt claw or ungual mark. The average pace angulation of the LXIU-O10 trackway is 155°. The pes traces demonstrate outward and inward rotation, with an average deviation of 8° from the trackway axis.

The LXIN site preserves 48 ornithopod tracks and a number of isolated tracks (Figs. 18, 19; Supplemental Information 3; Table 6), the former including LXIN-O1-O48, ranging in length from 9 to 48.8 cm. The preservation of the ~296 tracks is rather limited, and most of them can only be identified by trackway features. Only a few tracks have clear diagnostic features.

Some of the better preserved tracks and specific trackways are described below (Fig. 18).

LXIN-O1 consists of six tracks, and only LXIN-O1-R3 has identifying features (Fig. 18A). R3 has a length of 33.9 cm, with an L/W ratio of 1.2 and a mesaxony of 0.42. The interdigital divarication of digits II–IV is 50°, and there is a distinct border between the heel and the three digits. The digits II–IV are almost equal in length. Each digit has a

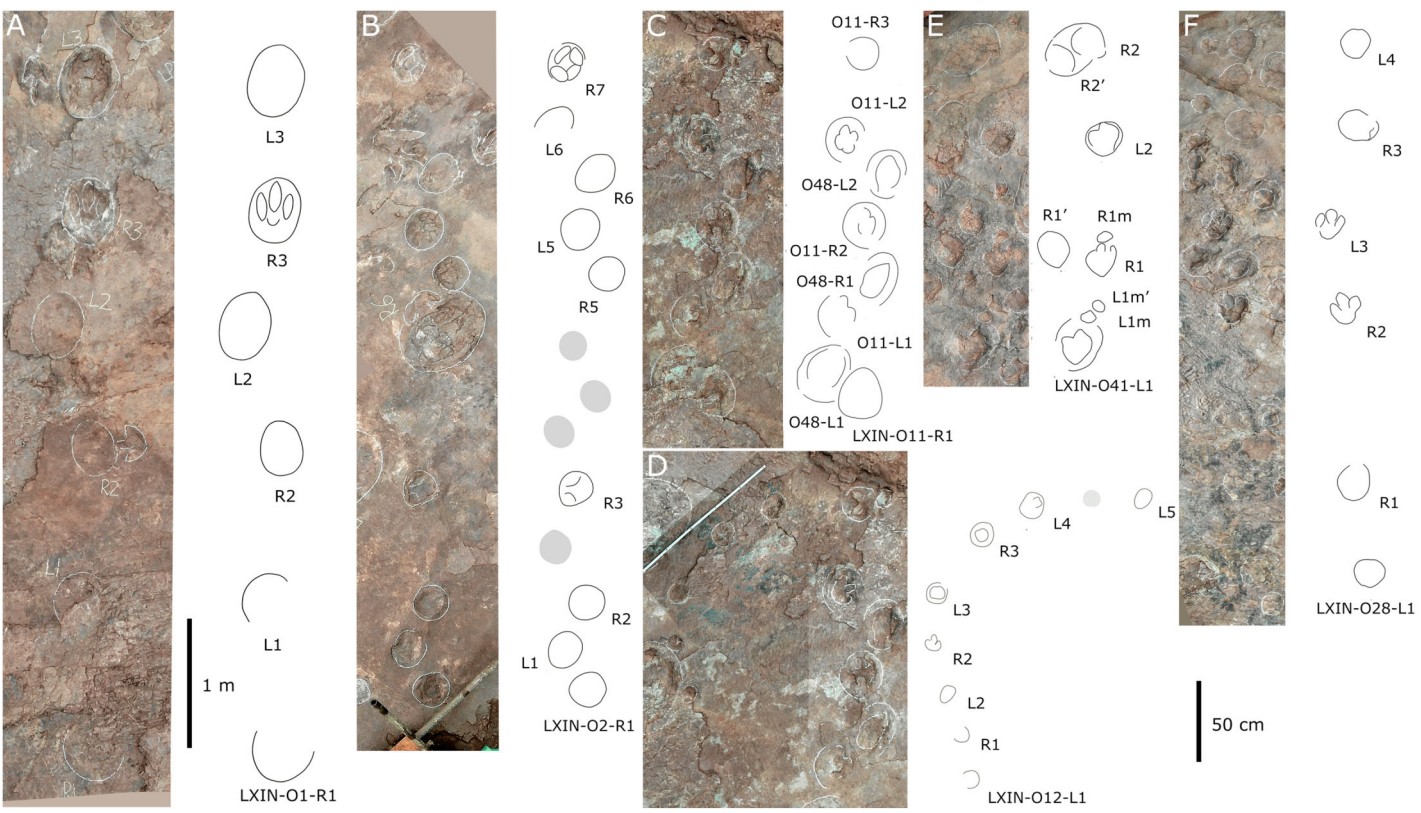

**Figure 18 Photographs and interpretive line drawings of typical *Caririchnium* morphotype trackways from LXIN.** LXIN-O1 (A, left) and the other trackways (B–F, right) share the similar scale bar separately.     

strong and blunt claw or ungual mark. The average angulation of the LXIN-O1 tracks is 167°.

LXIN-O2 consists of nine tracks and only LXIN-O2-R7 has the diagnostic characteristics (Fig. 18B). O2-R7 is 18.7 cm long, with an L/W ratio of 0.8 and a mesaxony of 0.23. The interdigital divarication of digits II-IV is 67° and there is a very clear border between the heel and the three digits. Digits II and IV are almost equal in length, digit III is about 1/2 the length of the two lateral digits. Each digit has a strong and blunt claw or ungual mark. The average pace angulation of LXIN-O2 is 131°, which is the lowest for that track point, which may represent a very slow walking gait.

LXIN-O11 and O48 are very close together and intersect to form a single very short pseudo-trackway (Fig. 18C). Only O11-L2 has some valid features, such as the presence of three relatively well defined toes with a heel. This interlaced trackway requires special care in identification. Overall, the group activity trackways are relatively consistent in their track lengths and step/stride length ratios.

LXIN-O12 is a poorly preserved turning trackway (Fig. 18D), and the trackmaker of this trackway was probably influenced by the large ornithopod LXIN-O13, which began to turn as it approached O13-L1 (detailed figure see Supplemental Information 3). From O12-L3 the trackway turns about 90°. In contrast to the usual decrease in stride length during

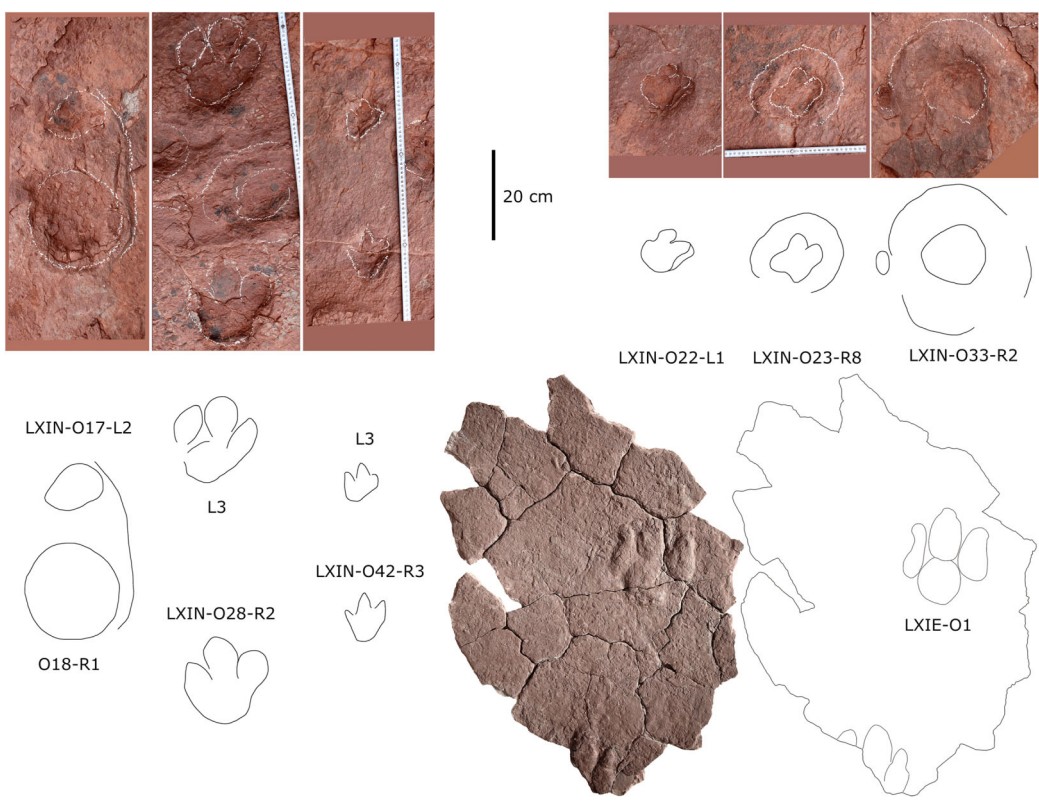

**Figure 19 Close-up photographs and interpretive line drawings of *Caririchnium* morphotype tracks from trackways in LXIN, and an isolated track from LXIE.**

the change of direction, the stride length of LXIN-O12 shows an increasing trend, gradually increasing from 53.7 cm at L1 to 83.6 cm at L3 and decreasing to 68.8 cm at the end of the turn. Due to poor preservation of the tracks, the rotation angles could not be confirmed.

LXIN-O28 contains six tracks forming a discontinuous trackway with one missing in the middle (Fig. 18F). R2 and L2 are fairly well preserved with typical tridactyl morphology (Fig. 19). These two functionally tridactyl tracks with a mean length at 19.3 cm, with the average length/width ratio of 1.0 and the mean mesaxony of 0.30. The interdigital divarication of digits II–IV is 50°. There is a clear boundary between the digits, whereas the boundary between the digits and the heel is blurred. The digits II and IV are almost the same length. Each digit has a strong and blunt claw or ungual mark. The average pace angulation of LXIN-O28 trackway is 149°. The pes traces show an average outward rotation of 16° from the trackway axis.

LXIN-O41 contains eight tracks with an average length of 25 cm, an L/W ratio of 1.2 and a mesaxony of 0.42, with no clear boundary between the heel and the three digits (Fig. 18E). The average pace angulation of LXIN-O41 trackway is 164°. Superimposed tracks are probably present in both the manus track of L1 and the pes tracks of R1 and R2. The superimposed manus tracks were displaced laterally, while the superimposed pes

tracks moved towards the centre of the trackway. Tracks of almost equal size appeared over a distance considerably smaller than the track width. This indicates a high frequency of trampling and may suggest that the trackmaker may have been injured. The latter possibility is more likely than the slipperiness of the ground, as neither LXIN-O41 nor the other nearby trackways showed sliding tracks or metatarsal marks.

Only two well-preserved tracks can be found on the slab from LXIE site (Fig. 19), which are in the similar orientation and forming a short trackway LXIE-O1. In which only O1-R1 is complete, with 21.5 cm in length, 14.7 cm in length, and forming L/W ratio of 1.5. The divarication angle of digit II–IV is 46°, with a mesaxony of 0.27. The incomplete L1 with only the anterior part can compare to R1 in width and mesaxony.

### Comparison and discussion

After the revisions by *Lockley et al. (2014c)* and *Diaz-Martinez et al. (2015)* and *Xing et al. (2024b)*, the number of ichnogenera belonging to Iguanodontipodidae is quite reduced. The most controversial is "Ornithopodichnus" (*Kim et al., 2009*), which was originally defined in Korea and has been used to describe the transversely elongated ornithopod tracks with particularly flattened anterior triangle from NE Asia exclusively. The latest view considered that "Ornithopodichnus" is a *nomen dubium* (*Xing et al., 2024b*), but this does not rule out whether some of the well-preserved tracks from "Ornithopodichnus" can be considered as a valid ichnospecies in *Caririchnium*.

*Caririchnium* (*Leonardi, 1984*) is widely distributed in Berriasian–Albian sediments in Brazil, Portugal, Spain, England, Switzerland, South Korea, Canada, Japan, China, and the USA, with the upper limit of this geological period slightly earlier than the Longxiang age (Cenomanian). The Longxiang specimens fit the diagnosis of *Caririchnium*: pes tracks belonging to Iguanodontipodidae, with a large heel impression that is rounded, centered and wide (wider than the width of the proximal part of the digit III impression); short, wide digit impressions (*Diaz-Martinez et al., 2015*). Longxiang specimens have a wide range of sizes, which is common in the *Caririchnium*. The ratio between the length and width of the "heel" pad varies between ichnospecies. In general, pes tracks are as wide as or wider than long. The size is highly variable, with common pes tracks between 30 and 60 cm long (*Leonardi, 1987*; *Meyer & Thüring, 2003*; *Xing et al., 2007*; *Razzolini et al., 2017*), but there may be smaller or larger examples (*e.g.*, *Lockwood, Lockley & Pond, 2014*; *Xing & Lockley, 2016*; *Xing et al., 2018c*; *Lee et al., 2023*).

*Caririchnium* currently includes *C. magnificum* (*Leonardi, 1984*), *C. kortmeyeri* (*Currie & Sarjeant, 1979*), *C. billsarjeanti* (*Meyer & Thüring, 2003*), *C. lotus* (*Xing et al., 2007, 2015*), and *C. yeongdongensis* (*Kim, Lockley & Chun, 2016*), a total of five ichnospecies. Among these ichnospecies, the morphology of the manus tracks is very distinctive and useful to differentiate between ichnospecies: elliptical, rectangular, and cloverleaf-like subtriangular (*Leonardi, 1987*; *Meyer & Thüring, 2003*; *Xing et al., 2007*; *Kim, Lockley & Chun, 2016*). There is limited information on the morphology of the manus tracks in the Longxiang specimens, but they are essentially elliptical, making them similar to *C. lotus* as a whole (*Xing et al., 2007, 2015*).

The *Caririchnium* lotus in the Longxiang tracksite extends the chronological range of this ichnogenus into the early Late Cretaceous.

The abundance of Longxiang *Caririchnium* illustrates a diverse range of ectomorphological distortion, which is likely caused by the presence of muddy or watery sediments. In this case, only a small proportion of tracks result in distinctly quartered tracks when the substrate is moderately strengthed. Such slippery substrate condition also causes the aformentioned elongated 'heels' in the tracks, and these additional traces can be devided into both wide and narrow patterns, possibly caused by sliding or as metatarsal impressions, as in LXIU-O11 (Fig. 15). Additionally, some manus tracks are situated medially to the pes track, rather than laterally to the midpoint of digits III-IV (*e.g.*, LXIN-O33-R2), which may also be indicative of the influence of a slippery substrate.

It is also noteworthy that some specific behaviour-related features are present, including the occurrence of a large number of tracks in a limited area, forming a pattern similar to that of the manus pes-sets of quadruped trackways (*e.g.*, LXIN-O17-L2 and O18-L1 in Fig. 19; *Stevens, Ernst & Marty, 2016*). Some special trackways can also be observed, including LXIU-O4 (Fig. 16), which features a difference in the size of the right and left pes tracks; LXIU-O7 and LXIN-O12, which exhibit directional changes in the midst (Fig. 16); and LXIN-O41, which contains overlapping tracks that may have been left by injured individuals (Fig. 18E). Meanwhile, in the northwestern part of the LXIN site, the orientation of the small *Caririchnium* trackways is notably distinct, forming an assemblage that extends in an east-west direction. This observation aligns with the commonly documented hypothesis that the smaller trackmakers within this ichnogenus exhibit gregarious behaviour (*e.g.*, *Xing et al., 2015*).

### Trackmaker

The trackmaker of *Caririchinium* is typically referred to the basal ornithopod, especially the basal ankylopollexian or styracosternan, that are comparable in sizes, and shares comparable temporal and spatial interval locally (*Diaz-Martinez et al., 2015*). For example, the early large *Caririchinium* from Upper Jurassic of Iberian Peninsula can be linked to *Oblitosaurus* with 6–7 m in length (*Sánchez-Fenollosa, Verdú & Cobos, 2023*). However, the distribution of ornithopod tracks is much more extensive than their trackmakers (*Noè et al., 2020*; *Sánchez-Fenollosa, Verdú & Cobos, 2023*). For example, there is a paucity of large quadrupedal ornithopod skeletal remains, while large *C. magnificum* have been defined in Brazil (*Bandeira et al., 2024*). Although southeastern China also lacks contemporaneous ankylopollexian trackmakers (*Xu et al., 2018*), however, complete geographic barriers did not exist between South and North China during the Cretaceous theoretically (*Ke & Meng, 2009*). There are large basal ankylopollexian in China that are contemporaneous with or earlier than the Longxiang *Caririchnium*, including *Lanzhousaurus* from Gansu (NW; *You, Ji & Li, 2005*), and *Bayannurosaurus* from Inner Mongolia (NW; *Xu et al., 2018*); basal hadrosaurids including *Bolong* from Liaoning (NE; *Wu & Godefroit, 2012*) and *Equijubus* from Gansu (*You et al., 2003*). At the same time, given that the pes track compared in size to *Camptosaurus* (~5 m) pes is more than 20 cm (~25 cm in estimation) in length, and which to *Brachylophosaurus* (~10 m) pes may reach

**Table 7 Speed estimation (m and km/h) of *Caririchnium* trackways from Longxiang site I.** All data are averaged within precise trackway.

| Small ornithopods (L ≤ 0.25 cm) | | | | | | | | Large ornithopods (L > 0.25 cm) | | | |
|---|---|---|---|---|---|---|---|---|---|---|---|
| Trackway | L | SL/h | Speed | Trackway | L | SL/h | Speed | Trackway | L | SL/h | Speed |
| LXID-O2 | 0.228 | 1.10 | 3.46 | LXIN-O25 | 0.202 | 1.09 | 3.20 | LXID-O1 | 0.331 | 0.95 | 3.64 |
| LXID-O3 | 0.208 | 1.02 | 2.88 | LXIN-O26 | 0.145 | 1.45 | 4.36 | LXIU-O4 | 0.260 | 1.48 | 6.73 |
| LXID-O4 | 0.219 | 1.15 | 3.64 | LXIN-O27 | 0.129 | 1.07 | 2.48 | LXIU-O5 | 0.270 | 1.21 | 4.90 |
| LXIU-O3 | 0.100 | 1.48 | 3.74 | LXIN-O28 | 0.186 | 1.25 | 3.85 | LXIU-O8 | 0.456 | 0.82 | 3.35 |
| LXIU-O6 | 0.111 | 1.50 | 4.03 | LXIN-O29 | 0.090 | 1.02 | 1.91 | LXIU-O10 | 0.390 | 0.80 | 2.95 |
| LXIU-O7 | 0.183 | 1.16 | 3.42 | LXIN-O30 | 0.128 | 1.38 | 3.82 | LXIU-O11 | 0.369 | 0.78 | 2.74 |
| LXIU-O9 | 0.100 | 2.17 | 7.13 | LXIN-O31 | 0.107 | 1.51 | 4.00 | LXIU-O12 | 0.395 | 0.84 | 3.17 |
| LXIU-O13 | 0.122 | 1.04 | 2.30 | LXIN-O32 | 0.182 | 1.15 | 3.31 | LXIN-O1 | 0.463 | 0.77 | 2.99 |
| LXIN-O2 | 0.229 | 0.60 | 1.26 | LXIN-O33 | 0.122 | 1.75 | 5.51 | LXIN-O8 | 0.488 | 0.79 | 3.24 |
| LXIN-O4 | 0.205 | 1.76 | 7.16 | LXIN-O34 | 0.129 | 1.42 | 3.96 | LXIN-O9 | 0.262 | 0.82 | 2.52 |
| LXIN-O5 | 0.246 | 1.65 | 7.06 | LXIN-O35 | 0.141 | 1.18 | 3.06 | LXIN-O13 | 0.421 | 0.82 | 3.20 |
| LXIN-O6 | 0.244 | 2.09 | 10.44 | LXIN-O36 | 0.193 | 0.95 | 2.48 | LXIN-O14 | 0.462 | 0.67 | 2.41 |
| LXIN-O10 | 0.208 | 1.13 | 3.46 | LXIN-O37 | 0.225 | 1.23 | 4.14 | LXIN-O16 | 0.314 | 0.70 | 2.12 |
| LXIN-O11 | 0.223 | 0.97 | 2.77 | LXIN-O39 | 0.158 | 1.23 | 3.49 | LXIN-O20 | 0.256 | 0.88 | 2.81 |
| LXIN-O12 | 0.109 | 1.28 | 3.06 | LXIN-O40 | 0.175 | 1.17 | 3.35 | LXIN-O41 | 0.250 | 0.89 | 2.84 |
| LXIN-O15 | 0.149 | 1.47 | 4.54 | LXIN-O42 | 0.092 | 2.00 | 5.94 | | | | |
| LXIN-O17 | 0.153 | 1.50 | 4.79 | LXIN-O43 | 0.192 | 0.89 | 2.23 | | | | |
| LXIN-O18 | 0.220 | 1.04 | 3.10 | LXIN-O44 | 0.167 | 1.30 | 3.89 | | | | |
| LXIN-O19 | 0.200 | 1.19 | 3.71 | LXIN-O45 | 0.237 | 0.94 | 2.74 | | | | |
| LXIN-O21 | 0.196 | 1.31 | 4.28 | LXIN-O46 | 0.172 | 0.86 | 2.02 | | | | |
| LXIN-O22 | 0.116 | 2.06 | 7.02 | LXIN-O47 | 0.128 | 1.24 | 3.13 | | | | |
| LXIN-O23 | 0.101 | 1.40 | 3.46 | LXIN-O48 | 0.238 | 1.13 | 3.71 | | | | |
| LXIN-O24 | 0.146 | 1.38 | 4.03 | | | | | | | | |

**Note:**
L, Track length (in m); SL/h, the ratio of stride length/estimated hip height.

60 cm in length (*Gierliński & Karol, 2008*; *Paul, 2024*), it can thus be postulated that the large trackmaker of Longxiang *Caririchnium* may be similar in size to the above taxa from lower Cretaceous in China, especially the basal ankylopollexians among them.

### Speed estimates

For estimating hip height (h), *Thulborn (1990)* suggests that for large ornithopods (track length > 25 cm) h = 5.9 × foot length, and for small ornithopods (track length < 25 cm) h = 4.8 × foot length. The velocity of the trackmakers are estimated by the formula of *Alexander (1976)*, and the relative speed (gait) are represented by stride length (SL)/h. For the large Longxiang ornithopods, the relative speed range from 0.67 to 1.48, suggesting a walking gait, and the speed of the trackmakers is estimated to be between 2.12 and 6.73 km/h (Table 7). For the smaller trackmakers, the relative speed range from 0.6 to 2.17, suggesting a walking to slow-running gait, and the speed of the trackmakers is estimated to be between 1.26–10.44 km/h. The evidence suggests that smaller ornithopods display a greater degree of bipedalism in their locomotory behaviour, which is relatively

independent of factors such as body weight and limb proportions. This allows them to achieve higher speeds, as evidenced by the skeletal records (*Maidment et al., 2014*; *Barrett & Maidment, 2017*).

Furthermore, among the Longxiang large *Caririchnium* trackways, three of them indicate a relatively greater speed of slightly over 2 (2.06–2.17), including LXIU-O9, LXIN-O6 and LXIN-O22 (Supplemental Information 3). These constitute 7% of the total 45 large ornithopod trackways. This particular slow-running gait is uncommon among *Caririchnium* and other large ornithopod ichnogenera. On the contrary, two other tracksites that also preserve a substantial number of *Caririchnium* tracks in China, including the Lower Cretaceous Zhaojue site (*Xing et al., 2014a*) and the Mid-Cretaceous Lotus site from Sichuan (*Xing et al., 2015*). In these cases, the related trackways all exhibit a walking gait.

With regard to the track layer of the LXIU site, it can be posited that the relative speed of the trackways with clearly quartered tracks would have been slightly higher than that of the other poorly preserved trackways. This may indicate that substrate condition (*e.g.*, water content) affected the speed of the trackmakers, as previously mentioned. However, on the track layer of the LXIU site, the majority of tracks are poorly preserved, and the trackmakers can still reach slow-running gait, thereby indicating that there is no prominent positive correlation between the speed of the trackmakers and the moderately featured or hardened substrate.

### Didactyl tracks

The LXIU-T1 and T3 trackways in the LXIU site are didactyl trackways, which are typically interpreted as deinonychosaurian trackways (Fig. 20).

The average length of LXIU-T1 is approximately 11 cm, and it can be classified within the ichnogenus *Velociraptorichnus* (*Niu & Xing, 2023*). The T3 tracks are markedly larger, measuring ~36 cm in length. They have been classified as a new ichnotaxon, *Fujianipus*, and represent the largest troodontid deinonychosaurian track documented to date (*Xing et al., 2024a*). These two trackways will not be discussed further in this article. Some isolated tracks, such as LXIU-TI10 could be compared to LXIU-T1 in track size and the snout, paralleled digit III–IV and the pointed mark of digit II, and can also suggest an *cf. Velociraptorichnus* affinity.

It is interesting to note that, this kind of reliable didactyl morphotype only exist in LXUI site, even the LXI sites share the similar track surface. This phenomenon is difficult to attribute entirely to preservation bias related to the trackmaker's body size. After all, the LXIN site contains a greater number of tiny (<10 cm) ornithopod trackways and also features isolated small tridactyl theropod tracks (grallatorid). In contrast, large-sized tridactyl theropod tracks are noticeably less common at the LXIU site compared to the LXIN site. This may suggest that the activity ranges of the two major types of theropod trackmakers were either non-overlapping or had only limited overlap. At the very least, smaller didactyl theropods may have actively avoided entering the activity range of the numerous larger tridactyl trackmakers with stronger pes.

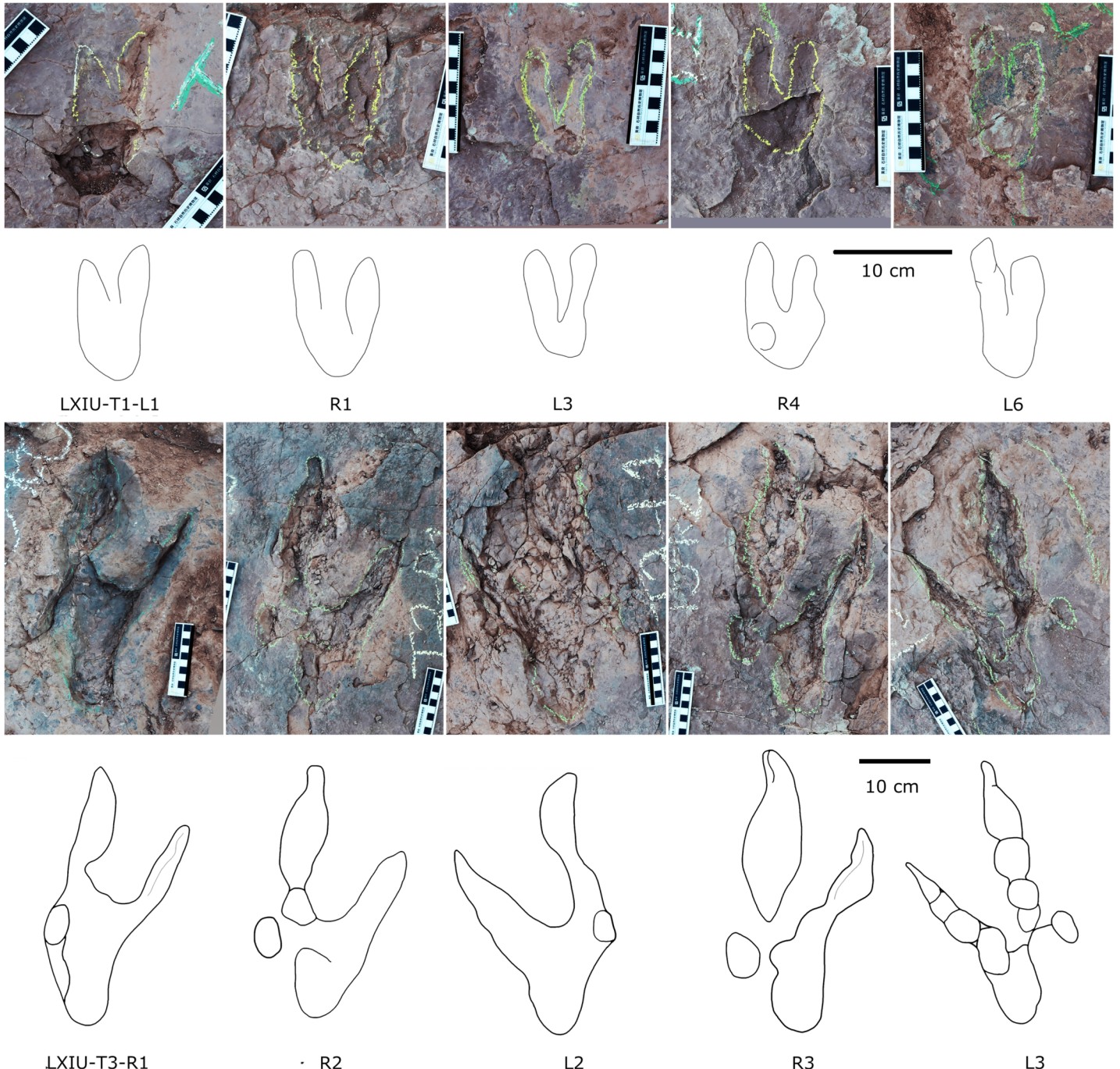

**Figure 20** Close-up photographs and interpretive line drawings of *Velociraptorichnus* and *Fujianipus* trackways from LXIU.

### Possible large didactyl tracks

**Description and comparison.** In the site LXID, the trackway LXID-T2 comprises four consecutive tracks and exhibits a distinctive feature among theropod tracks (Fig. 21; Table 8). The depth of the T2 tracks is notably shallower than that of the majority of tracks

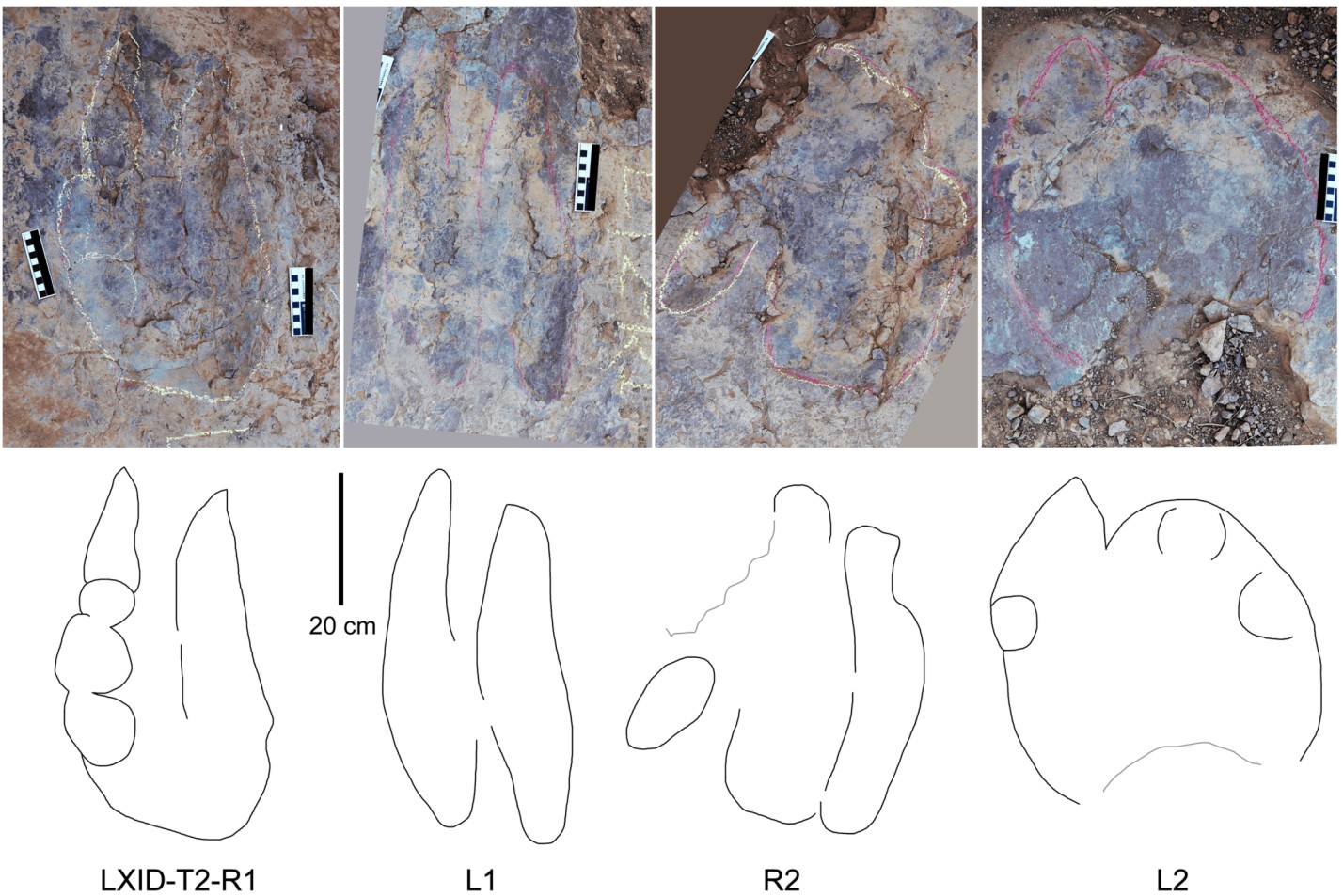

**Figure 21** **Close-up photographs and interpretive line drawings of tracks from possible large didactyl theropod trackway LXID-T2.**

in the same layer. This may indicate that the T2 tracks are likely to be undertracks, yet they still exhibit some discernible morphological and trackway characteristics. The trackway LXID-T2 comprises functionally didactyl, digitigrade pes tracks. The average length of the tracks is 53.4 cm, with an L/W ratio of 3.8. The trace of digit II is either absent or present as an oval imprint proximally. In contrast, the traces of digits III and IV are essentially parallel (with a divarication angle of approximately 15°), and their lengths are similar, resulting in low mesaxony. The anterior margin of each digit trace exhibits sharp claw marks. The tracks exhibit minimal rotation from the trackway axis, with an average pace angulation of 179°.

In the most well-preserved track in the LXID-T2, T2-R1, can be observed that digit III has four phalangeal pads, or three phalangeal pads and one rounded clow mark. The length of digit III is 51.9 cm, while which of digit IV measures 52.1 cm. The proximal ends of digits III and IV exhibit a minimal metatarsophalangeal area. The L1 only preserves the impressions of digit III and digit IV, which have a length of 50.8 and 53.6 cm, respectively. It is possible that the R2 may preserve the impression of digit II, which is ~17.6 cm in

**Table 8 Measurements (in cm and degrees) comparison of possible large didactyl tracks from Longxiang site ID.**

|  | L | L-II | L-III | L-IV | W | W-max III–IV | W-max II–IV | L/W | III–IV | PL | SL | PA | R |
|---|---|---|---|---|---|---|---|---|---|---|---|---|---|
| LXID-T2-R1 | 56.0 | — | 51.9 | 52.1 | 15.8 | 32.9 | — | 3.5 | 17 | 292.2 | 552.3 | 178 | \|3\| |
| LXID-T2-L1 | 56.3 | — | 50.8 | 53.6 | 13.5 | 28.3 | — | 4.2 | 13 | 260.1 | 532.4 | 180 | 6 |
| LXID-T2-R2 | 52.1 | 17.6 | 50.3 | 45.5 | 14.6 | 28.4 | 43.8 | 3.6 | 16 | 272.2 | — | — | — |
| LXID-T2-L2 | 49.1 | — | — | — | — | — | 48.1 | — | — | — | — | — | — |
| Mean | 53.4 | 17.6 | 51.0 | 50.4 | 14.7 | 29.9 | 45.9 | 3.8 | 15 | 274.9 | 542.3 | 179 | \|5\| |

**Note:**

L, Track length; L-II, III and IV, length of digit II, III and IV; W, track width between the claw tip of digit III and IV; W-max III–IV and II–IV, maximum width between digit III–IV and II–IV; L/W, length/width ratio; III–IV, the divarication angle between digit III–IV; PL, pace length; SL, stride length; PA, pace angulation; R, rotation angle; —, no measurement available.

length, and the impressions of digit III and digit IV are ~50.3 and 45.5 cm in length, respectively. The digit II trace is isolated, with the digit II depression situated posteromedial to the margin of digit III. The L2 has only one overall oval impression.

LXID-T2 is comparable to typical deinonychosaur tracks, which are currently classified into six ichnogenera (*Velociraptorichnus, Dromaeopodus, Dromaeosauripus, Menglongipus, Sarmientichnus* and *Fujianipus*) (*Xing et al., 2024a*). The length of 53.4 cm in LXID-T2 is undoubtedly significantly larger than all current ichnogenera and unclassified species. In a recent review, the largest and most conclusive didactyl track was identified as *Fujianipus* (36.4 cm), while the probable isolated didactyl troodontid track from Nanxiong (Guangdong, China) was found to be 39 cm in length. Among the didactyl morphotypes represent deinonychosaurian affinities, it is noteworthy that both the *Fujianipus* and Nanxiong tracks exhibit clear troodontid affinities with relatively short, laterally extended digit IV (*Xing et al., 2024a*). In contrast, the largest record of typical dromaeosaur-type tracks is that of *cf. Dromaeopodus* (29.8 cm) from the Mid Cretaceous, Gulin (Sichuan, China; *Xing et al., 2016*). The R1 and L1 tracks of LXID-T2 exhibit morphological similarities to digit-only *Dromaeosauripus hamanensis* (*Kim et al., 2008*) and *D. jinjuensis* from Korea (*Kim & Lockley, 2012*), with strong, elongated, parallel, and almost equal lengths of digit III and IV.

The insufficiency in the details of LXID-T2 is somewhat reminiscent of specially preserved tracks, such as the possibility as flattened tracks of large *cf. Dromaeopodus* (*Xing et al., 2013*; *Lockley & Xing, 2015*). It is also possible that the vague margins indicate that the tracks may be undertracks. Furthermore, while the LXID-T2 tracks lack sufficient detail to provide definitive morphological traits, their large size is also reminiscent of bipedal ornithopod tracks from the same site. Nevertheless, the LXID-O1 tracks, which are approximately 40 cm in length, do not exhibit the parallel and nearly equal lengths of the digits observed in the LXID-T2 tracks; the almost straight trackway of LXID-T2 tracks (pace angulation = 179°) is also larger than that of the aforementioned LXID-O1 (pace angulation = 161°), which is more akin to a theropod-type trackway.

It is notable that, though lacking the signs of kick-of scours in classic swimming tracks (*McAllister & Kirby, 1998*), the feature of low divarication angle, the lack of heel, and the combination of rounded posterior region and awl-shaped anterior part could also be referred to non-pointed punting-related swimming tracks (*Romilio, Tucker & Salisbury,*

2013; *Navarro-Lorbés et al., 2023*). The absence of digit II and the proximal linkage of digit III and IV may related to the lateral rotation of the hip/pelvic girdle (*Milàn, 2006*; *Tanaka, 2021*). However, like previously discussed in 4.2.3, the complete absence of heel traces (including the posteriorly supporting digit I in the bird-like tracks, see *Tanaka (2021)*) can be hardly interpreted by certain origin, so we merely retain the possibility of a *cf. Dromaeopodus* affinity of LXID-T2.

**Speed estimates.** Under the assumption that LXID-T2 is of deinonychosaurian trackmaker origin, the ratio based on hip-height/foot-length (h/Fl) ratio is 4.32 for dromaeosaurids as proposed by *Tsukiji et al. (2021)*. In contrast, the standard equation for estimating hip height in small/medium theropods is $h = 3.06 \, FL^{1.14}$ (*Weems et al., 2006*). In the case of the LXID-T2 tracks, the estimated average hip height could be 2.3 or 2.85 m, depending on the equation used. *Xing et al. (2024a)* proposed that the length of digit III and deinonychosaurian body length is 5.11% on average, and the body length of LXID-T2 track maker would be estimated as at least 9.98 m.

**Trackmaker.** The estimated size of LXID-T2 trackmaker is not only dramatically larger than which of the troodontid *Fujianipus* (~5 m), but also than the skeletal records of large members from Dromaeosauridae, including *Utahraptor ostrommaysorum* from Utah (*Molina-Pérez et al., 2019*), measuring 4.65 to 6 m in length and 1.5 m in hip height; *Dakotaraptor steini* from Dakota (*DePalma et al., 2015*), measuring 4.35 to 6 m in length and 1.4 m in hip height; *Austroraptor cabazai* from Argentina (*Novas et al., 2009*), measuring up to ~6 and 1.5 m in hip height; and the asian *Achillobator giganticus* from Mongolia (*Perle, 1999*), measuring 3.9 to 5 m in length and 1.25 m in hip height. All the end-members of estimated size are from *Molina-Pérez et al. (2019)* and *Paul (2024)*, the smaller records are from the former; the hip height estimations are from *Molina-Pérez et al. (2019)*. Among them, the overlapping of the temporal range of the Shaxian Formation (*Fujianipus*) and Bayan Shireh Formation (*A. giganticus*) thereby rendering a biogeographic distribution from Mongolia to southeastern China entirely feasible for different clades within deinonychosaurians to become such considerable size (*Xing et al., 2024a*).

The morphological characteristics of LXID-T2 and *Fujianipus* trackmaker are entirely distinct, with the former displaying clear dromaeosaur-like, equal and paralleled digits III and IV. The discovery of additional specimens of the LXID-T2 morphological type would constitute significant evidence of the gigantism of dromaeosaurid deinonychosaurs, whose body size is undoubtedly at the top of the local food chain.

It is important to note that, however, the digit III-body length proportion provided by *Xing et al. (2024a)* does not encompass the length of the drag mark in the average-preserved footprints, which occurs in LXID-T2. However, it should be noted that this is not necessarily representative of the true length of digit III. With regard to track width, LXID-T2 is marginally shorter than that of *Fujianipus* (*Xing et al., 2024a*), and longer than those of large *Dromaeopodus*, which measure ~30 cm in length (*Xing et al., 2016*). Furthermore, the estimation exclusively considered well-preserved small deinonychosaurians (*Xing et al., 2024a*) and lacked a sufficient number of large, complete, and referable deinonychosaurian specimens. In conclusion, when considered as a

deinonychosaurian, the LXID-T2 trackmakers are likely to be smaller than the aforementioned estimation, but still fall within the range of large deinonychosaurian, which may reach 5 to 6 m.

Besides, for the speed of the trackmaker, the relatively short stride lengths of LXID-T2 trackway is indicative of a trotting gait, with SL/h ratios between 1.9 and 2.35, which is consistent with the trackways of *Velociraptorichnus* isp. and *Fujianipus* from the Longxiang site. According to the formula proposed by Alexander, the estimated speeds of the trackways are 13.93 and 17.86 km/h, respectively.

### Dinosaur tracks indeterminate

The LXII site contains three isolated tracks with fairly limited preservation (Fig. 8), with a preservation status of level 0 on the *Belvedere & Farlow (2016)* scale. Of the three isolated tracks, LXII-TI1 is 21.6 cm long with an L/W ratio of 1.4, while the other two tracks are 28.8 and 26.1 cm long with fairly high L/W ratios of 2.0 and 2.3. LXII-TI1 is the first track found in the Longxiang area, and the edge of the track has a distinctive displacement rim. It is morphologically a possible tridactyl track, although only one lateral and one median digit have been preserved. Because of the relatively high L/W ratio, these tracks may have a stronger affinity to theropod tracks.

## DISCUSSION

Due to the formation of southeastern coastal mountains and the correlated volcanism (*Cao, 2018*), northern Guangdong and southern Jiangxi share more similarities in topography and overall climatic change with the Fujian region among SE China during mid-Cretaceous (*Chen, 2000*; *Li et al., 2013*; *Wang et al., 2022*). In contrast, the Zhejiang region, which is relatively close to the northern part of the mountain range, is more similar, at least during the Cenomanian-Turonian period, to the more northerly Jiangsu region, which is not covered by the mountain range (*Yu, 2013*; *Wang et al., 2022*).

In the aforementioned mountainous area, with the exception of the Zhejiang region, which is less affected by the mountains, only Fujian and Jiangxi have dinosaur records in the mid-Cretaceous of SE China, specifically in the Shanghang and Ganzhou basins, respectively (*Xing et al., 2019*, *2024a*; *Niu & Xing, 2023*). However, the fossil record of Jiangxi is and the mid-Cretaceous record is limited to one ankylosaurid, *Datai* (*Xing et al., 2023*), one possible medium-sized tyrannosaurid track (~7.5 m for the trackmaker; *Xing et al., 2019*), and some small-sized tridactyl theropod (?oviraptorosaur) tracks in Ganzhou Basin (LX personal observation). The relatively scarce of Cenomanian-Turonian dinosaur record make Longxiang tracksite in Fujian is an important mid-Cretaceous fossil locality.

### The summary of Longxiang sites ichnofauna

Among the aforementioned morphotypes, small (<40 cm) *Brontopodus*–like tracks from sauropods; the majority of Theropod morphotype A (*Eubrontes* isp.) from large-sized tridactyle theropods, and some smaller, isolated tracks similar in outline (*cf. Eubrontes*) may originate from a small theropod trackmaker with a similar body plan to *Eubrontes*; Theropod morphotype B (*cf. Eubrontes*) very possible from (medium- to) large-sized
tridactyle theropods, and *cf. Dromaeosauripus* and *Velociraptorichnu*s from large/small deinonychosaurian; two types of *Caririchnium* isp. from the medium- to large-sized/ small-sized ("Ornithopodichnus"-like) ornithopods respectively the has been reported by *Niu & Xing (2023)*, while medial- to large sized *Eubrontes* tracks from LXIN, large (>60 cm) *Brontopodus*–like trackway from LXIN, and a possible large didactyl trackway from LXID has been newly distinguished in this article. Furthermore, additional specimens belong to highly digitigrade *cf. Eubrontes.* from LXIN, *Caririchnium* isp. from LXID and LXIN has been discovered on the basis of *Niu & Xing (2023)* (The distribution of size and morphotype see Supplemental Information 4).

Among these tracksites, ornithopod, sauropod and theropod tracks can be identified in LXIII sites, but only LXI sites (main track surface) have recognisable trackways. The tracks of all major dinosaur clades have been found in LXI and LXIII sites, except for the sauropod tracks in LXID. However, within each clade, despite some degree of size variation, the morphotypes covered by numerous trackways and isolated footprints remain relatively uniform. This may be partially attributed to the prevalent deformation observed in the tracks from the Longxiang site, as well as behavioral variations within the same trackway (*e.g.*, transitions between semi-digitigrade and digitigrade postures), which hinder further classification based on finer morphological details. Nevertheless, this also suggests, at least within this region, that aside from the presence of large deinonychosaurians, the trackmakers within each clade were likely morphologically homogeneous overall.

Certain ornithopod tracks classified in *Caririchnium* are the most prevalent in all LXI sites, representing 84% of all trackways and 89%, 57% and 76% of the tracks from LXIN, LXID and LXIU, respectively. Large trackways (≥25 cm) account for 19%, 25% and 62% of the trackways from the above three sites within *Caririchnium* morphotype, respectively, and very large ones reaching ~40 cm occurs in LXIN (5) and LXIU (3) (Table 1). The largest ornithopod trackway in LXIU is even comparable in size to the sauropod trackway in terms of pes length. This suggests that contemporaneous large consumers in the region were probably concentrated in ornithopods, *i.e.*, that the relatively cooler (compared to lowland regions at the same latitude), at least seasonally dry, coastal plateau with mountain ranges acting as barriers during the mid-Cretaceous period in southeast China was suitable for these ankylopollexian survival (*Chen, 2000*; *Li et al., 2009*; *Lü et al., 2019*; *Zhang et al., 2021*). Besides, It should be noted that for the three tracksites in LXI, particularly LXIN and LXIU areas, there is a difference in the size at which the mutation occurs in the ornithopod trackmaker. For the LXIN site, there is a clear gap between 32 and 41 cm, whereas for the LXIU site, this gap occurs between 19 and 26 cm. For the LXIN site, there is a clear gap between 32 and 41 cm, whereas for the LXIU site, this gap occurs between 19 and 26 cm. For the entire LXI ornithopod trackways, track length predominantly falls within the small to medium size range, with peaks observed at 10–15 cm and 20–25 cm, while a few tracks exceed 40 cm in length (Fig. 22A). In terms of track width, a unimodal distribution is evident, centering around 15–20 cm, with a noticeable trough in the 25–30 cm range (Fig. 22B). Based on statistical comparisons and morphological differences across different areas, the observed statistical discrepancy between track length and width likely results

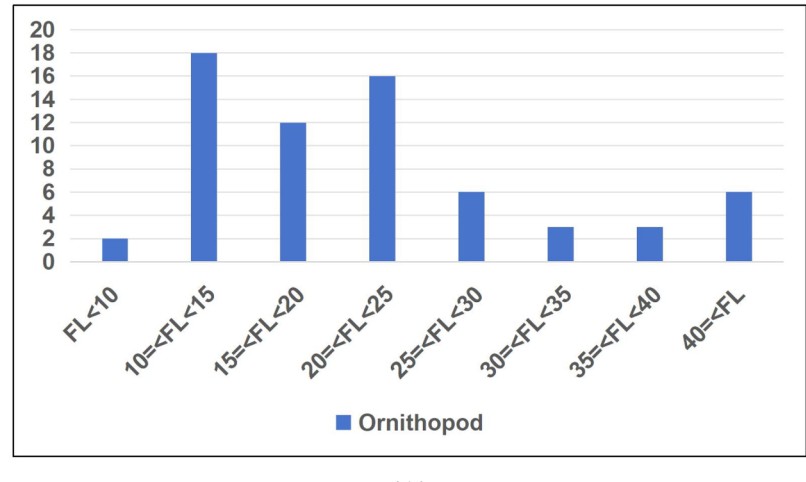

(A)

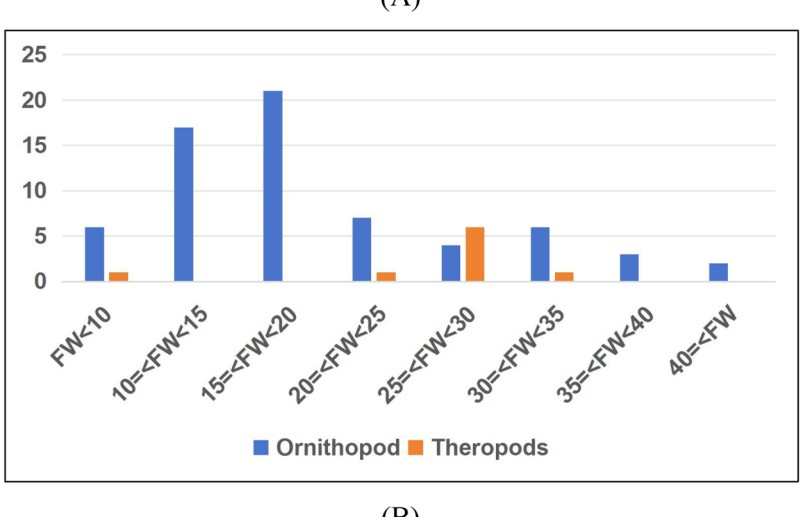

(B)

**Figure 22 Size frequency of tridactyl trackways from LXIs sites. Only the average length is considered.** (A) Considering track length, only the ornithopod trackways exhibit relatively lower axial deformation are included; (B) Considering track width, both theropod and ornithopod trackways are included.

from variations in deformation patterns across different regions. Therefore, small- to medium-sized ornithopod tracks are considered as a single category in this study.

Given that juveniles of large animals can occupy the niche of smaller ones (*Lockley & Xing, 2021*; *Schroeder, Lyons & Smith, 2021*; *Wyenberg-Henzler, Patterson & Mallon, 2022*), it can be posited that the discrepancy of the two track layers may be due to differences in the ontogenetic stage, rather than the variation in clade.

## The regional comparison of mid-Cretaceous dinosaur fauna in SE China

The richest and the most continuous record of the mid-Cretaceous dinosaur fauna in SE China is found in Zhejiang (Fig. 3; *Yu, 2013*; *Xi et al., 2019*), and the timing of its faunal

changes aligns with the sedimentary environment shifts reflected in the palynological records (*Yu, 2013*).

In Zhejiang region, the skeletal record from the Early Cretaceous is sparse, consisting of only a few small to medium-sized hadrosaurioids and poorly preserved ankylosaurians. However, by the onset or early stages of the Cenomanian-Turonian, with a relative decrease in temperature and an increase in precipitation, a diverse assemblage of herbivorous dinosaurs began to appear in Zhejiang. Sauropods and possible therizinosaurids were the first to emerge in the region (*Qian et al., 2012*). In the later stages, despite a slight temperature rise, continued abundant precipitation was accompanied by a more diverse record of titanosauriforms and titanosaurs (*Tang et al., 2001*; *Lü et al., 2008*), as well as medium-sized pterosaurs and small turtle tracks (*Wu et al., 2018*). Notably, during Cenomanian-Turonian, the only suspected ornithischian record consists of two poorly detailed large ornithopod trackways (*Wu et al., 2018*), and ankylosaurian remains—present in earlier and later intervals—have yet to be identified.

In the post-Turonian, Zhejiang and the neighboring Jiangsu region experienced further temporary warming and prolonged aridification. During this phase, sauropod skeletal and track remains disappeared entirely, replaced by a resurgence of ornithischians. This includes abundant ankylosaurians (*Lü et al., 2007*; *Yu, 2013*; *Zheng et al., 2018*), small basal ornithischians (*Zheng et al., 2012*), and possible small hadrosauroids. Among theropods, therizinosaurians and avians appear in the skeletal record, and fragmentary remains indicate the presence of both carnosaurs and coelurosaurians across multiple basins (*Yu, 2013*). Pterosaurs from this interval are also known from skeletal fossils (*Unwin & Lü, 1997*). Compared to the relatively scarce track fossils, post-Turonian strata contain egg fossils attributed to Macroelongatoolithus (up to 43 cm in length), Longatoolithus (10–17 cm in length), and Prismatoolithidae (size unknown) (*Yu, 2013*). These oofamily records suggest that a diverse assemblage of maniraptorans, including troodontids, flourished in the region as arid conditions intensified.

The fossil record from the Zhejiang region suggests that, despite being part of the same southeastern coastal mountain system, the Shanghang site—located further south and more strongly influenced by orographic barriers—did not experience a temporary dominance of sauropods during the Cenomanian-Turonian. Furthermore, it lacks a long-term ankylosaurian presence, which is consistently recorded in other stages of Zhejiang's fossil record. Notably, ankylosaurians also appeared in the adjacent Ganzhou Basin during the Turonian–early Coniacian (*Xing et al., 2023*).

In comparison to other dinosaurs, including other ornithischians, ornithopods demonstrate a more pronounced and efficient high-fibre herbivory (*Button et al., 2023*). Additionally, the dental texture of this clade has been found to be significantly rougher in the Late Cretaceous period (*Kubo et al., 2023*). In the case of hadrosaurids, representatives of the ankylopollexia clade, which underwent distinct gigantism, the auxiliary abrasive capacity of their teeth has been further enhanced (*Kubo et al., 2023*). Concurrently, a comparable herbivorous homogenisation is observed in the latest Cretaceous global cooling process. During this process, hadrosaurids with more advantageous border herbivorous diets and larger feeding ranges occupied at least the ecological space of

ankylosaurs and ceratopsians (*Condamine et al., 2021*). These may be contributing factors to the distinctive domination of large-sized ornithopod tracks at Longxiang sites in the Fujian region relative to Zhejiang, as the former experienced a more gradual climate change (cooling and changes in precipitation levels) during this period.

Another noteworthy feature of the Longxiang tracksite in southeastern China is its distinctive large didactyl track, which includes the *Fujianipus*. For the large deinonychosaurians record to date, with the exception of *Utahraptor* from the Hell Creek Formation (*Pearson et al., 2002*; *DePalma et al., 2015*), the remaining fauna lacks large tyrannosaurid or abelisaurid predators, which large deinonychosaurian all occur as currently discovered apex predators of contemporary terrestrial fauna (*Novas et al., 2009*; *Tsuihiji et al., 2012*; *Kirkland et al., 2016*; *Rolando et al., 2021*; *Paul, 2024*). Concurrently, the associated fauna also exhibits a paucity of large marginocephalians and giant (>15 m) macronarians (*Paul, 2024*).

In terms of chronology and palaeogeography, the ichnofauna from Fujian is more closely related to the contemporaneous Bayan Shireh skeletal fauna from Mongolia. However, Longxiang site does not contain definitive large tridactyl theropod tracks as which from the pre-Albian or pre-CenomanianYanguoxia tracksite in NW China (>40 cm; *Xing et al., 2018c*, unpublished manuscript), which posited between Bayan Shireh and Longxiang sites. This may indicate that large deinonychosaurians dispersed in East Asia during the latest Early Cretaceous period, or that there was convergence in response to similar environments, including the partly comparable climate conditions and the lack of pre-existing large apex predators, such as tyrannosaurids (*Brusatte, Benson & Xu, 2010*; *Xing et al., 2024a*). The paucity of definite large theropod record except deinonychosaurians may be attributed to similar factors to those which led to the domination of ornithopods compared to Zhejiang fauna.

The absence of ankylosaurians at the Longxiang site may be attributed to the relative scarcity of contemporaneous tracksites in Fujian, leading to a preservation bias. However, the distinct proportions of sauropods and ornithischians—particularly ornithopods, and the early presence of large deinonychosaurian, likely reflect a fundamental difference in the Cenomanian-Turonian faunal composition between adjacent regions within SE China. This suggests that regional signal variations, rather than globally comparable climatic or ecological factors, may have had a greater influence on both the composition of local faunas in SE China and the preservation of their fossil record. Specifically, this influence appears to stem from the region's topography, with mountainous terrain playing a role in moderating temperature fluctuations (or creating refugia with a broader thermal range; *Antonelli et al., 2018*) and reducing peak precipitation levels. A detailed discussion of these effects is beyond the scope of this study and warrants further investigation in future research.

## Brief comparison with other mid-Cretaceous low-latitude dinosaur fauna and the environmental influence

Globally, the mid-Cretaceous period coincided with two oceanic anoxic events (OAE1 and OAE2), occurring during the Aptian–Albian and Cenomanian–Turonian periods,

respectively (*Schlanger & Jenkyns, 1976*). During OAE2, average global temperatures reached their peak for the entire Cretaceous period (*Forster et al., 2007*; *Jenkyns, 2010*), and this interval also witnessed the Plenus Cold Event, a significant cooling episode evident in globally comparable carbon isotope records (*Forster et al., 2007*; *O'Connor et al., 2020*). During this period of extreme heat and climatic instability, dinosaur faunas were still widely recorded across low-latitude regions. In particular, extensive records exist from the northern Gondwanan landmasses, which dominated the continental deposits of these latitudes at the time. Notable examples include the Kem Kem Group in Morocco (*Mannion & Barrett, 2013*; *Belvedere et al., 2013*), the Candeleros Formation in Neuquén Basin of Argentina (lower Neuquén Group; *Coria & Salgado, 2005*; *Tomaselli, David & González Riga, 2021*), and the lower Bauru Group in southern Brazil (*Navarro et al., 2025*).

Beyond the Longxiang site, contemporaneous representative dinosaur-bearing formations from Laurasia include the Dakota Formation (*Lockley et al., 1992*; *Lim, Kim & Gierlinski, 2014*) and the Woodbine Group (*Noto et al., 2022*) in the eastern and southern United States, and the Cenomanian–Turonian deposits of the Iberian Peninsula (*Csiki-Sava et al., 2015*). However, most dinosaur faunas from North America, Europe, and Africa during this interval are derived from coastal and island ecosystems that expanded due to the rising sea levels, whereas some of the South American records originate from fully continental deposits, making them more directly comparable to those of China.

The most noticeable difference between the South America and East Asia during mid-Cretaceous lies in the fundamental composition of their Mesozoic faunas. In South America, which was located along the southwestern margin of Gondwana and had already separated from Gondwana in its southern part by the Late Cretaceous, small notosuchian mesoeucrocodylians, which first appeared before the late Early Cretaceous and became highly diverse after OAE2, played a significant role in the ecosystem (*Pol & Leardi, 2015*). In contrast, in East Asia, small dinosaur still occupied the similar niche occupied by notosuchian, with oviraptorosaurian theropods represented as prosperous herbivores and omnivores (*Lü et al., 2016*). Additionally, in terms of traditional herbivores, South America maintained a high degree of sauropod diversity (*Tomaselli, David & González Riga, 2021*), while its ornithopods evolved towards more agile forms, leading to the emergence of a diverse assemblage of elasmarian ornithopods (*Bandeira et al., 2024*). Meanwhile, in East Asia, a substantial radiation of large hadrosaurioids had already begun in the Early Cretaceous (*Xu et al., 2018*).

Additionally, there are significant differences between the two regions in terms of altitude, land-sea positioning, and relative temperature. During the mid-Cretaceous, particularly the Cenomanian-Turonian, the Fujian region (SE China), where the Longxiang site is located, was part of the previously mentioned linear mountain chains associated with an Andean-type subduction system. The overall regional temperature was notably lower than that of the mid-latitude hotspot regions in central South America (*Scotese et al., 2025*). However, despite differences in geological origins, both regions were generally arid (*Scotese et al., 2025*). It is important to note that, unlike South America—situated along the newly formed Atlantic Ocean with limited evaporative moisture supply (*Stommel, 1957*; *Carvalho & Leonardi, 2024*)—the aridity in part of the SE China was

largely influenced by the aforementioned orographic barriers that restricted moisture transport. This likely resulted in variations in precipitation across different intermontane basins in Zhejiang and Fujian, depending on altitude and proximity to the ocean during the mid-Cretaceous. Locally sufficient precipitation (*Wang et al., 2022*), combined with montane environments that could provide relatively cooler habitats within small areas (*Antonelli et al., 2018*), may have facilitated the coexistence of morphologically competitive taxa capable of adapting to a wider range of ecological conditions.

Returning to the discussion of faunal changes across different environments, the mid-Cretaceous terrestrial fossil records from South America, represented by the Bauru Group in Brazil and the Neuquén Basin in Argentina, reveal a strongly regionalized faunal turnover process during the Cenomanian-Turonian interval, which appears to be correlated with paleolatitude (*Coria & Salgado, 2005*; *Navarro et al., 2025*). At least part of these turnovers may reflect the impact of global climatic perturbations during OAE2, such as the temporally corresponding Plenus Cold Event (PCE) (*Navarro et al., 2025*). However, within the sedimentary terrestrial arc-related basins of SE China, current palynological evidence suggests that only the temperature variation records from the Jiangsu-Zhejiang region (*e.g.*, *Wang et al., 2022*), located beyond the northern boundary of the mountain range, can be correlated with OAE2-related marine events.

Besides, from the perspective of dinosaur fauna, the primary herbivorous taxa in the Jiangsu-Zhejiang region differ significantly from those reported at the Longxiang site in Fujian. Given that skeletal fossils from Zhejiang prior to the Cenomanian, though limited, still include records of small- to medium-sized ornithopods—comparable to those at the Longxiang site—it can be hypothesized that the faunal composition within SE China was relatively consistent before the Cenomanian. If this assumption holds, the Longxiang site represents a record that did not undergo a fundamental faunal turnover. While in the Zhejiang region, faunal transitions during relatively extreme climatic periods were temporary, and groups that had not completely migrated out of the area would return relatively quickly when climatic conditions became favorable for their activity.

Moreover, although precise geochronological data for Cretaceous terrestrial strata in China remain scarce, some patterns can still be observed. At the Cenomanian-Turonian boundary, the temperature variation trends inferred from palynoflora appear to be the exact opposite of the overall marine temperature changes associated with OAE2 (*Forster et al., 2007*; *Yu, 2013*; *Wang et al., 2022*), despite the expected increase in precipitation. Given the currently low precision of the available data, it can only be speculated that these environmental and faunal differences, if driven by changes in marine conditions, might result from a delayed response to OAE2 or an amplification of the PCE due to certain regional factors.

The example from the coastal region in SE China provides partial insight into how the orographic barrier effect of local mountain ranges may have partly reshaped basin conditions on the leeward side, potentially overriding the influence of global marine-driven events. This hypothesis requires further validation through high-resolution paleoenvironmental reconstructions based on palynological and isotopic evidence across the various arc-related basins of SE China, as discussed earlier.

## CONCLUSIONS

New material of dinosaur tracks and trackways form Longxing (LX) tracksites in mid-Cretaceous of Shanghang Basin, Fujian, China are reported in this article on the basis of earlier researches of the similar site. New medial- to large-sized theropod tracks with diverse ectomorphological variations from LXIN site, large (>60 cm) *Brontopodus*–like trackway from LXIN site, and a possible large didactyl trackway from LXID site has been newly distinguished, and *Caririchnium* isp. from LXID, LXIN and LXIE has been newly discovered.

The Longxing tracksite is confirmed to be dominated by ankylopollexian ornithopod trackways (>84%), with a proportion of large members within ornithopod (>27% over 25 cm in length), and indicates the flourishing of large ornithopods among the herbivores occurring within this period. The Longxiang sites also demonstrate the presence of concurrent, distinct, large didactyl tracks, which diverge from the dominated apex predators in most late Cretaceous fauna.

The Longxiang tracksite represents the only extensive mid-Cretaceous dinosaur tracksite in southeastern China, providing a temporal comparison with the mid-Cretaceous skeletal record of Zhejiang (SE China) and Mongolia in East Asia. Compared to the Zhejiang region, which belongs to the southeastern coast mountains and exhibits a continuous faunal succession, the Longxiang site not only differs significantly in the richness of its fossil record but also in its herbivorous composition. The herbivore assemblage at Longxiang appears more closely aligned with pre-Cenomanian faunas of Zhejiang that before significant climatic changes.

Globally, among contemporaneous dinosaur faunas, the terrestrial assemblages of central South America provide a more suitable comparison to those of southeastern China. Based on current fossil evidence and associated chronological data, faunal turnover in both the Longxiang site and Zhejiang appears to have been relatively mild, with faunal compositions recovering rapidly when climatic conditions reverted. Furthermore, the observed changes in faunal composition and timing do not align directly with the OAE2. Instead, they may correspond to a mid-OAE2 cooling event followed by the PCE warming phase, lacking records of the initial warming phase.

These findings suggest that the mid-Cretaceous faunas of southeastern China and the preservation of their fossil records were shaped primarily by regional factors—such as topographic barriers created by mountainous landscapes, the correlated diverse habitats, and volcanic activity in adjacent intervals—rather than by related to changes in marine conditions, globally climatic or ecological drivers. Further research is necessary to refine the understanding of these regional influences, and improve the temporal framework of faunal changes in SE China.

## ABBREVIATIONS OF MEASUREMENTS

| | |
|---|---|
| **L** | Track length |
| **W** | Track width |
| **PL** | Pace length |

| L/W | Length/width ratio |
| II–III | The divarication angle between digit II and III |
| III–IV | The divarication angle between digit III and IV |
| II–IV | The divarication angle between digit II and IV |
| PL | Pace length |
| SL | Stride length |
| PA | Pace angulation |
| R | Rotation of tracks (outward rotation as positive) |
| H | Heteropody (=the ratio of the manus/pes track area) |
| P'ML | The distance between the paired pes and manus track |
| WAP | Width of angulation pattern (=Gauge width) |
| WAP/P'ML | The ratio of WAP and P'ML |

## ACKNOWLEDGEMENTS

We thank Ismar Carvalho and two anonymous reviewers for their constructive comments and proposals for improvements.

### Funding

This research was funded by the "Deep-time Digital Earth" Science and Technology Leading Talents Team Funds for the Central Universities for the Frontiers Science Center for Deep-time Digital Earth, China University of Geosciences (Beijing) (Fundamental Research Funds for the Central Universities; grant number: 2652023001), the 111 project (B20011), and Fundamental Research Funds for Central Universities (265QZ201903). The funders had no role in study design, data collection and analysis, decision to publish, or preparation of the manuscript.

### Grant Disclosures

The following grant information was disclosed by the authors:
"Deep-time Digital Earth" Science and Technology Leading Talents Team Funds for the Central Universities for the Frontiers Science Center for Deep-time Digital Earth.
China University of Geosciences (Beijing).
Fundamental Research Funds for the Central Universities: 2652023001.
111 Project: B20011.
Fundamental Research Funds for Central Universities: 265QZ201903.

### Competing Interests

The authors declare that they have no competing interests.

## Author Contributions

- Lida Xing conceived and designed the experiments, performed the experiments, analyzed the data, prepared figures and/or tables, authored or reviewed drafts of the article, and approved the final draft.
- Kecheng Niu conceived and designed the experiments, performed the experiments, authored or reviewed drafts of the article, and approved the final draft.
- Qiyan Chen analyzed the data, prepared figures and/or tables, authored or reviewed drafts of the article, and approved the final draft.
- Hendrik Klein analyzed the data, authored or reviewed drafts of the article, and approved the final draft.
- Anthony Romilio performed the experiments, analyzed the data, prepared figures and/or tables, and approved the final draft.
- Runsheng Chen performed the experiments, authored or reviewed drafts of the article, and approved the final draft.
- Min Lin performed the experiments, authored or reviewed drafts of the article, and approved the final draft.
- Ke Deng performed the experiments, authored or reviewed drafts of the article, and approved the final draft.
- Jianrong Tang performed the experiments, authored or reviewed drafts of the article, and approved the final draft.

## Data Availability

Raw data are available at Zenodo:

Xing, L., Niu, K., & Chen, Q. (2025). The supplementary information 1 to 4 of "Dinosaur track assemblages from mid-Cretaceous of Fujian Province, southeastern China: ichnotaxonomy and faunal comparison" [Data set]. In Dinosaur track assemblages from mid-Cretaceous of Fujian Province, southeastern China: ichnotaxonomic review and faunal comparison. Zenodo. https://doi.org/10.5281/zenodo.15544368.

## Supplemental Information

Supplemental information for this article can be found online at http://dx.doi.org/10.7717/peerj.19597#supplemental-information.

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
