# Peer review of "Dinosaur track assemblages from mid-Cretaceous of Fujian Province, southeastern China: ichnotaxonomic review and faunal comparison"

_PeerJ, doi:10.7717/peerj.19597_

## Round 0.1 · original submission · Major Revisions

Please, take into account all reviewers' comments and address them. There are some serious issues especially regarding geological information.

Reviewer 1 ·

Basic reporting

The manuscript results very difficult to follow. Many problems with the language exists, particularly referred to the geological background. The ichnotaxonomomic approach is unusual, amendments are not in formal way and formal treatment of morphotypes is unnecessary, as a non-formal category in the ICZN.
The abstract do not reflect the manuscript content and all the climatic interpretation is pointed out in excess.
The total absence of sedimentological treatment of lithologies is with mistakes layer or bed expression of grain size etc. The surfaces description and details about the substrate is lacking. Moreover, some figures, contain drawings that are not explained on the left side. 50 figures (!) is an exaggeration. Why photogrammetry is done just in partial?.

The reference list is incomplete, but strangely the two mention about South America are lacking (Gonzalez Riga and Leonardi), and also any reference about the rich ichnofauna of the Neuquén Basin is mentioned, even when share the same time range.
The manuscript contains lots of unbalanced information, and it is not clear for a researcher not familiar with the Chinese names to follow the plethora of names. In my opinion the organization, structure, sedimentology and separation between diagnosis from comparison (e.g. 518) is mandatory before consider any possibility of publication.
I have no opinion about comparisons, but also I do not understand why some names are in different font or no italics.

Sincerely

Experimental design

The aothors have to explain why use one or other way of analysis. Then sizes with photogrametry but wihout reference ot substrate features or even potentian undertrack levels is not discussed at all.

Validity of the findings

The opinions about the flaws make me consider that conclusions are not completely validated

·

Basic reporting

It is an important study concernig the China's ichnofaunas. The description of the ichnosites and ichnofossil are clear and unambiguous. It is only necessary small changes in some word spellings.
Literature references are extensive, but very restricted to the material from China and North America. I consider important, specially concerning the definition of the new ichnospecies a more wide evaluation of other ichnospecies defined in South America (Brazil and Argentina), Africa and Europe. Concerning the sauropods it is very important to compare with the tracks from Bolivia (Toro-Toro) and Brazil (Rio do Peixe Basins).
The manuscript present a professional structure. Figures and tables are of good quality and easy readible. It is important a more detail information in captions, specially to allow the identification of the new ichnospecies

Experimental design

The study is original within aims and scope of the journal, and the objectives of the study are well defined. The manuscript shows a rigorous investigation and high technical and ethical standard
Concerning the methods I suggest the inclusion of more recent studies of Falkingham concerning procedures to investigate in ichnonology.
The evalutation of the new ichnospecies should introduce more information concerning why it is a distinct ichnospecies and not only a morphotype related to the substrate conditions.

Validity of the findings

The findings are validity and novel. All the information concerning the changes and questions concerning methods and necessity of comparions are included in the revised PDF file

Additional comments

It is an important study. There are some minor changes in the text, but I vividly suggest some comparisons with other ichnofaunas from South America, Africa and Europe. There is a great tendency to restrict the information and comparisions with materials described from the USA - it is important to remind that dinosaurs walked around all the Mesozoic world and it is not an exclusivity of the americans researches to describe their footprints - the chinese studies on them are a prove of this!

Reviewer 3 ·

Basic reporting

This manuscript describes the dinosaurian ichnofauna of a series of large Cretaceous trackways from the Longxian region of China. The authors have provided an enormous amount of data and a rather dense text to describe these trackways, so much so that perhaps this contribution would have been better as a series of papers. There is extensive information and dozens of figures documenting tracks of sauropod, theropod, and ornithopod dinosaurs. Much of this is in close to publishable form, but there are also several opportunities for improvement, some of which are essential.

At present, the “Introduction” is actually part of the “geological setting,” and is fairly technical at that. This paper needs a proper introduction that states what the paper is about and which answers the classic “journalistic questions” of who, what, when, where, and why. Who found/is studying the tracks, where are they, when were they deposited, where are they, and how/why are they significant? This needs to be much more concise than the present “introduction.”

The “Methods” section would benefit greatly from a paragraph defining the many abbreviations used in the text. I know that many of these are standard to ichnologists, but in a text this extensive having one place where all the abbreviations are defined is extremely helpful.

The Supplementary Information is doubtless a gold mine for many ichnologists, but some tables in the primary text summarizing key aspects of that data would be advantageous. For an example, see lines 179-193—much of that information would be more easily retrieved from a table that distilled the key details.

I think a table of the speed estimates obtained from various ichnotaxa would be helpful.

Regarding the previous two points—the more of the data the authors can put into tabular format or figures, the better chance it has of getting cited going forward.

Take the discussion and conclusions as an example—this is a very dense text with a lot of ichnotaxonomy, lists of myriad stratigraphic units in disparate basins, all across a significant chunk of Cretaceous time. All of this would be a lot easier to digest if there were a couple figures—one a map showing the distribution of these data in space and another chart showing the ages of different units in time.

The heading at line 1017 of “Speed estimates and trackmaker” should (1) Be changed to “Trackmaker and speed estimates” to reflect the order of the following text; and (2) probably ought to be two distinct section.

Figures:
The figures are a real strength of this paper, but the captions are often insufficient. If the trackway is identified to an ichnogenus in the text, please match that ichnotaxonomy in the caption, especially in the case of the new ichnospeices (e.g., Figs. 18-20, T. longyanensis morphotype A and 21-22, morphotype B).
Just generally throughout the manuscript I feel that more call-outs would help guide the reader to the key footprints and observations.

Figure 2. “Not exposed” rather than “unfound,” where is this ~10 m section with int eh formation?
Figure 7. Very difficult to see the subject
Figure 8. More space than drawing, combine with another?
No Figure 10 or Figure 11 included in the review copy
Figure 15—Confusing —when is R1?
Figure 17. Not really used
Figure 23, while clearly informative and one of the more technologically advanced figures in the paper, is only called out 3 times, seems out of place, and its caption is inadequate. I would consider moving it up in the manuscript with the more overview figures of the tracksties.

Experimental design

This paper generally follows accepted norms of practice in the description and identification of dinosaur tracks, and does a commendable job of illustrating an enormous number of those track.

As mentioned previously, a better introduction would help make this more clear.

Validity of the findings

Generally, I think that the findings are accurate and defensible. For those not familiar with "middle" Cretaceous tracks and/or the Cretaceous paleontology of China, however, I think that they could be explicated with some summary figures and/or tables, as suggested earlier in the review. It is an enormous (monograph-length) manuscript, and the discussion and conclusions are both examples of where a figure might do a better job of explaining the results/implications/etc., than a thousand words.

One point of concern that I have is that it is unclear why the tracks of Velociraptorichnus merit 2 figures (24-25) but only 7 lines of text. I think that, unless previously published, they deserve the depth of description and comparison that the other tracks receive.

I paste the following in here again as an example of how I think the paper could be improved in this aspect:
Take the discussion and conclusions as an example—this is a very dense text with a lot of ichnotaxonomy, lists of myriad stratigraphic units in disparate basins, all across a significant chunk of Cretaceous time. All of this would be a lot easier to digest if there were a couple figures—one a map showing the distribution of these data in space and another chart showing the ages of different units in time.

The heading at line 1017 of “Speed estimates and trackmaker” should (1) Be changed to “Trackmaker and speed estimates” to reflect the order of the following text; and (2) probably ought to be two distinct section.

Additional comments

Miscellaneous comments on the text (numbers refer to lines of the review pdf):

The paper would benefit from a thorough proofreading; while generally acceptable there’s some issues with the English; a variety of issues that require redress are listed here, but there’s many other, more trivial word choice issues (I may scan the ms).

I made no effort to check the completeness of bibliography.

40: There are no grasslands in the Cretaceous.
78: What are “red basins”?
93: “hot, dry” not “dry hot”
114: What are “planting pits”?
116: I believe that the “54 tracks” are actually trackways.
141: “entire” is a better word than “whole”
151: “centering” not “centring”
246: Another abbreviation (WAP) introduced
259: Missing italics; also it should be Parabrontopodus
260: and thereafter: Italicize Polonyx
321: “without influenced by” is incorrect
345: “attributed to”
358 and thereafter: “trotting” is misspelled throughout.
361 and thereafter: I would write out hippopotamus/hippopotomi and rhinoceros/rhinoceroses rather than “hippos” and “rhinos”
400-401: Very awkward wording, hard to tell what this is
420: Again, no Figure 10 provided
509: Please call out all figures with the new ichnospecies (T. longyanensis).
683: Excellent example of a paragraph where more call-outs to the figures would be appreciated.
698-699: “Clear troodontid affinities” and 744-745 “clear dromaeosaur-liek traits”—Both of these merit further discussion/amplification for the non-specialist.
765: ornithopod, not ornithopods
798: This is unclear
810: Sentence is repeated
817, 841: Is “ectomorphological” the same as “extramorphological” variation sensu Peabody?
857: Should call out figure 39.
973: What is/are the ichnospecies of Ornithopodichnus? This paragraph has minimal value to a non-ichnologist without greater context.
979: Centered
1032, 1087, and anywhere else: Please delete “etc” as this is not acceptable text.

---

## Round 0.2 · Minor Revisions

Please, look at the minor comments and suggestions of the reviewer and implement them.

·

Basic reporting

It is a detailed and important study concerning the dinosaur tracks from China. The revision presented by the authors now allow the publication of the manuscript.
There are only minor changes to be included in the final version that are attached in the PDF of the manuscript.

Experimental design

The study is orignal and it is within the aims and scope of the journal. After the revision the research question is well defined and relevant. It is a rigorous investigation and the methodos are suficient detailed.

Validity of the findings

The findings are relevant to paleontology and conclusions are well state, linked to the original research.

Additional comments

Minor changes:
lines 58 - 71 - The study is very rich in information and conclusions.
This information is not relevant to the reader. I particularly would like to be introduced to what will be presented in the study and the importance of it. Previous references can be included (Xubg et al., 2024 a, Niu and Xing, 2023), but the other information is not necessary.
This change is not mandatory, but evaluate it.

line 183 - Review this number. I think it is wrong

line 1877 - Carvalho, I.S., Leonardi, G., 2024 (instead...Carvalho, I.D.S.

line 2023 - Leonardi, G., Carvalho, I.S., 2021. (instead....Carvalho, I.)

---

## Round 0.3 · accepted · Accept

Thank you for addressing the reviewer's comments. I have checked the current version and I am happy with it. In my opinion, the manuscript is ready for publication.